# Optimal Unconstrained Self-Distillation in Ridge Regression: Strict Improvements, Precise Asymptotics, and One-Shot Tuning

Hien Dang [1]  Pratik Patil [1]  Alessandro Rinaldo [1]

## Abstract

Self-distillation (SD) is the process of retraining a student on a mixture of ground-truth labels and the teacher's own predictions using the same architecture and training data. Although SD has been empirically shown to often improve generalization, its formal guarantees remain limited. We study SD for ridge regression with an unconstrained mixing weight $\xi \in \mathbb{R}$. Conditional on the training data and without any distributional assumptions, we prove that for any squared prediction risk $R$ (including out-of-distribution), the optimally mixed student strictly improves upon the ridge teacher at every regularization level $\lambda$ where the teacher risk is not stationary ($R'(\lambda) \neq 0$). We also characterize the optimal mixing weight $\xi^\star$ in terms of the risk derivative $R'$, showing that it can be negative, which is the case in over-regularized regimes. To quantify SD risk improvements, we derive exact risk asymptotics in the proportional asymptotics regime for general anisotropic covariance and deterministic signals. From a practical standpoint, we propose a consistent one-shot tuning method to estimate $\xi^\star$ without grid search, sample splitting, or refitting. Experiments on real-world datasets and pretrained neural network features support our theory and the one-shot tuning method.

## 1. Introduction

Knowledge distillation, introduced by Buciluă et al. (2006); Ba & Caruana (2014); Hinton et al. (2015), is conventionally used for model compression, transferring knowledge from a large teacher to a smaller student. Recently, this paradigm has been adapted to the setting where the teacher and student share identical architecture and training data,

[1]Department of Statistics and Data Sciences, University of Texas at Austin, Austin, TX 78712, USA. Correspondence to: Hien Dang <hiendang@utexas.edu>.

*Proceedings of the 43$^{rd}$ International Conference on Machine Learning*, Seoul, South Korea. PMLR 306, 2026. Copyright 2026 by the author(s).

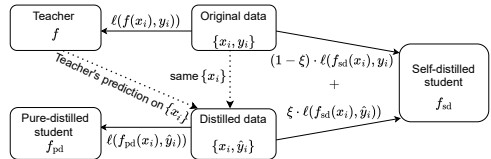

*Figure 1.* Visual illustration of the self-distillation process.

a process known as *self-distillation* (SD) (Furlanello et al., 2018; Zhang et al., 2021). While it may seem counterintuitive that a model would improve by learning from its own predictions, extensive empirical evidence shows that SD can in fact boost generalization (Chen et al., 2017; 2022; Li et al., 2017; Ahn et al., 2019; Li et al., 2021; Gou et al., 2021). Despite these successes, it remains unclear whether and when such improvements can be guaranteed.

Formally, let $f$ be a teacher trained on $\{x_i, y_i\}_{i=1}^n$ using loss $\ell$. SD trains a student $f_{\mathsf{sd}}$ on the *same* data by minimizing a mixed loss that interpolates between ground-truth labels $y_i$ and the teacher's predictions $f(x_i)$ on data features $x_i$:

$$\sum_{i=1}^n (1 - \xi) \cdot \ell(y_i, f_{\mathsf{sd}}(x_i)) + \xi \cdot \ell(f(x_i), f_{\mathsf{sd}}(x_i)), \quad (1)$$

where $\xi$ is the mixing parameter (Lopez-Paz et al., 2015); see Figure 1. When $\xi = 1$, the student learns solely from teacher's predictions, which we call *pure-distillation* (PD) and denote the resulting predictor by $f_{\mathsf{pd}}$.

The mixing parameter $\xi$ balances the influence of the ground-truth labels against the teacher's predictions. Standard distillation methods restrict $\xi$ to the range $[0, 1]$, interpreting the target as a convex combination. Recent work by Das & Sanghavi (2023) shows that this constraint can be suboptimal under high label noise, where the optimal mixing weight $\xi^\star$ may in fact be found to be greater than 1. Motivated by this, we adopt a fully unconstrained perspective and allow $\xi \in \mathbb{R}$, including negative values. Note that setting $\xi = 0$ recovers the teacher predictor, hence optimizing over $\xi$ cannot perform worse than the teacher. With this in mind, we pose the key questions about SD:

(Q1) When does the optimally mixed student $f_{\mathsf{sd}}$ trained using an optimal $\xi^\star \in \mathbb{R}$ strictly outperform the teacher $f$, and how large can the gain be?

(Q2) Can optimal SD from a suboptimal teacher achieve performance comparable to an optimally tuned teacher?

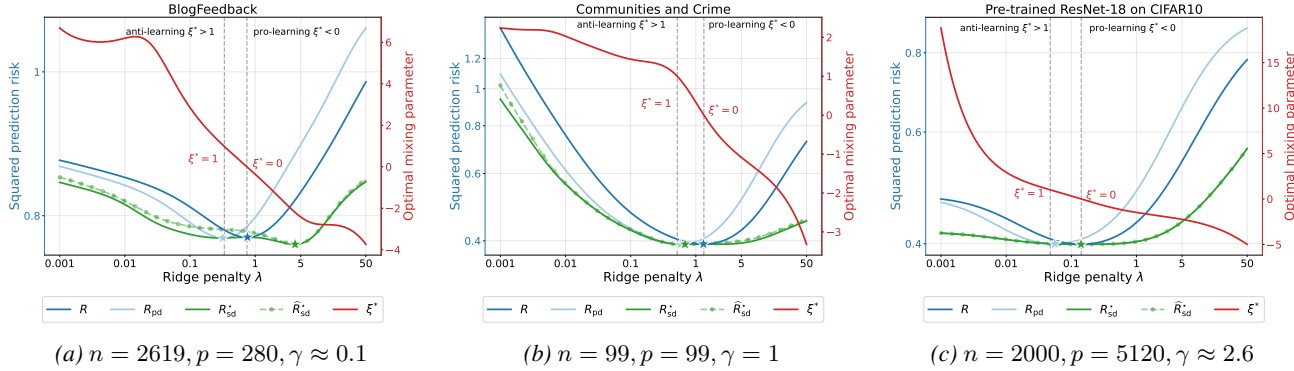

*(a)* $n = 2619, p = 280, \gamma \approx 0.1$   *(b)* $n = 99, p = 99, \gamma = 1$   *(c)* $n = 2000, p = 5120, \gamma \approx 2.6$

*Figure 2.* **Strict improvement of SD risk with unconstrained mixing.** Test squared prediction risk of ridge regression ($R$, in blue), pure-distilled ridge ($R_{\mathsf{pd}}$, in light blue) and optimal self-distilled ridge ($R_{\mathsf{sd}}^\star$, in green) as functions of the ridge penalty $\lambda$. Results are shown on raw features from real-world datasets: BlogFeedback and Communities and Crime datasets, and on pretrained ResNet-18 features. The optimal mixing parameter $\xi^\star(\lambda)$ is in red and the one-shot risk estimate $\widehat{R}_{\mathsf{sd}}^\star(\lambda)$ computed from the training data is shown in green dashed line. Note that $\xi^\star(\lambda)$ lies in $[0, 1]$ only for a narrow range of $\lambda$ and can be strongly negative for large $\lambda$. We also observe that: (i) $R_{\mathsf{sd}}^\star(\lambda)$ is strictly smaller than $R(\lambda)$ at every $\lambda$ that is not the stationary point of $R(\lambda)$, (ii) the sign of $\xi^\star(\lambda)$ is opposite to the sign of $R'(\lambda)$, and (iii) the sign change of $\xi^\star$ happens at the stationary point of $R(\lambda)$. Experiments with $\xi$ restricted to $[0, 1]$ appear in Figure 16.

(Q3) How can we *efficiently tune* the optimal $\xi^\star \in \mathbb{R}$ without computationally expensive grid search?

We provide complete answers to all these questions for ridge regression, a model in which SD admits an explicit affine path (in the response) and the risk of both the teacher and the students can be characterized sharply, capturing the interplay between regularization and distillation. Our main findings are summarized below, and illustrated in Figure 2.

### 1.1. Summary of Paper Contributions and Outline

- **Structural non-asymptotic results (Section 2).** Addressing (Q1), we derive oracle identities for self-distilled ridge that hold conditionally on the observed training data, without any distributional assumptions, and for *any* squared prediction risk (including the out-of-distribution risk). In particular, we show that for every $\lambda > 0$ such that the teacher ridge-path risk $\lambda \mapsto R(\lambda)$ is nonstationary (i.e., $R'(\lambda) \neq 0$), optimal mixing yields a *strict* improvement over the teacher (Theorem 2.2). Addressing (Q2), we provide a curvature-based sufficient condition under which the global minimum over $\lambda$ of the SD risk $R_{\mathsf{sd}}^\star(\lambda)$, obtained using the optimal mixing $\xi^\star(\lambda)$, is strictly smaller than the smallest ridge risk of the teacher (Proposition 2.3).

- **Proportional asymptotic characterization (Section 3).** Returning to (Q1) in the proportional regime – in which the sample and feature sizes $n, p \to \infty$ but the aspect ratio $p/n \to \gamma \in (0, \infty)$ – we derive exact deterministic equivalents for the optimal SD risk and mixing weight under general anisotropic covariance and deterministic signals (Theorem 3.2). These formulas quantify SD gains in terms of $\gamma$, the signal-to-noise ratio, and the signal–covariance alignment. In particular, they characterize precisely when the optimal mixing weight $\xi^\star$ becomes *negative* (Corollary 3.3). Regarding (Q2), under an isotropic random signal we show

that even for extremely under- or over-regularized teachers, optimal SD can closely approach the risk of the optimally tuned ridge predictor (Proposition 3.4).

- **One-shot tuning (Section 4).** Addressing (Q3), we propose a consistent one-shot estimator of $\xi^\star$ based on generalized cross-validation (Theorem 4.1). The method avoids sample splitting and grid search over $\xi$ (and hence does not require any refitting across candidate mixing weights), while remaining consistent in proportional asymptotics. We validate our theory and tuning methodology through extensive experiments on synthetic data, UCI regression benchmarks, and pretrained ResNet features.

- **Extensions and variants (Section 5).** We study some variants of the SD ridge framework, including multi-round distillation with a risk monotonicity property (Section E.1), self-distillation using teacher predictions on fresh features, which we show to be dominated by the same-$X$ setup in an isotropic setting (Section E.2), and extensions to generalized ridge and kernel ridge regression (Section E.3).

### 1.2. Related Works and Comparisons

**Fixed-design analyses.** Early theoretical work on distillation focused on when and how well a student can mimic a teacher (Phuong & Lampert, 2019; Ji & Zhu, 2020); for regression problems, this has typically been addressed using fixed-design. In kernel ridge regression, Mobahi et al. (2020) show that repeated SD shrinks the effective function class, resembling an increase in regularization; their emphasis is on the implicit regularization induced by SD rather than on guaranteeing performance improvements. In ridge regression, Das & Sanghavi (2023) analyze the bias–variance trade-off induced by the mixing weight $\xi$ and give conditions under which the *globally* optimal SD risk (obtained by optimizing over $\lambda$) can go below the optimally tuned teacher ridge risk. Pareek et al. (2024) extend this line of work to

*Table 1.* **Settings overview in the self-distillation ridge/ridgeless literature**. Unless stated otherwise, risks in the proportional asymptotics setting are in-distribution squared prediction risks.

| Paper | Focus | | Nonasymptotic setting | | | Proportional asymptotics | | | Weights | | Multi-rounds |
|---|---|---|---|---|---|---|---|---|---|---|---|
| | Type | Family | Design | Response | Risk | Features | Signal | Model | Range | Tuning | |
| DS23 | SD | Ridge | Fixed | Linear | Estimation | | | | $\mathbb{R}$ | | |
| PDO24 | SD | Ridge | Fixed | Linear | Fixed-X prediction | | | | $\mathbb{R}$ | Split CV (three refits) | Yes |
| IGT+25 | PD | Ridgeless | Random | Linear | In-distribution prediction | | | | $\{1\}$ | | |
| MH25 | PD | Ridge (generalized) | | | | Isotropic | Isotropic | Linear | $\{1\}$ | | |
| GBS25 | SD | Ridge (infinite rounds) | | | | Anisotropic | Deterministic | Linear | $[0, 1]$ | | Infinite rounds only |
| Ours | SD | Ridge | Random | Any | Any squared prediction | Anisotropic | Deterministic | Any | $\mathbb{R}$ | GCV (one-shot) | Yes |

multiple SD rounds, quantifying gains under an alignment condition when $\lambda$ and $\xi$ are tuned in each round. A key gap in these analyses is the question of pointwise improvement: whether SD can *strictly* improve a mis-regularized teacher at a fixed, suboptimal $\lambda$, which we show in Section 2. Such pointwise gains are important because finding global optimal $\lambda$ is often challenging; for example, Stephenson et al. (2021) show that the leave-one-out cross-validation loss is generally neither convex nor even quasi-convex in $\lambda$.

A further distinction is the data model. Both the linear regression analysis in Das & Sanghavi (2023); Pareek et al. (2024) operate under fixed design and well-specified linear response models. While analytically convenient, fixed-design analyses do not capture the contribution of test-feature randomness to prediction error, which is central in high-dimensional generalization (Hastie et al., 2022). In contrast, our structural results hold conditionally on the observed training data under random design and for *any* squared prediction risk, including out-of-distribution risk. We also work in proportional random-design asymptotics ($n, p \to \infty$ with $p/n \to \gamma$) in Section 3 and provide exact characterizations of generalization effects driven by feature and signal structures.

**Random-design analyses.** Closer to our setting, Emrullah Ildiz et al. (2025) analyze KD under random design with anisotropic covariance, focusing on scaling laws and "weak-to-strong" generalization, a phenomenon where a strong student model is supervised by a weaker teacher (Burns et al., 2023). They provide nonasymptotic characterizations of pure-distilled ridge ($\xi = 1$ in our notation) and its scaling behavior. They primarily focus on the *ridgeless* case (minimum $\ell_2$-norm interpolator where $\lambda \to 0^+$). While interesting, such analysis obscures the interaction between explicit ridge penalty ($\lambda > 0$) and distillation weight $\xi$, which is central to our results. In particular, we identify over-regularized regimes where optimal SD requires a *negative* optimal mixing $\xi^\star$ to account for excessive shrinkage and thus strictly improve the teacher. In contrast, in pure-distillation ridge with $\lambda > 0$, the student can only improve

upon the teacher for a range of small $\lambda$, and can fail to improve under over-regularization (or even being worse than the teacher) (Moniri & Hassani, 2025); see Figure 2. Saglietti & Zdeborová (2022) study the performance of the PD student in weak-to-strong regime with logistic regression.

**Unconstrained mixing and "anti-learning".** Standard KD restricts $\xi$ to $[0, 1]$. Das & Sanghavi (2023) challenge this convention in the context of label noise, showing that in high-noise regimes, the optimal weight satisfies $\xi > 1$. This corresponds to a negative weight on the noisy ground truth, a phenomenon they term "anti-learning". Javanmard et al. (2025) further support this intuition in the context of binary classification, deriving a non-linear Bayes-optimal aggregator that effectively subtracts noisy labels during retraining. Unlike Javanmard et al. (2025), who rely on Approximate Message Passing for generalized linear models, we derive closed-form asymptotic risks for the linear estimator. This tractability allows us to rigorously classify the optimal mixing into three regimes: interpolation ($0 \le \xi \le 1$), extrapolation ("anti-learning" $\xi > 1$), and correcting for over-regularization ($\xi < 0$), which we term as "pro-learning".

**Tuning and risk estimation.** A notable gap in existing theoretical results about SD is the choice of the optimal mixing hyperparameter $\xi$ in practice. Most analyses assume oracle access to population risks or detailed spectral knowledge. In practice, cross-validation over $\xi$ is computationally costly due to grid search and repeated refitting, and it can be statistically inefficient in high dimensions due to sample splitting (see, e.g., Rad & Maleki (2020)). Building on consistent risk estimation for high-dimensional ridge/ridgeless regression (see, e.g., Patil et al., 2021; 2022b; Wei et al., 2022; Han & Xu, 2023; Bellec et al., 2025; Koriyama et al., 2024 and references therein), we propose a one-shot generalized cross-validation estimator for SD ridge and prove its consistency in proportional asymptotics (Theorem 4.1). This enables data-efficient selection of unconstrained $\xi$ without grid search over candidate mixing weights or hold-out sets.

Overall, relative to prior SD ridge analyses, we provide the following novel results: (i) pointwise strict improvement

guarantees at any nonstationary $\lambda > 0$ together with a sign characterization of $\xi^\star$ (Theorem 2.2); (ii) exact proportional-asymptotic risk characterizations under anisotropic feature covariance and deterministic signals (Theorem 3.2); and (iii) a consistent one-shot GCV-based tuning method (Theorem 4.1). At the same time, our settings are more general than those considered in the literature so far: (a) a nonasymptotic setting with no distributional assumptions and any squared prediction risk; and (b) proportional asymptotics with general feature covariance and deterministic signals, without imposing a well-specified response model. A compact comparison of settings and assumptions across several related works is provided in Table 1.

## 2. Structural Non-Asymptotic Results

This section provides *deterministic* "structural" identities for optimal self-distillation under *any squared prediction risk* (including the out-of-distribution risk). These results hold conditionally on the observed training data and do not require any distributional assumptions.

### 2.1. Self-Distillation with Ridge Regression

We denote the original dataset with $n$ samples and $p$ covariates as $\mathcal{D} = (X, y) \in \mathbb{R}^{n \times p} \times \mathbb{R}^n$, where $X = [x_1 \dots x_n]^\top$ is the design matrix and $y$ the response vector. Given a regularization parameter $\lambda > 0$, let $x \in \mathbb{R}^p \mapsto f_\lambda(x) \in \mathbb{R}$ denote the *teacher* ridge predictor trained on $\mathcal{D}$, given by:

$$x \in \mathbb{R}^p \mapsto x^\top \operatorname*{argmin}_{\beta \in \mathbb{R}^p} \{ \|y - X\beta\|_2^2/n + \lambda\|\beta\|_2^2 \}.$$

Let $\widehat{y}_\lambda = f_\lambda(X) \in \mathbb{R}^n$ denote the training predictions of $f_\lambda$, where $\widehat{y}_{\lambda,i} = f_\lambda(x_i)$. We define the *pure-distillation* (PD) predictor $f_{\mathsf{pd},\lambda}(x)$ as the ridge predictor training on the "distilled" dataset $(X, \widehat{y}_\lambda)$. For any *mixing parameter* $\xi \in \mathbb{R}$, the *self-distilled* (SD) predictor $x \mapsto f_{\mathsf{sd},\lambda}(x)$ is obtained using the mixed loss:

$$x^\top \operatorname*{argmin}_{\beta \in \mathbb{R}^p} \{ (1-\xi)\|y - X\beta\|_2^2/n + \xi\|\widehat{y}_\lambda - X\beta\|_2^2/n + \lambda\|\beta\|_2^2 \}. \tag{2}$$

By linearity of the ridge map in the response, the resulting predictor satisfies the affine identity (details in Section B):

$$f_{\mathsf{sd},\lambda}(x) = (1-\xi) \cdot f_\lambda(x) + \xi \cdot f_{\mathsf{pd},\lambda}(x). \tag{3}$$

For a test point $(x_0, y_0)$ (potentially out-of-distribution), we measure the performance of any (possibly data dependent) predictor $f$ using squared prediction risk:

$$R(f) := \mathbb{E}[(y_0 - f(x_0))^2 \mid \mathcal{D}]. \tag{4}$$

For a given $\lambda > 0$, let $R(\lambda)$, $R_{\mathsf{pd}}(\lambda)$, and $R_{\mathsf{sd}}(\lambda, \xi)$ denote the prediction risks of the teacher predictor $f_\lambda$, the PD predictor $f_{\mathsf{pd},\lambda}$ and the SD predictor $f_{\mathsf{sd},\lambda}$, respectively.

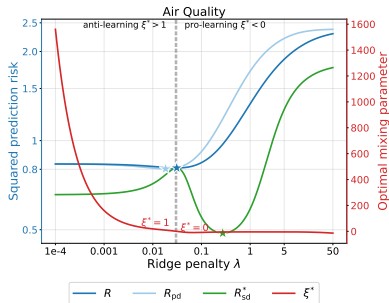

*Figure 3.* **Out-of-distribution SD risk improvement** Test prediction risk of ridge and optimal SD ridge on Air Quality dataset. SD yields strict improvements across $\lambda$ and achieves a substantially smaller global minimum.

### 2.2. Optimal SD Risk Decomposition

We first provide a useful decomposition of the optimal SD risk. Let $C$ denote the correlation between the residual errors of the teacher $f_\lambda$ and of the PD predictor $f_{\mathsf{pd},\lambda}$, defined as:

$$C(\lambda) := \mathbb{E}[(y_0 - f_\lambda(x_0))(y_0 - f_{\mathsf{pd},\lambda}(x_0)) \mid \mathcal{D}]. \tag{5}$$

Define $\xi^\star(\lambda)$ as the minimizer of the SD risk $R_{\mathsf{sd}}(\lambda, \xi)$ over $\xi \in \mathbb{R}$, and let $R_{\mathsf{sd}}^\star(\lambda)$ be the corresponding minimum risk.

**Proposition 2.1** (Optimal SD risk decomposition). *Assume that $R + R_{\mathsf{pd}} - 2C \neq 0$. Then, we have the decompositions:*[1]

$$\xi^\star = \frac{R - C}{R + R_{\mathsf{pd}} - 2C}, \quad R_{\mathsf{sd}}^\star = R - \frac{(R - C)^2}{R + R_{\mathsf{pd}} - 2C}. \tag{6}$$

*In particular, $R_{\mathsf{sd}}^\star \leq R$ for all $\lambda$, and $\xi^\star$ may be negative.*

The SD risk improvement $R - R_{\mathsf{sd}}^\star$ depends on: (i) the difference $R - C$ between the teacher risk and the residual correlation between the teacher and the PD predictors (which also drives the sign of $\xi^\star$), and (ii) the non-negative quantity $R + R_{pd} - 2C$, which can be written as the separation between their predictions on a test input (see Section B):

$$R + R_{\mathsf{pd}} - 2C = \mathbb{E}[(f_\lambda(x_0) - f_{\mathsf{pd},\lambda}(x_0))^2 \mid \mathcal{D}] \geq 0. \tag{7}$$

This term only vanishes when $f_\lambda \equiv f_{\mathsf{pd},\lambda}$ on the test distribution, in which case $R_{\mathsf{sd}}(\xi) \equiv R$ for all $\xi$, and any $\xi$ is optimal. Thus, assuming $R + R_{\mathsf{pd}} - 2C \neq 0$ is innocuous.

### 2.3. Strict Pointwise Improvement and Sign of $\xi^\star(\lambda)$

Building on Proposition 2.1, we next characterize when optimal SD is *strictly* better than the teacher, and determine the sign of the optimal mixing weight in terms of the derivative of the teacher risk.

**Theorem 2.2** (Strict improvement and sign rule). *Assume that $R + R_{\mathsf{pd}} - 2C \neq 0$. Then,*

$$\xi^\star = \frac{-(\lambda/2) \cdot R'}{R + R_{\mathsf{pd}} - 2C}, \quad R_{\mathsf{sd}}^\star = R - \frac{(\lambda^2/4) \cdot (R')^2}{R + R_{\mathsf{pd}} - 2C}. \tag{8}$$

---

[1] For space reasons, we will often omit dependence of the risk and correlation functions and the optimal mixing parameter on $\lambda$.

*Thus, if $R' \neq 0$, then $R_{sd}^\star < R$ and $\text{sign}(\xi^\star) = -\text{sign}(R')$.*

Plainly, optimal self-distillation strictly improves the teacher at every *nonstationary* point of the teacher ridge risk (i.e., whenever $R'(\lambda) \neq 0$). Moreover, $\xi^\star(\lambda)$ is positive in under-regularized regimes (where $R'(\lambda) < 0$) and negative in over-regularized regimes (where $R'(\lambda) > 0$), a behavior we refer to as "pro-learning" to contrast it with "anti-learning" (Das & Sanghavi, 2023). While prior work has noted improvements of optimal SD, to our knowledge, Theorem 2.2 is the first result establishing pointwise strict improvements for optimal self-distilled ridge, along with the sign rule for optimal mixing, at any nonstationary $\lambda$, for any squared prediction risk and without distributional assumptions. The strict improvement and sign rule are consistently observed in Figures 2 and 3.

### 2.4. Can Self-Distillation Beat Optimally Tuned Ridge?

Theorem 2.2 above guarantees improvements at any subopti­mal $\lambda$, but it implies that, at the optimal ridge penalty $\lambda^\star$, $R_{sd}^\star(\lambda^\star) = R(\lambda^\star)$. It is therefore natural to study whether the *global* minimum of the SD risk curve is strictly lower than the teacher's global minimum. We provide a sufficient condition based on the local curvature of the risk profile.

**Proposition 2.3** (Curvature test at the ridge-optimal $\lambda$). *Let $\lambda^\star \in \arg\min_{\lambda>0} R(\lambda)$. If the following curvature test at $\lambda^\star$ holds, i.e. if*

$$R(\lambda^\star) + R_{pd}(\lambda^\star) - 2C(\lambda^\star) < \frac{\lambda^{\star 2}}{2} R''(\lambda^\star), \quad (9)$$

*then $R_{sd}^\star$ has negative curvature at $\lambda^\star$, and therefore*

$$\min_{\lambda>0} R_{sd}^\star(\lambda) < \min_{\lambda>0} R(\lambda). \quad (10)$$

Proposition 2.3 formalizes a phenomenon visible in our ex­periments: the curves $R(\lambda)$ and $R_{sd}^\star(\lambda)$ necessarily *touch* at ridge-optimal $\lambda^\star$, but $R_{sd}^\star$ may bend downward elsewhere and achieve a strictly smaller global minimum. The curva­ture test (9) is purely structural (no distributional assump­tions). This can be seen as a substantial generalization of Theorem 3.8 in Das & Sanghavi (2023) as our result holds generally for any squared risk (rather than only the esti­mation risk) and does not require the response $y$ follows a well-specified linear model.

As mentioned above, the results in this section are concerned with the out-of-distribution prediction risk, where $(x_0, y_0)$ may be drawn from a distribution different from that of the training samples. In Figure 3, we showcase a real-data example on the Air Quality dataset with a distribution shift between the training and test sets due to time-dependent effects. We also numerically verify the curvature test in Table 2 for all of our real-data experiments and observe exact agreement in all cases.

*Table 2.* Curvature test at the ridge-optimal $\lambda^\star$. Global gain is ticked when the difference between the minimum SD risk (over the fine grid of $\lambda$) and the minimum teacher risk exceeds $10^{-3}$.

| Dataset | Eqn. (9) satisfies? | Global gain? |
|---|---|---|
| BF (Fig. 2a) | ✓ | ✓ |
| CC (Fig. 2b) | ✗ | ✗ |
| CIFAR10 (Fig. 2c) | ✗ | ✗ |
| AQ (Fig. 3) | ✓ | ✓ |

While the results above guarantee a strict improvement at every nonstationary $\lambda$ and require no distributional assump­tions, they do not quantify the *magnitude* of the gain, i.e., how much smaller $R_{sd}^\star(\lambda)$ can be than $R(\lambda)$. In the next section, we consider standard distributional assumptions in the proportional-asymptotics literature and derive determin­istic equivalents for these risks and their improvements as explicit functions of the problem parameters.

## 3. Proportional Asymptotic Results

We now refine the structural identities from Section 2 by deriving *deterministic* limits for the optimal SD risk $R_{sd}^\star(\lambda)$ and the optimal mixing weight $\xi^\star(\lambda)$ in the proportional asymptotics regime $n, p \to \infty$ with $p/n \to \gamma \in (0, \infty)$. Our goals are to: (i) obtain computable asymptotic expres­sions for the SD risk $R_{sd}^\star(\lambda)$ and its gain over $R(\lambda)$, and (ii) quantify how they depend on aspect ratio $\gamma$, signal-to-noise ratio SNR and signal–covariance alignment (to be defined).

### 3.1. Data Assumptions

We assume $\{(x_i, y_i)\}_{i=1}^n$ are drawn i.i.d. from a distribution $P_{x,y}$ satisfying the following:

**Assumption 3.1** (Data distribution). The covariate vector $x \sim P_x$ is such that $x = \Sigma^{1/2} z$, where $\Sigma \in \mathbb{R}^{p \times p}$ is a deterministic, positive semi-definite covariance matrix with eigenvalues uniformly bounded away from 0 and $\infty$, and $z \in \mathbb{R}^p$ contains i.i.d. entries with mean 0, variance 1, and uniformly bounded $(4 + \mu)$-th moment for some $\mu > 0$. The response $y \sim P_y$ has mean 0 and uniformly bounded $(4 + \nu)$-th moment, for some $\nu > 0$.

The feature structure imposed in Assumption 3.1 is standard in random matrix analyses of high-dimensional regression; see, e.g., (Bai & Silverstein, 2010; Bartlett et al., 2021; Misiakiewicz & Montanari, 2024). We do *not* assume a well-specified linear model for the response $y$. Instead, we parameterize $P_{x,y}$ by $(\Sigma, \beta, \sigma^2)$, where $\beta := \Sigma^{-1}\mathbb{E}[xy]$ de­notes the parameter of the population linear $L_2$ projection of $y$ onto $x$ and $\sigma^2 := \text{Var}(y - x^\top \beta)$ denotes the correspond­ing residual variance. Finally, we denote the signal energy by $r^2 := \|\beta\|_2^2$ and $\text{SNR} := r^2 / \sigma^2$. Throughout this section, we focus on the in-distribution squared prediction risk in which the test point $(x_0, y_0)$ in (4) is an independent copy drawn from the same distribution as the training samples.

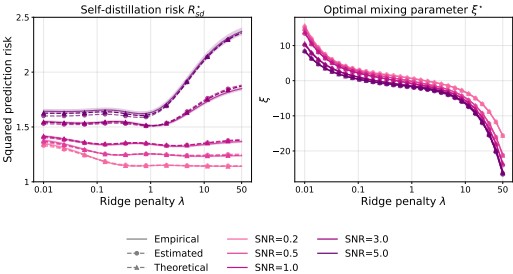

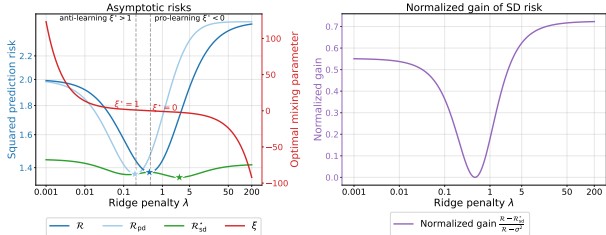

*Figure 4.* **Theoretical versus empirical risks.** Asymptotic SD risk and optimal mixing parameter over various SNR ratios. Empirical curves are averaged over 30 numerical simulations. Estimated curves are obtained using proposed tuning method (Section 4, averaged over 30 runs) and theoretical curves are from Theorem 3.2. $n = 400$, $p = 200$, $\sigma^2 = 1$ and $r^2 = \sigma^2\text{SNR}$. Covariance $\Sigma$ is AR1, deterministic signal aligns with the top 10% eigenvalues of $\Sigma$, with alignment factor 0.9 (details in Section G).

### 3.2. Asymptotics of Optimal Risk and Mixing Weight

By the structural identity (6), for any given $\lambda > 0$ the optimal SD risk is a function of the three random variables $R$, $R_{\text{pd}}$, and $C$. Under proportional asymptotics, these quantities concentrate and converge to deterministic limits, which we denote by $\mathscr{R}$, $\mathscr{R}_{\text{pd}}$, and $\mathscr{C}$, respectively, which depend only on $(\Sigma, \beta, \sigma^2)$, along with $\gamma$ and $\lambda$. To define these functions, we next introduce some parameters.

Define the scalar $\kappa = \kappa(\lambda) > 0$ as the unique solution to:

$$\kappa = \lambda + \gamma\kappa\,\text{tr}(\Sigma(\Sigma + \kappa I_p)^{-1})/p. \qquad (11)$$

Let $G := (\Sigma + \kappa I_p)^{-1}$. Define following covariance-moment trace and signal-covariance alignment functionals:

$$t_k := \gamma\,\text{tr}(\Sigma^2 G^k)/p, \quad q_k := \beta_\star^\top G^k \Sigma\,\beta_\star, k \in \{2, 3, 4\}. \qquad (12)$$

Finally, define the coefficients: $a_2 := bE^2 + b^4\kappa^2\lambda^2 t_4 + b^5\kappa^2\lambda^2 t_3^2$, $a_3 := 2b^2\kappa\lambda E$, $a_4 := b^3\kappa^2\lambda^2$, and the variance trace limits: $u_2 := t_2 b$, $u_3 := t_3 b^3$, $u_4 := t_4 b^4 + 2t_3^2 b^5$, where $b = (1 - t_2)^{-1}$ and $E = \kappa - b\lambda + b^2\kappa\lambda t_3$. We are now ready to state our main result of this section:

**Theorem 3.2** (Risk asymptotics). *Under Assumption 3.1, as $n, p \to \infty$ with $p/n \to \gamma \in (0, \infty)$, we have*

$$\xi^\star \xrightarrow{\text{p}} \frac{\mathscr{R} - \mathscr{C}}{\mathscr{R} + \mathscr{R}_{\text{pd}} - 2\mathscr{C}}, \quad R_{\text{sd}}^\star \xrightarrow{\text{p}} \mathscr{R} - \frac{(\mathscr{R} - \mathscr{C})^2}{\mathscr{R} + \mathscr{R}_{\text{pd}} - 2\mathscr{C}},$$

*where the individual component limits are as follows*

$$\mathscr{R} := \sigma^2 + \kappa^2 b\, q_2 + \sigma^2\, u_2,$$
$$\mathscr{C} := 2\kappa^2 b\, q_2 - \kappa bE\, q_2 - \kappa^2 b^2\lambda\, q_3 + \sigma^2\, (u_2 - \lambda u_3 + 1),$$
$$\mathscr{R}_{\text{pd}} := \sigma^2 + 4\kappa^2 b\, q_2 - 4\kappa bE\, q_2 - 4\kappa^2 b^2\lambda\, q_3$$
$$+ (a_2 q_2 + a_3 q_3 + a_4 q_4) + \sigma^2\, (u_2 - 2\lambda u_3 + \lambda^2 u_4).$$

To our knowledge, this is the first precise characterization of the optimal SD risk in the proportional asymptotics regime

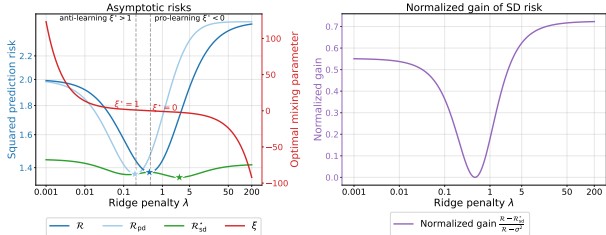

*Figure 5.* **Asymptotic gain over the teacher.** Asymptotic risks and gain curve. Same setting as Figure 4 with $r^2 = \sigma^2 = 1$.

that holds under the general setting of anisotropic covariance and deterministic signal. The spectrum of $\Sigma$ enters through resolvent traces such as $t_2, t_3, t_4$, while the signal enters through the alignment functionals $q_2, q_3, q_4$. As illustrated in Figure 4, the theoretical predictions closely match empirical risks even for moderate $n$ and $p$.

In the isotropic-signal specialization when $\beta \sim \mathcal{N}(0, (r^2/p)I_p)$, these quadratic forms collapse to traces; see Section C.7. The strict improvement of self-distilled risk and the sign rule of $\xi^\star$ are very apparent in a such setting:

**Corollary 3.3.** *Under Assumption 3.1 with $\beta \sim \mathcal{N}(0, (r^2/p)I_p)$, we have*

(a) $\mathscr{R}_{\text{sd}}^\star(\lambda) < \mathscr{R}(\lambda)$ *for all $\lambda \neq \lambda^\star := \gamma\frac{\sigma^2}{r^2}$, and*

(b) $\xi^\star(\lambda) < 0$ *iff $\lambda > \lambda^\star$, and $\xi^\star(\lambda) > 0$ iff $\lambda < \lambda^\star$, i.e., $\text{sign}(\xi^\star(\lambda)) = \text{sign}(\lambda^\star - \lambda)$ for all $\lambda \neq \lambda^\star$.*

Thus, in the isotropic-signal setting, optimal SD strictly improves upong ridge at every suboptimal value of $\lambda$, and it matches the (statistically optimal, for the isotropic design) ridge risk only at the ridge-optimal value $\lambda^\star$. Moreover, the sign of the optimal mixing weight is determined solely by whether the ridge model is under-regularized ($\lambda < \lambda^\star$) or over-regularized ($\lambda > \lambda^\star$). Simulation studies show in Figure 5 are in close agreement with these findings: (i) $\mathscr{R}_{\text{sd}}^\star(\lambda)$ lies below $\mathscr{R}(\lambda)$ for all nonstationary $\lambda$, (ii) $\xi^\star(\lambda)$ flips sign at $\lambda^\star$, and (iii) the gain is largest in strongly under- or over-regularized regimes. Additional illustrations for other covariance and signal geometries are provided in Section F.3.

### 3.3. Self-Distillation Risks with Extreme Regularization

Next, we study how close optimal SD can get to the best possible predictor when the teacher is *extremely* under- or over-regularized. We focus on the isotropic design and isotropic signal setting ($\Sigma = I_p$ and $\beta \sim \mathcal{N}(0, (r^2/p)I_p)$), where the ridge-optimal predictor is Bayes-optimal and its asymptotic risk $\mathscr{R}^\star$ is known in closed form (Dobriban & Wager, 2018; Hastie et al., 2022). We compare the limiting SD risk $\mathscr{R}_{\text{sd}}^\star(\lambda)$ to $\mathscr{R}^\star$ as $\lambda \to 0$ and $\lambda \to \infty$.

**Proposition 3.4** (Comparison with the optimal ridge). *Assume $\Sigma = I_p$ and $\beta \sim \mathcal{N}(0, (r^2/p)I_p)$ and let $S^\star(\text{SNR}, \gamma) := \frac{1}{2\gamma}(\text{SNR}(\gamma - 1) - \gamma +$*

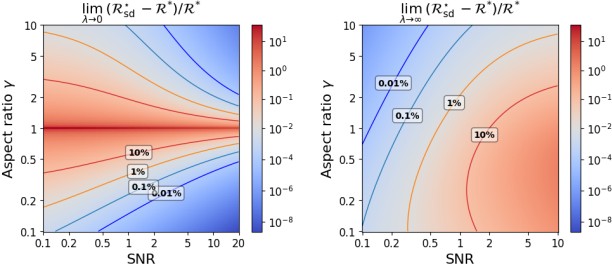

*Figure 6.* **Approximating the best predictor.** Percentage difference between $\mathscr{R}^\star_{\mathsf{sd}}(\text{SNR}, \gamma)$ and the Bayes-optimal ridge risk $\mathscr{R}^\star(\text{SNR}, \gamma)$ in the isotropic design and signal setting.

$$\sqrt{4\text{SNR}\gamma^2 + (\text{SNR}(\gamma-1)-\gamma)^2}).\ \textit{Then,}$$

$$\lim_{\lambda \to 0} \frac{\mathscr{R}^\star_{\mathsf{sd}}(\lambda) - \mathscr{R}^\star}{\mathscr{R}^\star} =$$
$$\begin{cases} \frac{\text{SNR}(1-\gamma)^2+\gamma}{(\text{SNR}(1-\gamma)^3+\gamma(1-\gamma^2))(S^\star+1)} - 1, & \gamma \in (0,1), \\ \frac{\text{SNR}^2(\gamma-1)^4+\text{SNR}\gamma(2\gamma+1)(\gamma-1)^2+\gamma^4}{(\text{SNR}\gamma(\gamma-1)^3+\gamma^2(\gamma^2-1))(S^\star+1)} - 1, & \gamma \in (1,\infty), \end{cases}$$

$$\lim_{\lambda \to \infty} \frac{\mathscr{R}^\star_{\mathsf{sd}}(\lambda) - \mathscr{R}^\star}{\mathscr{R}^\star} = \frac{\text{SNR}^2\gamma + \text{SNR}(2\gamma+1)+\gamma}{(\text{SNR}(\gamma+1)+\gamma)(S^\star+1)} - 1.$$

Proposition 3.4 expresses the relative suboptimality of optimal SD under extreme regularization scenarios explicitly in terms of $(\gamma, \text{SNR})$ under the isotropic design and signal setting. The resulting percentage gaps are illustrated in Figure 6. Across a wide range of $(\gamma, \text{SNR})$, optimal SD can be remarkably close to Bayes-optimal performance even when the teacher is extremely under- or over-regularized (e.g., within $0.01\%$ for $(\text{SNR}, \gamma) = (2, 0.2)$ as $\lambda \to 0$). The heatmaps also suggest a qualitative dichotomy: for low SNR, over-regularization is preferable across $\gamma$, while for high SNR, under-regularization is preferable except near the square design when $\gamma \approx 1$. For comparison, we also report the analogous extreme-$\lambda$ gaps between the original ridge risk $\mathscr{R}(\lambda)$ and the ridge-optimal risk $\mathscr{R}^\star$ in Section F.4, along with experiments with other covariance structures.

## 4. One-Shot Tuning and Risk Estimation

The results in Section 3 quantify the benefits of optimal self-distillation. However, these deterministic equivalents in Theorem 3.2 depend on population quantities (e.g., the covariance spectrum and signal–covariance alignment), which are unknown in practice. For the practical question for tuning the mixing parameter $\xi$, a standard method involves grid search combined with split cross-validation (CV), but this is computationally expensive (requiring repeated retrainings over candidate $\xi$) and statistically inefficient in high dimensions due to sample splitting (see, e.g., Rad & Maleki (2020)). In this section, we propose a computationally efficient one-shot procedure that estimates $\xi^\star(\lambda)$ from the training data without grid search or hold-out sets.

### 4.1. Risk Estimators via Generalized Cross-Validation

Our starting point is the closed-form identity for optimal SD given in Proposition 2.1, which expresses $\xi^\star(\lambda)$ and $R^\star_{\mathsf{sd}}(\lambda)$ in terms of the teacher risk $R(\lambda)$, the PD risk $R_{\mathsf{pd}}(\lambda)$, and the residual correlation term $C(\lambda)$. We construct estimators for these three terms using training data by means of generalized cross-validation (GCV) and its variants, along with plug-in estimation of $\xi^\star(\lambda)$.

Write the fitted values of the ridge teacher and the pure-distilled predictor as $\widehat{y}_\lambda := f_\lambda(X)$ and $\widehat{y}_{\mathsf{pd},\lambda} := f_{\mathsf{pd},\lambda}(X)$. Our estimators involve the notion of *effective degrees of freedom* from the theory of statistical optimism (Efron, 1983; 1986). The degrees of freedom $\mathsf{df}(f)$ of a (possibly nonlinear) predictor $f$ is measured by the trace of the operator $y \mapsto (\partial/\partial y) f(y)$ (Hastie & Tibshirani, 1990; Stein, 1981). In particular, for linear smoothers, this corresponds to the trace of the smoothing matrix (the so-called "hat" matrix).

Set $\mathsf{df}_\lambda := \mathsf{df}(f_\lambda)$ and $\mathsf{df}_{\mathsf{pd},\lambda} := \mathsf{df}(f_{\mathsf{pd},\lambda})$ and define the GCV-corrected residuals

$$\widehat{r}_\lambda := \frac{y - \widehat{y}_\lambda}{1 - \mathsf{df}_\lambda/n}, \quad \widehat{r}_{\mathsf{pd},\lambda} := \frac{y - \widehat{y}_{\mathsf{pd},\lambda}}{1 - \mathsf{df}_{\mathsf{pd},\lambda}/n}. \quad (13)$$

We estimate the teacher and PD prediction risks, as well as their residual correlation term, by

$$\widehat{R}(\lambda) := \frac{\|\widehat{r}_\lambda\|_2^2}{n}, \widehat{R}_{\mathsf{pd}}(\lambda) := \frac{\|\widehat{r}_{\mathsf{pd},\lambda}\|_2^2}{n}, \widehat{C}(\lambda) := \frac{\langle \widehat{r}_\lambda, \widehat{r}_{\mathsf{pd},\lambda}\rangle}{n}, \quad (14)$$

respectively. Here $\widehat{R}(\lambda)$ coincides with the standard ridge GCV estimator, while $\widehat{R}_{\mathsf{pd}}(\lambda)$ and $\widehat{C}(\lambda)$ extend the same df correction principle to the PD and cross-term quantities appearing in Proposition 2.1.

### 4.2. One-Shot Estimators for $\xi^\star$ and $R^\star_{\mathsf{sd}}$

Plugging (14) into the exact identity (6) yields

$$\widehat{\xi}^\star := \frac{\widehat{R} - \widehat{C}}{\widehat{R} + \widehat{R}_{\mathsf{pd}} - 2\widehat{C}}, \ \widehat{R}^\star_{\mathsf{sd}} := \widehat{R} - \frac{(\widehat{R} - \widehat{C})^2}{\widehat{R} + \widehat{R}_{\mathsf{pd}} - 2\widehat{C}}. \quad (15)$$

We note that $\widehat{R} + \widehat{R}_{\mathsf{pd}} - 2\widehat{C} = \|\widehat{r}_\lambda - \widehat{r}_{\mathsf{pd},\lambda}\|_2^2/n$. We next show that our estimators are consistent.

**Theorem 4.1** (Consistency of one-shot SD tuning). *Under Assumption 3.1, as $n, p \to \infty$ with $p/n \to \gamma \in (0,\infty)$, for each fixed $\lambda > 0$, we have*

$$\widehat{\xi}^\star(\lambda) - \xi^\star(\lambda) \xrightarrow{\mathrm{p}} 0, \quad \widehat{R}^\star_{\mathsf{sd}}(\lambda) - R^\star_{\mathsf{sd}}(\lambda) \xrightarrow{\mathrm{p}} 0. \quad (16)$$

Compared with grid-search CV over $\xi$, the one-shot procedure has two key advantages: (i) CV requires retraining a student model for each candidate $\xi$, whereas (15) selects $\widehat{\xi}^\star(\lambda)$ in closed form from a single set of fitted quantities at the given $\lambda$. (ii) Split CV reduces the effective training sample size (e.g., to $4/5$) and suffers from nonzero bias in

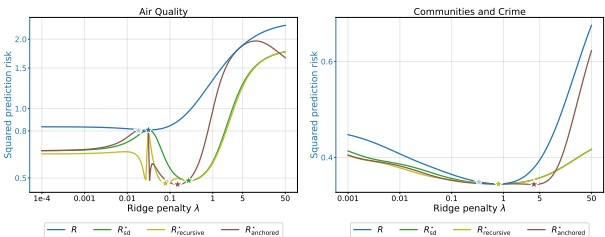

*Figure 7.* **Recursive versus anchored multi-round self-distillation.** Test risks of the teacher ridge ($R$, in blue, trained using $y^{(0)}$), one-round ($k = 1$) optimal self-distilled ridge ($R^\star_{\mathsf{sd}}$, in green, trained using $y^{(0)}$ and $\widehat{y}^{(0)}$), two-round ($k = 2$) recursive ($R^\star_{\mathsf{recursive}}$, in olive, trained using $y^{(1)}$ and $\widehat{y}^{(1)}$) and anchored ($R^\star_{\mathsf{anchored}}$, in brown, trained using $y^{(0)}$ and $\widehat{y}^{(1)}$) self-distillation. Recursive mixing is monotone when $\xi_k$ is optimized each round: the two-round risk curve $R^\star_{\mathsf{recursive}}(\lambda)$ uniformly dominates the optimal one-round $R^\star_{\mathsf{sd}}(\lambda)$. Anchored mixing can be nonmonotone: the two-round risk curve $R^\star_{\mathsf{anchored}}(\lambda)$ may be larger than $R^\star_{\mathsf{sd}}(\lambda)$.

high-dimensional settings where $p$ is comparable to $n$; in contrast, the one-shot estimators use all $n$ samples without hold-out sets.

### 4.3. Real Data Experiments

To illustrate the utility of the one-shot tuning method, we apply (15) across a range of $\lambda$ values on several real datasets. For regression, we consider UCI BlogFeedback and Communities and Crime. For classification, we apply ridge regression on pretrained neural network features (ResNet-18/34) for CIFAR10 and CIFAR100.

Across these tasks, Figures 2 and 14 show that $\widehat{R}^\star_{\mathsf{sd}}(\lambda)$ closely tracks the test risk of the optimally distilled student, particularly in settings with small train and test discrepancy (e.g., CIFAR dataset benchmarks). Moreover, when the teacher is over-regularized, the one-shot estimate correctly selects negative $\widehat{\xi}^\star(\lambda)$, so that the SD predictor corrects excessive shrinkage. This regime would be missed by restricting $\xi$ to $[0, 1]$ (see Figure 16 for constrained SD risks as a comparison). Additional experiments and sample-size variations are provided in Section F.2, and for CIFAR10/CIFAR100 we also report the corresponding test accuracies in Figure 15.

## 5. Extensions and Variants

Our results so far focus on optimal *one-round* SD for *ordinary* ridge regression, where the student is refit on the *same* design matrix $X$ as the teacher. In this section, we briefly outline several extensions that are naturally captured by the same "structural" viewpoint from Section 2. Due to space constraints, full details and proofs are deferred to Section E.

### 5.1. Multiple Rounds of Self-Distillation

Define $y^{(0)} := y$ and fix $\lambda > 0$. A natural *recursive* multi-round distillation scheme can be described as follows. For

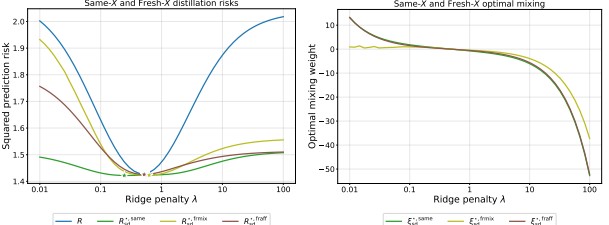

*Figure 8.* **Same-$X$ versus fresh-$X$ self-distillation risks.** Empirical test risks and optimal mixing weights (averaged over 30 simulations) in an isotropic setting with $n = 200$, $p = 100$, $r^2 = 1$, and $\sigma^2 = 1$. For each $\lambda$, we choose the fresh-$X$ mixing weight $\xi$ by grid search over 3,000 values in $[-100, 100]$, restricting to values for which the mixed-loss objective (17) is strictly convex, and picking the one with the lowest empirical test risk.

each $k \geq 0$, let $f^{(k)}_\lambda$ be the teacher ridge predictor trained on $(X, y^{(k)})$ and write $\widehat{y}^{(k)} := f^{(k)}_\lambda(X)$ for its fitted value. Given a mixing weight $\xi_{k+1} \in \mathbb{R}$, define the round-$(k + 1)$ SD predictor $f^{(k+1)}_{\mathsf{sd},\lambda,\xi_{k+1}}$ as the ridge fit obtained from the same mixed-loss construction as in (2), but with base labels $y^{(k)}$ and teacher pseudo-labels $\widehat{y}^{(k)}_\lambda$; see Figure 11 for an illustration. A simplification shows this is equivalent to fitting ridge regression on the mixed labels $y^{(k+1)}_{\lambda,\xi_{k+1}} := (1 - \xi_{k+1}) y^{(k)} + \xi_{k+1} \widehat{y}^{(k)}_\lambda$. Since $\xi_{k+1} = 0$ recovers $f^{(k)}_\lambda$, optimizing $\xi_{k+1}$ in each round yields a nonincreasing risk sequence (Proposition E.1); see Figure 7. Notice that this recursion scheme simply re-applies the structural identities from Section 2 with $y^{(k)}$ playing the role of the "base" labels. In contrast, several repeated-distillation formulations instead remain *anchored* to the original (ground-truth) labels (Garg et al., 2025; Alemohammad et al., 2023): $y^{(k+1}_{\lambda,\xi_{k+1}} := (1 - \xi_{k+1})y^{(0)} + \xi_{k+1}\widehat{y}^{(k)}$, for which the round-$k + 1$ family need not contain the round-$(k)$ predictor. As a result, the risk monotonicity in the number of rounds $k$ will in general fail; see Figure 7 for risk comparisons.

### 5.2. Self-Distillation with Fresh Unlabeled Features

Our structural results in Section 2 rely on the student refit using the *same* design matrix $X$ as the teacher. A common pseudo-labeling variant instead refits using additional *fresh* unlabeled covariates $\widetilde{X} \in \mathbb{R}^{n \times p}$ be independent of $\mathcal{D} = (X, y)$, and define the teacher pseudo-labels on $\widetilde{X}$ by $\widetilde{y}_\lambda := f_\lambda(\widetilde{X}) \in \mathbb{R}^m$. In direct analogy with (2), one may define the fresh-$X$ *mixed-loss* SD predictor $f^{\mathrm{frmix}}_{\mathsf{sd},\lambda,\xi}(x)$ as

$$x^\top \underset{\beta \in \mathbb{R}^p}{\operatorname{argmin}} \left\{ (1-\xi)\|y-X\beta\|_2^2/n + \xi\|\widetilde{y}_\lambda - \widetilde{X}\beta\|_2^2/m + \lambda\|\beta\|_2^2 \right\}. \tag{17}$$

Unlike the same-$X$ case, (17) does not reduce to ridge regression on a single mixed-label vector: because the two quadratic losses involve different Gram matrices, the map $\xi \mapsto f^{\mathrm{frmix}}_{\mathsf{sd},\lambda,\xi}$ is generally *not* an affine path, and the risk $R^{\mathrm{frmix}}_{\mathsf{sd}}(\lambda, \xi) := R(f^{\mathrm{frmix}}_{\mathsf{sd},\lambda,\xi})$ is no longer quadratic in $\xi$ (see

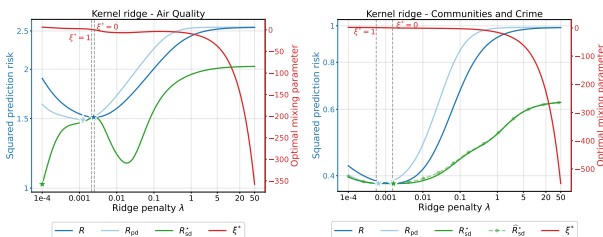

*Figure 9.* **Kernel ridge and kernel SD ridge regression.** Test risks of Gaussian kernel ridge and kernel SD ridge.

Lemma E.3).[2] Consequently, the two-predictor identities for $\xi^\star(\lambda)$ and $R_{\mathsf{sd}}^\star(\lambda)$ in Proposition 2.1 and the derivative identities in Theorem 2.2 do not apply directly in this fresh-$X$ mixed-loss setting.

To retain an affine structure, one may instead define a fresh-$X$ PD refit and then mix predictors explicitly. Let $f_{\mathsf{pd},\lambda}^{\mathsf{fr}}$ denote ridge regression trained on the pseudo-labeled dataset $(\widetilde{X}, \widetilde{y}_\lambda)$ with the same penalty $\lambda$. Define the fresh-$X$ *affine* SD family

$$f_{\mathsf{sd},\lambda,\xi}^{\mathrm{fraff}}(x) := (1-\xi)\,f_\lambda(x) + \xi\,f_{\mathsf{pd},\lambda}^{\mathsf{fr}}(x). \qquad (18)$$

Because (18) is an explicit two-predictor affine path, the corresponding risk $R_{\mathsf{sd}}^{\mathrm{fraff}}(\lambda,\xi) := R(f_{\mathsf{sd},\lambda,\xi}^{\mathrm{fraff}})$ is quadratic in $\xi$. Thus, the identities in Proposition 2.1 apply directly (with $f_{\mathsf{pd},\lambda}$ replaced by $f_{\mathsf{pd},\lambda}^{\mathsf{fr}}$) to characterize the optimal mixing $\xi_{\mathsf{sd}}^{\star,\mathrm{fraff}}(\lambda) \in \operatorname{argmin}_{\xi \in \mathbb{R}} R_{\mathsf{sd}}^{\mathrm{fraff}}(\lambda,\xi)$ and the optimal risk $R_{\mathsf{sd}}^{\star,\mathrm{fraff}}(\lambda) := \min_{\xi \in \mathbb{R}} R_{\mathsf{sd}}^{\mathrm{fraff}}(\lambda,\xi)$. However, the same-$X$ coupling behind the derivative-based characterization Theorem 2.2 is absent here, so pointwise strict improvements are no longer structurally guaranteed.

We provide a prototype theoretical comparison in an isotropic in-distribution setting. Specializing the asymptotic analysis of Section 3 for same-$X$ (Lemma E.4) and deriving a corresponding asymptotic analysis for affine fresh-$X$ (Lemma E.5) under isotropic covariance and isotropic signal, we show that when $p/n \to \gamma$ and $p/m \to \gamma$, the same-$X$ optimal SD risk uniformly dominates the fresh-affine optimal SD risk. The formal statement appears in Theorem E.2. A full analysis under general covariance and signal structure, and for the mixed-loss fresh-$X$ student (17), is left for future work.

### 5.3. Self-Distillation with Other Ridge Variants

The analysis of the results from Section 2 relies on two key facts: (i) the predictor is *linear* in the response vector $y$ (a linear smoother), and (ii) the ridge refit on pseudo-labels uses the *same* smoothing family. As a result, the affine SD path

---

[2]When $\xi \notin [0,1]$, the objective in (17) need not be convex because it mixes two different quadratic forms. Throughout, we interpret (17) only for values of $\xi$ such that the Hessian in $\beta$ is positive definite, i.e., $(1-\xi)\,X^\top X/n + \xi\,\widetilde{X}^\top \widetilde{X}/m + \lambda I_p \succ 0$, so the displayed $\operatorname{argmin}$ is well-defined.

(3) and the derivative identity behind Theorem 2.2 extend to several ridge-type predictors. We record two examples.

**Generalized ridge regression.** Fix a penalty matrix $\Omega \succeq 0$ (independent of $y$). For $\lambda \geq 0$, define the generalized ridge predictor $f_\lambda^\Omega(x) := x^\top (X^\top X + n\lambda\,\Omega)^{-1} X^\top y$ with $\widehat{y}_\lambda^\Omega := f_\lambda^\Omega(X)$. Define the corresponding pure-distilled predictor by refitting on pseudo-labels $\widehat{y}_\lambda^\Omega$ with the same penalty, and define $f_{\mathsf{sd},\lambda}^\Omega$ by the same mixed-label construction as in Section 2. Then, all the results from Section 2 extend to the generalized ridge under any squared risk (Section E.3). This includes feature-wise shrinkage ($\Omega$ diagonal), graph/Laplacian regularization, and spline-type penalties.

**Kernel ridge regression.** Let $k(\cdot,\cdot)$ be a PSD kernel with kernel matrix $K \in \mathbb{R}^{n \times n}$, i.e. $K_{ij} = k(x_i, x_j)$, and let $k_x := (k(x, x_1), \ldots, k(x, x_n))^\top$. For $\lambda > 0$, the kernel ridge predictor is $f_\lambda^{\mathrm{kern}}(x) := k_x^\top (K + n\lambda I_n)^{-1} y$ with $\widehat{y}_\lambda^{\mathrm{kern}}$. Define $f_{\mathsf{pd},\lambda}^{\mathrm{kern}}$ by refitting on $\widehat{y}_\lambda^{\mathrm{kern}}$, and $f_{\mathsf{sd},\lambda}^{\mathrm{kern}}$ by mixing as in (3). Similar results as in Section 2 also hold for kernel ridge regression; see Figure 9 and Section E.3.

## 6. Concluding Remarks

In this paper, we asked whether self-distillation can provably deliver strict gains ((Q1)), when it can rival an optimally tuned teacher ((Q2)), and how to tune it efficiently ((Q3)). For ridge regression, and more broadly, for a class of resolvent-based ridge smoothers, we provide affirmative and sharp answers. On the theory side, we derive nonasymptotic structural identities that hold conditionally on the observed training data and for *any* squared prediction risk, including out-of-distribution risks. Under a mild nondegeneracy condition, we show that the optimally mixed student *strictly* improves the teacher at every nonstationary $\lambda$ along the ridge path, with an optimal mixing weight of opposite sign to the ridge-risk derivative. We further derive deterministic equivalents for the optimal SD risk in the proportional regime $p/n \to \gamma \in (0,\infty)$ under anisotropic covariance and deterministic signals, quantifying how SD gains depend on $(\gamma, \mathrm{SNR})$ and signal–covariance alignment. On the algorithmic side, we propose a consistent one-shot GCV-based tuning method that avoids grid search, repeated retraining, and data splitting, while matching predicted risk curves well in real-data experiments.

Taken together, our results suggest a simple conceptual message: at least in ridge-type problems, optimal self-distillation acts as a cheap and tractable correction for a mis-regularized teacher using the teacher's own fitted values to move *along* the regularization path in a direction determined by the risk slope. An important direction is to understand how much of this regularization path geometry persists beyond ridge regression, including classification and more general learners, and to develop equally sharp guarantees for fresh-$X$ and other practically motivated distillation variants.

## Impact Statement

This paper presents work whose goal is to advance the field of Machine Learning. There are many potential societal consequences of our work, none which we feel must be specifically highlighted here.

## Software and Data

The source code for generating all of our figures is included in the supplementary material. (The file named `README.md` lists the organizational structure.) It is also available at an anonymous repository (which will be made public after paper acceptance) at: `https://github.com/hhd357/optimal_self_distillation_ridge`

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

# Supplement

This supplement serves as a companion to the paper titled "Optimal Unconstrained Self-Distillation in Ridge Regression: Strict Improvements, Precise Asymptotics, and One-Shot Tuning". We provide an outline of the supplement in Table 3.

## Organization

*Table 3.* Outline of the supplement.

| Section | Subsection | Purpose |
|---|---|---|
| Further details in Section 1 | | |
| Section A | | Further related works |
| Proofs in Section 2 | | |
| Section B | Section B.1 | Preliminaries |
| | Section B.2 | Proof of Equation 3 |
| | Section B.3 | Proof of Proposition 2.1 |
| | Section B.4 | Proof of Theorem 2.2 |
| | Section B.5 | Proof of Proposition 2.3 |
| Proofs in Section 3 | | |
| Section C | Section C.1 | Preliminaries |
| | Section C.2 | Helper results (concentration components) for the proof of Theorem 3.2 |
| | Section C.3 | Helper results (deterministic equivalents) for the proof of Theorem 3.2 |
| | Section C.4 | Proof of Theorem 3.2 |
| | Section C.5 | Proofs of helper results (concentration components) for the proof of Theorem 3.2 |
| | Section C.6 | Proofs of helper results (deterministic equivalents) for the proof of Theorem 3.2 |
| | Section C.7 | Proof of Corollary 3.3 |
| | Section C.8 | Proof of Proposition 3.4 |
| Proofs in Section 4 | | |
| Section D | Section D.1 | Preliminaries |
| | Section D.2 | Helper results (risk and correlation components consistency) for the proof of Theorem 4.1 |
| | Section D.3 | Proof of Theorem 4.1 |
| | Section D.4 | Technical lemmas |
| Further details in Section 5 | | |
| Section E | Section E.1 | Results and proofs in Section 5.1 |
| | Section E.2 | Results and proofs in Section 5.2 |
| | Section E.3 | Results and proofs in Section 5.3 |
| Additional experiments | | |
| Section F | Section F.1 | Additional experiments on CIFAR datasets |
| | Section F.2 | Additional illustrations in Section 2 |
| | Section F.3 | Additional illustrations in Section 3.2 |
| | Section F.4 | Additional illustrations in Section 3.3 |
| Experimental details | | |
| Section G | Section G.1 | Real-world regression tasks |
| | Section G.2 | ResNet features extraction from CIFAR datasets |
| | Section G.3 | Synthetic asymptotic experiments |

*Table 4.* Summary of general notation used throughout the paper.

| Notation | Description |
|---|---|
| **Typography** | |
| Lowercase (e.g., $x$) | Scalars or vectors |
| Uppercase (e.g., $X$) | Matrices or linear operators |
| Calligraphic (e.g., $\mathcal{D}, \mathcal{R}$) | Sets, $\sigma$-fields, events or certain limiting functions |
| Blackboard bold (e.g., $\mathbb{R}, \mathbb{N}$) | Standard number systems |
| **Analysis** | |
| $\mathbb{Z}, \mathbb{N}, \mathbb{R}, \mathbb{R}_{\geq 0}, \overline{\mathbb{R}}$ | Integers, positive integers, reals, nonnegative reals, extended reals |
| $(a, b, c), \{a, b, c\}$ | Ordered tuple and (unordered) set |
| $[n]$ | Set $\{1, \ldots, n\}$ for a positive integer $n$ |
| $x \wedge y, x \vee y$ | $\min\{x, y\}$ and $\max\{x, y\}$ for real numbers $x, y$ |
| $\mathbb{1}_A$ | Indicator random variable associated with event or set $A$ |
| $\text{sign}(x)$ | Sign of a real number $x$ |
| $C^\infty$ | Function class of infinitely differentiable functions |
| **Linear algebra** | |
| $\text{tr}(A), \overline{\text{tr}}(A), \det(A)$ | Trace, normalized trace ($\text{tr}(A)/p$), and determinant of a square matrix $A \in \mathbb{R}^{p \times p}$ |
| $B^{-1}$ | Inverse of an invertible square matrix $B$ |
| $C^\dagger$ | Moore-Penrose inverse of a general rectangular matrix $C$ |
| $\text{diag}(d_1, \ldots, d_p)$ | Diagonal matrix with diagonal entries $d_1, \ldots, d_p$ |
| $U^{1/2}$ | Principal square root of a positive semidefinite matrix $U \succeq 0$ |
| $f(W)$ | Functional calculus for positive semidefinite matrix $W$ (apply $f$ to eigenvalues of $W$) |
| $I, 1, 0$ | The identity matrix, the all-ones vector, the all-zeros vector |
| **Inner products and norms** | |
| $\langle u, v \rangle$ | Euclidean inner product (or another inner product when specified) |
| $\|x\|_2$ (or simply $\|x\|$) | Euclidean norm of a vector $x$ |
| $\|x\|_q$ | $\ell_q$ norm of a vector ($q \geq 1$) |
| $\|x\|_A := \sqrt{x^\top A x}$ | $A$-seminorm for $A \succeq 0$ |
| $\|A\|_{\text{op}}$ | Operator/spectral norm of a matrix $A$ |
| $\|A\|_F$ | Frobenius norm of a matrix $A$ |
| $\|A\|_{\text{tr}}$ | Trace/nuclear norm (sum of singular values) |
| $\|f\|_{L_q}$ | $L_q$ norm of a function $f$ under the relevant measure ($q \geq 1$) |
| **Probability** | |
| $\mathbb{P}(\cdot), \mathbb{E}[\cdot]$ | Probability and expectation |
| $\mathbb{E}[\cdot \mid \mathcal{G}]$ | Conditional expectation given $\sigma$-field $\mathcal{G}$ (or given data, depending on context) |
| $\text{Var}(\cdot), \text{Cov}(\cdot, \cdot)$ | Variance and covariance |
| $X \sim \mathcal{N}(\mu, \Sigma)$ | Gaussian random vector with mean $\mu$ and covariance $\Sigma$ |
| $\overset{d}{=}$ | Equality in distribution |
| **Orders and asymptotics** | |
| $X = \mathcal{O}_\alpha(Y), X \lesssim_\alpha Y$ | Deterministic upper bounds with constant possibly depending on parameter $\alpha$ |
| $o(\cdot), \mathcal{O}(\cdot)$ | Deterministic little-$o$ and big-$O$ |
| $o_{\mathbb{P}}(\cdot), \mathcal{O}_{\mathbb{P}}(\cdot)$ | Probabilistic little-$o$ and big-$O$ |
| $\to, \overset{p}{\to}, \overset{a.s.}{\longrightarrow}, \overset{d}{\to}$ | Convergence, in probability, almost surely, in distribution |
| $\asymp$ | Asymptotic equivalence (see Definition C.6 for more details) |
| $C, C', c, c'$ | Generic positive constants (may change from line to line) |

# A. Further Related Works

**High-dimension risk characterization.** A large body of recent work has focussed ridge regression risk and its variants under proportional asymptotics using tools from random matrix theory and statistical physics (e.g., Dobriban & Wager, 2018; Hastie et al., 2022; Patil et al., 2024 and references therein), shedding light on phenomena such as benign overfitting (Bartlett et al., 2020) and double descent (Belkin et al., 2019). We extend this literature by deriving exact deterministic equivalents for optimal SD risk under anisotropic feature covariance and deterministic signals. Because the teacher and student are trained on the *same* data, SD risks involve dependent resolvent-type quantities; unlike settings with independent refits where one can directly combine known deterministic equivalents, our analysis requires higher-order deterministic equivalents obtained via block linearization (see Sections C.3 and C.6).

**Synthetic-label retraining and model collapse.** Beyond one-round SD, recent work has raised concerns about recursively training on model-generated labels leading to performance degradation, referred to as *model collapse* (Shumailov et al., 2024; Alemohammad et al., 2023; Dohmatob et al., 2024; Gerstgrasser et al., 2024). In contrast to the notion of "strong model collapse" in Dohmatob et al. (2025) (in which a model trained in the presence of synthetic data has worse asymptotic risk than a model trained solely on ground-truth data), optimal SD in our ridge setting exhibits no such degradation. Provided the nondegeneracy condition in Theorem 2.2 holds, the optimally mixed student strictly improves upon the teacher whenever $R'(\lambda) \neq 0$, with the two risks coinciding only at stationary points.

Closer to our setting, (He et al., 2025; Garg et al., 2025) analyze infinite-round schemes that mix ground-truth and synthetic labels with a fixed weight $w$ (analogous to our $\xi$) to prevent degradation under repeated synthetic training; the optimal mixing weight $w^\star$ in their settings lies in $[0, 1]$. In contrast, we characterize the risk-minimizing mixing for one-round SD and show that $\xi^\star$ can lie outside $[0, 1]$, including $\xi^\star < 0$ in over-regularized regimes. To highlight the distinction, we demonstrate a synthetic experiment below comparing optimal one-round SD risk to 20-round SD with optimal fixed constrained weight in $[0, 1]$ for every round. As shown in Figure 10, unconstrained mixing can be crucial for improving the teacher's performance, particularly in over-regularized regimes where negative weights becomes necessary. Finally, we also study a recursive multi-round variant with per-round unconstrained optimal mixing that yields a monotone (weakly) decreasing risk sequence (Section 5).

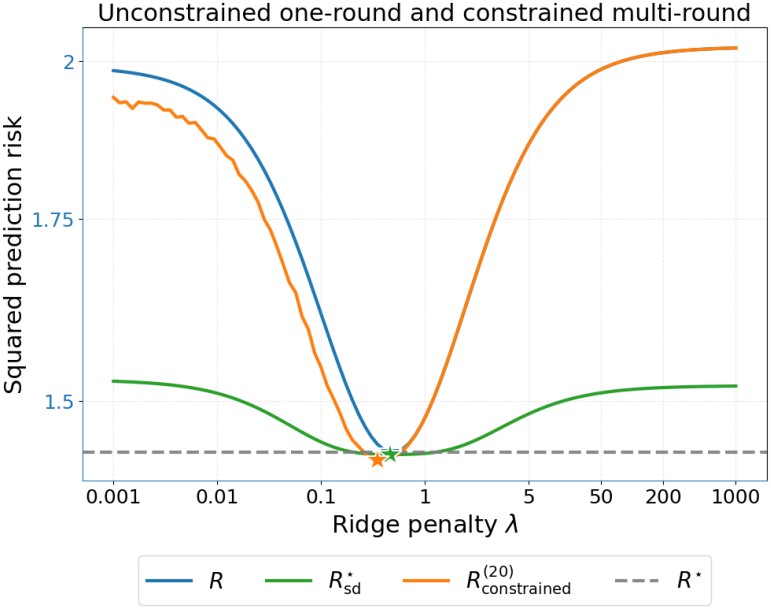

*Figure 10.* **One-round unconstrained versus multi-round constrained mixing weight.** Squared prediction risk empirical curves averaged over 20 simulations, $n = 400, p = 200$ with isotropic design and isotropic signal. We choose the optimal $\xi$ over a grid of 200 values in $[0, 1]$ and pick the one with the lowest risk at the 20-th round.

# B. Proofs in Section 2

Recall that we condition throughout on the observed training data $\mathcal{D} = (X, y)$, where $X \in \mathbb{R}^{n \times p}$ and $y \in \mathbb{R}^n$. Let $(x_0, y_0)$ denote a test pair (possibly out-of-distribution) satisfying $\mathbb{E}[y_0^2 \mid \mathcal{D}] < \infty$ and $\mathbb{E}[\|x_0\|_2^2 \mid \mathcal{D}] < \infty$. For any predictor $f : \mathbb{R}^p \to \mathbb{R}$, recall the conditional squared prediction risk is defined as:

$$R(f) := \mathbb{E}\big[(y_0 - f(x_0))^2 \mid \mathcal{D}\big].$$

All derivatives in $\lambda$ below are taken for $\lambda > 0$.

## B.1. Preliminaries

Define the empirical covariance matrix and empirical resolvent as

$$\widehat{\Sigma} := \frac{1}{n} X^\top X \in \mathbb{R}^{p \times p}, \quad \text{and} \quad Q_\lambda := (\widehat{\Sigma} + \lambda I_p)^{-1} \in \mathbb{R}^{p \times p}, \tag{19}$$

respectively. For any label vector $u \in \mathbb{R}^n$, define the ridge coefficient map and predictor as

$$\beta_\lambda(u) := Q_\lambda \frac{X^\top u}{n}, \quad \text{and} \quad f_{\lambda,u}(x) := x^\top \beta_\lambda(u), \tag{20}$$

respectively. In particular, the teacher predictor is $f_\lambda := f_{\lambda,y}$ and its fitted values are $\widehat{y}_\lambda := f_\lambda(X) = X\beta_\lambda(y)$. Note that the ridge predictor $f_{\lambda,u}(x) = x^\top \beta_\lambda(u)$ is linear in $u$.

The PD predictor $f_{\mathsf{pd},\lambda}$ is the ridge predictor trained on $(X, \widehat{y}_\lambda)$ with the same penalty $\lambda$, i.e., $f_{\mathsf{pd},\lambda} := f_{\lambda,\widehat{y}_\lambda}$. Its coefficient vector admits the closed form

$$\beta_{\mathsf{pd},\lambda} := \beta_\lambda(\widehat{y}_\lambda) = Q_\lambda \frac{X^\top (X\beta_\lambda(y))}{n} = Q_\lambda \widehat{\Sigma} \beta_\lambda(y). \tag{21}$$

Since $\widehat{\Sigma}$ and $Q_\lambda$ commute (both are polynomials in $\widehat{\Sigma}$), we have $\widehat{\Sigma}Q_\lambda = I_p - \lambda Q_\lambda$, and therefore

$$\beta_\lambda(y) - \beta_{\mathsf{pd},\lambda} = \big(I_p - Q_\lambda\widehat{\Sigma}\big)\beta_\lambda(y) = \lambda Q_\lambda \beta_\lambda(y). \tag{22}$$

## B.2. Proof of Equation 3

**Lemma B.1** (SD ridge representation). *Fix $\lambda > 0$ and $\xi \in \mathbb{R}$. Let $\widehat{y}_\lambda = f_\lambda(X)$ and define the mixed label vector*

$$y^{(1)}(\xi) := (1 - \xi)y + \xi\widehat{y}_\lambda \in \mathbb{R}^n. \tag{23}$$

*Then the SD predictor* (2) *coincides with ridge regression trained on $(X, y^{(1)}(\xi))$ with penalty $\lambda$, i.e., $f_{\mathsf{sd},\lambda,\xi} = f_{\lambda,\,y^{(1)}(\xi)}$.*

*Proof.* Expanding the SD objective (and dropping terms independent of $\beta$), we obtain that

$$(1 - \xi)\frac{1}{n}\|y - X\beta\|_2^2 + \xi\frac{1}{n}\|\widehat{y}_\lambda - X\beta\|_2^2 + \lambda\|\beta\|_2^2$$
$$= \frac{1}{n}\|X\beta\|_2^2 - \frac{2}{n}\beta^\top X^\top \big((1 - \xi)y + \xi\widehat{y}_\lambda\big) + \lambda\|\beta\|_2^2 + \text{const},$$

where const is a term that does not depend on $\beta$. This is exactly the ridge objective with response $y^{(1)}(\xi)$ as defined in (23). $\square$

*Proof of Equation* (3). By Lemma B.1, $f_{\mathsf{sd},\lambda,\xi} = f_{\lambda,\,(1-\xi)y+\xi\widehat{y}_\lambda}$. Using linearity of $f_{\lambda,u}$ in $u$ (cf. (20)),

$$f_{\mathsf{sd},\lambda,\xi} = (1 - \xi)f_{\lambda,y} + \xi f_{\lambda,\widehat{y}_\lambda} = (1 - \xi)f_\lambda + \xi f_{\mathsf{pd},\lambda},$$

which is (3). $\square$

## B.3. Proof of Proposition 2.1

Throughout, $\lambda > 0$ is fixed and we abbreviate $f := f_\lambda$ and $g := f_{\mathsf{pd},\lambda}$. Write

$$R := R(f), \qquad R_{\mathsf{pd}} := R(g), \qquad C := \mathbb{E}[(y_0 - f(x_0))(y_0 - g(x_0)) \mid \mathcal{D}],$$

and define the nonnegative quantity y

$$D := R + R_{\mathsf{pd}} - 2C = \mathbb{E}[(f(x_0) - g(x_0))^2 \mid \mathcal{D}] \geq 0.$$

*Proof of Proposition 2.1.* By (3), the SD predictor with mixing weight $\xi$ is $f_{\mathsf{sd}}(\xi) = (1 - \xi)f + \xi g$. Let the residuals be $r_f := y_0 - f(x_0)$ and $r_g := y_0 - g(x_0)$. Then

$$y_0 - f_{\mathsf{sd}}(\xi)(x_0) = (1 - \xi)r_f + \xi r_g,$$

and thus

$$
\begin{aligned}
R_{\mathsf{sd}}(\lambda, \xi) &= \mathbb{E}[((1 - \xi)r_f + \xi r_g)^2 \mid \mathcal{D}] \\
&= (1 - \xi)^2 \mathbb{E}[r_f^2 \mid \mathcal{D}] + \xi^2 \mathbb{E}[r_g^2 \mid \mathcal{D}] + 2\xi(1 - \xi)\mathbb{E}[r_f r_g \mid \mathcal{D}] \\
&= (1 - \xi)^2 R + \xi^2 R_{\mathsf{pd}} + 2\xi(1 - \xi)C \\
&= R - 2\xi(R - C) + \xi^2 D.
\end{aligned}
$$

If $D > 0$, this is a strictly convex quadratic in $\xi$ with a unique minimizer given by

$$0 = \partial_\xi R_{\mathsf{sd}}(\lambda, \xi) = -2(R - C) + 2\xi D \quad \Longrightarrow \quad \xi^\star = \frac{R - C}{D}.$$

Substituting back yields

$$R_{\mathsf{sd}}^\star = R - \frac{(R - C)^2}{D}.$$

If $D = 0$, then $f(x_0) = g(x_0)$ a.s. under the test distribution, hence $R_{\mathsf{sd}}(\lambda, \xi) \equiv R$ for all $\xi$ and every $\xi$ is optimal with $R_{\mathsf{sd}}^\star = R$. $\qquad\square$

## B.4. Proof of Theorem 2.2

We first establish a derivative identity relating the PD predictor to the derivative of the teacher along the ridge path.

**Lemma B.2** (Resolvent derivative rule). *For a fixed $\lambda > 0$ and symmetric $A \succeq 0$, let $Q_\lambda := (A + \lambda I)^{-1}$. Then,*

$$\partial_\lambda Q_\lambda = -Q_\lambda^2.$$

*Proof.* Differentiate $(A + \lambda I)Q_\lambda = I$ and left-multiply by $Q_\lambda$. $\qquad\square$

**Lemma B.3** (Derivative identity for ridge versus pure-distillation). *For any $\lambda > 0$ and $x \in \mathbb{R}^p$,*

$$f_\lambda(x) - f_{\mathsf{pd},\lambda}(x) = -\lambda \, \partial_\lambda f_\lambda(x). \tag{24}$$

*Proof.* From the definitions, $\beta_\lambda(y) = Q_\lambda X^\top y/n$ and $\beta_{\mathsf{pd},\lambda} = Q_\lambda \widehat{\Sigma} \beta_\lambda(y)$; see (20) and (21). By Lemma B.2,

$$\partial_\lambda \beta_\lambda(y) = (\partial_\lambda Q_\lambda)\frac{X^\top y}{n} = -Q_\lambda^2 \frac{X^\top y}{n} = -Q_\lambda \beta_\lambda(y).$$

Thus $-\lambda \, \partial_\lambda \beta_\lambda(y) = \lambda Q_\lambda \beta_\lambda(y)$. Using (22), $\lambda Q_\lambda \beta_\lambda(y) = \beta_\lambda(y) - \beta_{\mathsf{pd},\lambda}$. Multiplying by $x^\top$ yields (24). $\qquad\square$

Next we relate $R(\lambda) - C(\lambda)$ to $R'(\lambda)$. Before we proceed, we justify that $R'(\lambda)$ is indeed well-defined under our setup.

**Lemma B.4** (Smoothness of ridge predictions and risk). *Fix the training data $\mathcal{D} = (X, y)$ and assume $\mathbb{E}[y_0^2 \mid \mathcal{D}] < \infty$ and $\mathbb{E}[\|x_0\|_2^2 \mid \mathcal{D}] < \infty$. Then for every $\lambda > 0$, the map $\lambda \mapsto f_\lambda(x_0)$ is $C^\infty$ and the risk $R(\lambda) = \mathbb{E}[(y_0 - f_\lambda(x_0))^2 \mid \mathcal{D}]$ is $C^\infty$ on $(0, \infty)$. Moreover,*

$$R'(\lambda) = -2\,\mathbb{E}\big[(y_0 - f_\lambda(x_0))\, \partial_\lambda f_\lambda(x_0) \mid \mathcal{D}\big], \tag{25}$$

*where $\partial_\lambda f_\lambda(x) = -x^\top Q_\lambda \beta_\lambda(y)$ with $Q_\lambda = (\widehat{\Sigma} + \lambda I)^{-1}$.*

*Proof.* Fix $\lambda_0 > 0$ and consider $\lambda \in [\lambda_0/2, 3\lambda_0/2]$. Then $\|Q_\lambda\|_{\text{op}} \leq 2/\lambda_0$, and with $b := X^\top y/n$ we have $\beta_\lambda = Q_\lambda b$ and $\partial_\lambda \beta_\lambda = -Q_\lambda \beta_\lambda$. Hence there exist constants (depending on $\mathcal{D}$ and $\lambda_0$) such that uniformly over this interval,

$$|f_\lambda(x_0)| \leq c_1 \|x_0\|_2, \qquad |\partial_\lambda f_\lambda(x_0)| \leq c_2 \|x_0\|_2.$$

Therefore,

$$\big|\partial_\lambda(y_0 - f_\lambda(x_0))^2\big| = 2|y_0 - f_\lambda(x_0)|\,|\partial_\lambda f_\lambda(x_0)| \leq c_3\big(|y_0|\|x_0\|_2 + \|x_0\|_2^2\big),$$

whose conditional expectation is finite by Cauchy–Schwarz and the assumed second-moment bounds. Dominated convergence justifies differentiating under $\mathbb{E}[\cdot \mid \mathcal{D}]$, yielding (25). Higher derivatives follow similarly since $f_\lambda(x_0)$ is a rational (hence analytic) function of $\lambda$ for $\lambda > 0$. $\square$

**Lemma B.5** (Derivative identity for $R(\lambda) - C(\lambda)$). *For any $\lambda > 0$,*

$$R(\lambda) - C(\lambda) = -\frac{\lambda}{2}\,R'(\lambda), \tag{26}$$

*where $R(\lambda) = R(f_\lambda)$ and $C(\lambda) = \mathbb{E}[(y_0 - f_\lambda(x_0))(y_0 - f_{\text{pd},\lambda}(x_0)) \mid \mathcal{D}]$.*

*Proof.* Write $r_\lambda := y_0 - f_\lambda(x_0)$. First note that

$$R(\lambda) - C(\lambda) = \mathbb{E}[r_\lambda^2 - r_\lambda(y_0 - f_{\text{pd},\lambda}(x_0)) \mid \mathcal{D}] = \mathbb{E}[r_\lambda\,(f_{\text{pd},\lambda}(x_0) - f_\lambda(x_0)) \mid \mathcal{D}].$$

Using Lemma B.3, $f_{\text{pd},\lambda}(x_0) - f_\lambda(x_0) = \lambda\,\partial_\lambda f_\lambda(x_0)$, hence

$$R(\lambda) - C(\lambda) = \lambda\,\mathbb{E}\big[r_\lambda\,\partial_\lambda f_\lambda(x_0) \mid \mathcal{D}\big]. \tag{27}$$

On the other hand, from Lemma B.4, we have

$$R'(\lambda) = -2\,\mathbb{E}[r_\lambda\,\partial_\lambda f_\lambda(x_0) \mid \mathcal{D}].$$

Combining with (27) yields (26). $\square$

*Proof of Theorem 2.2.* Let $D(\lambda) := R(\lambda) + R_{\text{pd}}(\lambda) - 2C(\lambda)$. When $D(\lambda) > 0$, Proposition 2.1 gives

$$\xi^\star(\lambda) = \frac{R(\lambda) - C(\lambda)}{D(\lambda)}, \qquad R_{\text{sd}}^\star(\lambda) = R(\lambda) - \frac{(R(\lambda) - C(\lambda))^2}{D(\lambda)}.$$

Apply Lemma B.5 to substitute $R(\lambda) - C(\lambda) = -(\lambda/2)R'(\lambda)$, obtaining

$$\xi^\star(\lambda) = -\frac{\lambda}{2}\frac{R'(\lambda)}{D(\lambda)}, \qquad R_{\text{sd}}^\star(\lambda) = R(\lambda) - \frac{\lambda^2}{4}\frac{(R'(\lambda))^2}{D(\lambda)}.$$

If $R'(\lambda) \neq 0$, then the subtracted term is strictly positive (since $D(\lambda) > 0$), hence $R_{\text{sd}}^\star(\lambda) < R(\lambda)$. Finally, $\text{sign}(\xi^\star(\lambda)) = -\text{sign}(R'(\lambda))$ because $\lambda > 0$ and $D(\lambda) > 0$. $\square$

## B.5. Proof of Proposition 2.3

*Proof of Proposition 2.3.* Define

$$g(\lambda) := R(\lambda) - C(\lambda), \quad \text{and} \quad D(\lambda) := R(\lambda) + R_{\mathsf{pd}}(\lambda) - 2C(\lambda).$$

For $D(\lambda) > 0$, Proposition 2.1 gives

$$R_{\mathsf{sd}}^{\star}(\lambda) = R(\lambda) - \frac{g(\lambda)^2}{D(\lambda)}. \tag{28}$$

By Lemma B.5, $g(\lambda) = -(\lambda/2)R'(\lambda)$, hence at $\lambda^{\star}$,

$$g(\lambda^{\star}) = 0, \qquad g'(\lambda^{\star}) = -\frac{\lambda^{\star}}{2}R''(\lambda^{\star}). \tag{29}$$

Differentiate (28) twice and evaluate at $\lambda^{\star}$. All terms involving $g(\lambda^{\star})$ vanish, yielding

$$R_{\mathsf{sd}}^{\star\prime\prime}(\lambda^{\star}) = R''(\lambda^{\star}) - \frac{2(g'(\lambda^{\star}))^2}{D(\lambda^{\star})}. \tag{30}$$

Substituting (29) into (30) gives

$$R_{\mathsf{sd}}^{\star\prime\prime}(\lambda^{\star}) = R''(\lambda^{\star}) - \frac{\lambda^{\star\,2}}{2}\frac{(R''(\lambda^{\star}))^2}{D(\lambda^{\star})}.$$

Therefore, if $D(\lambda^{\star}) < \frac{\lambda^{\star\,2}}{2}R''(\lambda^{\star})$, then $R_{\mathsf{sd}}^{\star\prime\prime}(\lambda^{\star}) < 0$, so $\lambda^{\star}$ cannot be a local minimizer of $R_{\mathsf{sd}}^{\star}$. Since $g(\lambda^{\star}) = 0$, (28) gives $R_{\mathsf{sd}}^{\star}(\lambda^{\star}) = R(\lambda^{\star}) = \min_{\lambda\geq 0} R(\lambda)$. Hence there exists $\lambda > 0$ such that $R_{\mathsf{sd}}^{\star}(\lambda) < R_{\mathsf{sd}}^{\star}(\lambda^{\star}) = \min_{\lambda\geq 0} R(\lambda)$, which implies $\min_{\lambda>0} R_{\mathsf{sd}}^{\star}(\lambda) < \min_{\lambda>0} R(\lambda)$. $\qquad\square$

# C. Proofs in Section 3

## C.1. Preliminaries

Recall the random-design model in Assumption 3.1. The design matrix $X \in \mathbb{R}^{n\times p}$ has rows $x_i^{\top} = z_i^{\top}\Sigma^{1/2}$, where $z_i \in \mathbb{R}^p$ has i.i.d. entries with mean 0, variance 1, and uniformly bounded $(4 + \mu)$-th moment for some $\mu > 0$, and $\Sigma \in \mathbb{R}^{p\times p}$ is deterministic positive definite with spectrum uniformly bounded away from 0 and $\infty$. The response $y$ has mean 0 and uniformly bounded $(4 + \nu)$-th moment.

Define the population $L_2$-projection coefficient, residual, and residual variance by

$$\beta := \mathbb{E}[xx^{\top}]^{-1}\mathbb{E}[xy] = \Sigma^{-1}\mathbb{E}[xy], \qquad \varepsilon := y - x^{\top}\beta, \qquad \sigma^2 := \mathbb{E}[\varepsilon^2].$$

Then $\mathbb{E}[x\varepsilon] = 0$ (equivalently $\mathbb{E}[z\,\varepsilon] = 0$) and $\mathbb{E}[\varepsilon] = 0$. Throughout, we consider the in-distribution prediction setting, i.e., the test pair $(x_0, y_0)$ is an independent copy of $(x, y)$, so $y_0 = x_0^{\top}\beta + \varepsilon_0$ with $\mathbb{E}[x_0\varepsilon_0] = 0$ and $\mathbb{E}[\varepsilon_0^2] = \sigma^2$.

Write the sample covariance and ridge resolvent as

$$\widehat{\Sigma} := \frac{1}{n}X^{\top}X, \qquad Q_{\lambda} := (\widehat{\Sigma} + \lambda I_p)^{-1}, \qquad \lambda > 0,$$

and set $\overline{\mathrm{tr}}(A) := \mathrm{tr}(A)/p$.

Define the ridge estimator and its pure-distilled transform by

$$\beta_{\lambda} := Q_{\lambda}\frac{X^{\top}y}{n}, \qquad M_{\lambda} := \widehat{\Sigma}(\widehat{\Sigma} + \lambda I_p)^{-1} = \widehat{\Sigma}Q_{\lambda} = I_p - \lambda Q_{\lambda}, \qquad \beta_{\mathsf{pd},\lambda} := M_{\lambda}\beta_{\lambda}.$$

We use the $\Sigma$-inner product and norm

$$\langle u, v\rangle_{\Sigma} := u^{\top}\Sigma v, \qquad \|u\|_{\Sigma}^2 := \langle u, u\rangle_{\Sigma}.$$

For any estimator $\widehat{\beta} = \widehat{\beta}(X, y)$, the in-distribution prediction MSE decomposes as

$$\mathbb{E}[(y_0 - x_0^\top \widehat{\beta})^2 \mid X, y] = \|\widehat{\beta} - \beta\|_\Sigma^2 + \sigma^2,$$

and similarly, for two estimators $\widehat{\beta}, \widetilde{\beta}$,

$$\mathbb{E}[(y_0 - x_0^\top \widehat{\beta})(y_0 - x_0^\top \widetilde{\beta}) \mid X, y] = \langle \widehat{\beta} - \beta, \widetilde{\beta} - \beta \rangle_\Sigma + \sigma^2.$$

Thus, adding $\sigma^2$ to the teacher/PD risks and correlation term does not affect the optimal mixing weight (constants cancel), and it only adds $\sigma^2$ to the optimal mixed prediction MSE.

Accordingly, for a fixed training dataset $(X, y)$, we define the *excess* scalars

$$\overline{R}(\lambda) := \|\beta_\lambda - \beta\|_\Sigma^2, \qquad \overline{R}_{\mathsf{pd}}(\lambda) := \|\beta_{\mathsf{pd},\lambda} - \beta\|_\Sigma^2, \qquad \overline{C}(\lambda) := \langle \beta_\lambda - \beta, \beta_{\mathsf{pd},\lambda} - \beta \rangle_\Sigma,$$

and note that the full in-distribution quantities in the main paper are $\overline{R}(\lambda) + \sigma^2$, $\overline{R}_{\mathsf{pd}}(\lambda) + \sigma^2$, and $\overline{C}(\lambda) + \sigma^2$.

Recall from Section 2 that the self-distilled estimator is

$$\beta_{\mathsf{sd},\lambda}(\xi) := (1 - \xi)\beta_\lambda + \xi \beta_{\mathsf{pd},\lambda},$$

and its excess risk is

$$\|\beta_{\mathsf{sd},\lambda}(\xi) - \beta\|_\Sigma^2 = (1 - \xi)^2 \overline{R}(\lambda) + \xi^2 \overline{R}_{\mathsf{pd}}(\lambda) + 2\xi(1 - \xi)\overline{C}(\lambda).$$

Therefore the optimal mixing weight and optimal excess SD risk are given by

$$\xi^\star(\lambda) = \frac{\overline{R}(\lambda) - \overline{C}(\lambda)}{\overline{R}(\lambda) + \overline{R}_{\mathsf{pd}}(\lambda) - 2\overline{C}(\lambda)}, \qquad R_{\mathsf{sd}}^\star(\lambda) = \sigma^2 + \overline{R}(\lambda) - \frac{(\overline{R}(\lambda) - \overline{C}(\lambda))^2}{\overline{R}(\lambda) + \overline{R}_{\mathsf{pd}}(\lambda) - 2\overline{C}(\lambda)}, \tag{31}$$

whenever the denominator is positive. Thus, to prove Theorem 3.2, it suffices to show that $\overline{R}(\lambda), \overline{C}(\lambda), \overline{R}_{\mathsf{pd}}(\lambda)$ converge in probability to deterministic limits $\mathscr{R}(\lambda) - \sigma^2, \mathscr{C}(\lambda) - \sigma^2, \mathscr{R}_{\mathsf{pd}}(\lambda) - \sigma^2$ (i.e., the excess parts of the main-paper limits), and then apply the continuous mapping theorem to (31) (and finally add $\sigma^2$ back to match the full risks stated in Theorem 3.2).

## C.2. Helpers Results (Concentration Components) for the Proof of Theorem 3.2

Define the residual vector $\varepsilon := (\varepsilon_1, \ldots, \varepsilon_n)^\top$ with $\varepsilon_i := y_i - x_i^\top \beta$, so that $y = X\beta + \varepsilon$ holds identically.

**Lemma C.1** (Exact decomposition of coefficient errors). *For any $\lambda > 0$,*

$$\beta_\lambda - \beta = -\lambda Q_\lambda \beta + Q_\lambda X^\top \varepsilon / n,$$
$$\beta_{\mathsf{pd},\lambda} - \beta = (-2\lambda Q_\lambda + \lambda^2 Q_\lambda^2)\beta + (I_p - \lambda Q_\lambda)Q_\lambda X^\top \varepsilon / n.$$

*Proof.* Using $y = X\beta + \varepsilon$ and $\widehat{\Sigma} = X^\top X / n$,

$$\beta_\lambda = Q_\lambda(\widehat{\Sigma}\beta + X^\top \varepsilon / n) = (I_p - \lambda Q_\lambda)\beta + Q_\lambda X^\top \varepsilon / n,$$

so $\beta_\lambda - \beta = -\lambda Q_\lambda \beta + Q_\lambda X^\top \varepsilon / n$. Then $\beta_{\mathsf{pd},\lambda} = (I_p - \lambda Q_\lambda)\beta_\lambda$ and expanding yields the second identity. $\square$

Define

$$Q_2(\lambda) := \lambda^2 Q_\lambda \Sigma Q_\lambda, \qquad Q_3(\lambda) := \lambda^3(Q_\lambda \Sigma Q_\lambda^2 + Q_\lambda^2 \Sigma Q_\lambda), \qquad Q_4(\lambda) := \lambda^4 Q_\lambda^2 \Sigma Q_\lambda^2, \tag{32}$$

and

$$U_k(\lambda) := \frac{1}{n} \operatorname{tr}(\Sigma \widehat{\Sigma} Q_\lambda^k), \qquad k \in \{2, 3, 4\}. \tag{33}$$

**Lemma C.2** (Expansion of $\overline{R}(\lambda), \overline{C}(\lambda), R_{\mathsf{pd}}(\lambda)$). *Let $\lambda > 0$ be fixed. We have*

$$\overline{R}(\lambda) = \beta^\top Q_2(\lambda)\beta + \frac{1}{n^2}\varepsilon^\top X Q_\lambda \Sigma Q_\lambda X^\top \varepsilon - \frac{2\lambda}{n}\beta^\top Q_\lambda \Sigma Q_\lambda X^\top \varepsilon, \tag{34}$$

$$\overline{C}(\lambda) = \beta^\top \left(2Q_2(\lambda) - \tfrac{1}{2}Q_3(\lambda)\right)\beta + \frac{1}{n^2}\varepsilon^\top X Q_\lambda \Sigma(I_p - \lambda Q_\lambda)Q_\lambda X^\top \varepsilon + \mathrm{Lin}_C(\lambda), \tag{35}$$

$$\overline{R}_{\mathsf{pd}}(\lambda) = \beta^\top \left(4Q_2(\lambda) - 2Q_3(\lambda) + Q_4(\lambda)\right)\beta + \frac{1}{n^2}\varepsilon^\top X Q_\lambda (I_p - \lambda Q_\lambda)\Sigma(I_p - \lambda Q_\lambda)Q_\lambda X^\top \varepsilon$$

$$+ \mathrm{Lin}_{\mathsf{pd}}(\lambda), \tag{36}$$

where $\mathrm{Lin}_C(\lambda)$ and $\mathrm{Lin}_{\mathsf{pd}}(\lambda)$ are terms linear in $\varepsilon$. Moreover, the quadratic $\varepsilon$-terms in (35)–(36) can be rewritten as

$$\frac{1}{n^2}\varepsilon^\top X Q_\lambda \Sigma(I_p - \lambda Q_\lambda)Q_\lambda X^\top \varepsilon = \frac{1}{n^2}\varepsilon^\top X Q_\lambda \Sigma Q_\lambda X^\top \varepsilon \; - \; \lambda \cdot \frac{1}{n^2}\varepsilon^\top X Q_\lambda \Sigma Q_\lambda^2 X^\top \varepsilon, \tag{37}$$

$$\frac{1}{n^2}\varepsilon^\top X Q_\lambda (I_p - \lambda Q_\lambda)\Sigma(I_p - \lambda Q_\lambda)Q_\lambda X^\top \varepsilon = \frac{1}{n^2}\varepsilon^\top X Q_\lambda \Sigma Q_\lambda X^\top \varepsilon - 2\lambda \cdot \frac{1}{n^2}\varepsilon^\top X Q_\lambda \Sigma Q_\lambda^2 X^\top \varepsilon$$

$$+ \lambda^2 \cdot \frac{1}{n^2}\varepsilon^\top X Q_\lambda^2 \Sigma Q_\lambda^2 X^\top \varepsilon. \tag{38}$$

*Proof.* Plug the decompositions from Lemma C.1 into the definitions of $R = \|e\|_\Sigma^2$, $C = \langle e, \widetilde{e}\rangle_\Sigma$, $R_{\mathsf{pd}} = \|\widetilde{e}\|_\Sigma^2$ and expand. The bias-only terms give the stated $Q_2, Q_3, Q_4$ combinations (using symmetry to replace scalars of the form $\beta^\top Q_\lambda \Sigma Q_\lambda^2 \beta$ by $\tfrac{1}{2}\beta^\top (Q_\lambda \Sigma Q_\lambda^2 + Q_\lambda^2 \Sigma Q_\lambda)\beta$). The terms quadratic in $\varepsilon$ are the displayed quadratic forms. The linear terms $\mathrm{Lin}_C, \mathrm{Lin}_{\mathsf{pd}}$ collect the bias–$\varepsilon$ cross-terms. Finally, (37)–(38) follow by expanding $(I_p - \lambda Q_\lambda)$ and using symmetry of $Q_\lambda$ and $\Sigma$. $\quad\square$

In the misspecified setting, we cannot condition on $X$ and drop the bias–noise cross-terms. Instead we use leave-one-out concentration for ridge-type linear and quadratic forms.

**Lemma C.3** (Response-noise concentration for ridge-type bilinear/quadratic forms). *Assume Assumption 3.1. Fix $\lambda > 0$ and let $\varepsilon_i := y_i - x_i^\top \beta$, so that $\mathbb{E}[z_i \varepsilon_i] = 0$ and $\mathbb{E}[\varepsilon_i^2] = \sigma^2$. Then, as $n, p \to \infty$ with $p/n \to \gamma$:*

*1. All terms linear in $\varepsilon$ appearing in (34)–(36) satisfy $\mathrm{Lin}_\bullet(\lambda) = o_{\mathbb{P}}(1)$.*
*2. The quadratic $\varepsilon$-forms satisfy*

$$\frac{1}{n^2}\varepsilon^\top X Q_\lambda \Sigma Q_\lambda X^\top \varepsilon = \sigma^2 U_2(\lambda) + o_{\mathbb{P}}(1),$$

$$\frac{1}{n^2}\varepsilon^\top X Q_\lambda \Sigma Q_\lambda^2 X^\top \varepsilon = \sigma^2 U_3(\lambda) + o_{\mathbb{P}}(1),$$

$$\frac{1}{n^2}\varepsilon^\top X Q_\lambda^2 \Sigma Q_\lambda^2 X^\top \varepsilon = \sigma^2 U_4(\lambda) + o_{\mathbb{P}}(1),$$

*with $U_k(\lambda)$ defined in (33).*

*Proof outline.* This follows from leave-one-out expansions in Lemmas D.2 (linear forms) and D.3 (off-diagonal quadratic forms) of Patil & Du (2023). At a high level: (i) each linear term is a normalized sum $\frac{1}{n}\sum_{i=1}^n \alpha_i \varepsilon_i$ where $\alpha_i$ is a leave-one-out coefficient built from $Q_\lambda$ and $x_i$; Lemma D.2 gives $o_{\mathbb{P}}(1)$ under $\mathbb{E}[z_i \varepsilon_i] = 0$ and bounded moments; (ii) each quadratic form is $\frac{1}{n^2}\sum_{i,j}\varepsilon_i \varepsilon_j K_{ij}$ for a ridge-type kernel $K$. Lemma D.3 controls the off-diagonal sum $\sum_{i \neq j}$ (giving $o_{\mathbb{P}}(1)$), while the diagonal sum reduces to $\sigma^2 \frac{1}{n}\mathrm{tr}(\cdot)$ up to $o_{\mathbb{P}}(1)$ by the same leave-one-out decoupling. A detailed proof is given in Section C.5. $\quad\square$

Combining Lemmas C.2 and C.3 yields

$$\overline{R}(\lambda) = \beta^\top Q_2(\lambda)\beta + \sigma^2 U_2(\lambda) + o_{\mathbb{P}}(1), \tag{39}$$

$$\overline{C}(\lambda) = \beta^\top \left(2Q_2(\lambda) - \tfrac{1}{2}Q_3(\lambda)\right)\beta + \sigma^2 \left(U_2(\lambda) - \lambda U_3(\lambda)\right) + o_{\mathbb{P}}(1), \tag{40}$$

$$\overline{R}_{\mathsf{pd}}(\lambda) = \beta^\top \left(4Q_2(\lambda) - 2Q_3(\lambda) + Q_4(\lambda)\right)\beta + \sigma^2 \left(U_2(\lambda) - 2\lambda U_3(\lambda) + \lambda^2 U_4(\lambda)\right) + o_{\mathbb{P}}(1). \tag{41}$$

## C.3. Helper Results (Deterministic Equivalents) for the Proof of Theorem 3.2

Fix $\lambda > 0$ and define $\kappa = \kappa(\lambda) > 0$ as the unique solution of

$$\kappa = \lambda + \gamma\kappa\,\overline{\mathrm{tr}}\big(\Sigma(\Sigma + \kappa I_p)^{-1}\big). \tag{42}$$

Let $G := (\Sigma + \kappa I_p)^{-1}$ and define

$$t_k := \gamma\,\overline{\mathrm{tr}}(\Sigma^2 G^k), \qquad k \in \{2,3,4\}, \qquad b := \frac{1}{1 - t_2},$$

and the signal–covariance quadratic forms

$$q_k := \beta^\top G^k \Sigma\,\beta, \qquad k \in \{2,3,4\}.$$

Finally define

$$E := \kappa - b\lambda + b^2\kappa\lambda t_3,$$

$$a_2 := bE^2 + b^4\kappa^2\lambda^2 t_4 + b^5\kappa^2\lambda^2 t_3^2, \qquad a_3 := 2b^2\kappa\lambda E, \qquad a_4 := b^3\kappa^2\lambda^2,$$

and

$$u_2 := t_2 b, \qquad u_3 := t_3 b^3, \qquad u_4 := t_4 b^4 + 2t_3^2 b^5.$$

We use standard anisotropic deterministic-equivalent (local-law) results for sample-covariance resolvents under bounded-spectrum and bounded-moment assumptions; see, e.g., Knowles & Yin (2017); Dobriban & Wager (2018); Dobriban & Sheng (2020); Patil et al. (2022a; 2023). In particular, $Q_\lambda = (\widehat{\Sigma} + \lambda I_p)^{-1}$ admits deterministic equivalents uniformly over $\lambda$ in compact subsets of $(0,\infty)$, and these equivalents can be differentiated with respect to scalar parameters (see Section C.6.1 for more details; see also Appendix E of Patil & Du (2023) for a general background on asymptotic equivalents and a summary of various calculus rules for asymptotic equivalents).

**Lemma C.4** (Deterministic equivalents for $Q_2, Q_3, Q_4$)**.** *For each fixed $\lambda > 0$, we have*

$$Q_2(\lambda) \;\asymp\; \kappa^2 b\, G^2 \Sigma,$$

$$Q_3(\lambda) \;\asymp\; 2\kappa bE\, G^2\Sigma + 2\kappa^2 b^2 \lambda\, G^3\Sigma,$$

$$Q_4(\lambda) \;\asymp\; a_2\, G^2\Sigma + a_3\, G^3\Sigma + a_4\, G^4\Sigma.$$

**Lemma C.5** (Deterministic limits for $U_2, U_3, U_4$)**.** *For each fixed $\lambda > 0$, we have*

$$U_2(\lambda) \overset{\mathrm{p}}{\to} u_2, \qquad U_3(\lambda) \overset{\mathrm{p}}{\to} u_3, \qquad U_4(\lambda) \overset{\mathrm{p}}{\to} u_4.$$

*Proof outlines for Lemmas C.4 and C.5.* These follow from: (i) the anisotropic resolvent deterministic equivalent for $Q_\lambda$ together with the fixed point (42); (ii) differentiation of the two-point deterministic equivalent for $R_{\lambda_1}\Sigma R_{\lambda_2}$ to generate $R\Sigma R^2$, $R^2\Sigma R$, and $R^2\Sigma R^2$; and (iii) the identities $U_3 = -\frac{1}{2}U_2'$ and $U_4 = \frac{1}{6}U_2''$ together with deterministic-equivalent calculus. Detailed proofs are provided below in Section C.6. □

## C.4. Proof of Theorem 3.2

*Proof of Theorem 3.2.* Fix $\lambda > 0$. By (39)–(41),

$$\overline{R}(\lambda) = \beta^\top Q_2(\lambda)\beta + \sigma^2 U_2(\lambda) + o_{\mathbb{P}}(1),$$

$$\overline{C}(\lambda) = \beta^\top\Big(2Q_2(\lambda) - \tfrac{1}{2}Q_3(\lambda)\Big)\beta + \sigma^2\big(U_2(\lambda) - \lambda U_3(\lambda)\big) + o_{\mathbb{P}}(1),$$

$$\overline{R}_{\mathsf{pd}}(\lambda) = \beta^\top\Big(4Q_2(\lambda) - 2Q_3(\lambda) + Q_4(\lambda)\Big)\beta + \sigma^2\big(U_2(\lambda) - 2\lambda U_3(\lambda) + \lambda^2 U_4(\lambda)\big) + o_{\mathbb{P}}(1).$$

We now apply Lemma C.4 in bilinear forms with the deterministic vector $\beta$ (assumed to have $\|\beta\|_2 = O(1)$ so these bilinear forms are $O(1)$):

$$\beta^\top Q_2(\lambda)\beta \to \kappa^2 b\, q_2,$$

$$\beta^\top \Big(2Q_2(\lambda) - \tfrac{1}{2}Q_3(\lambda)\Big)\beta \to 2\kappa^2 b\, q_2 - \big(\kappa b E\, q_2 + \kappa^2 b^2 \lambda\, q_3\big),$$

$$\beta^\top \Big(4Q_2(\lambda) - 2Q_3(\lambda) + Q_4(\lambda)\Big)\beta \to 4\kappa^2 b\, q_2 - 2\big(2\kappa b E\, q_2 + 2\kappa^2 b^2 \lambda\, q_3\big) + (a_2 q_2 + a_3 q_3 + a_4 q_4).$$

Also by Lemma C.5,

$$U_2(\lambda) \to u_2, \qquad U_3(\lambda) \to u_3, \qquad U_4(\lambda) \to u_4.$$

Substituting yields the excess-risk limits:

$$\overline{R}(\lambda) \to \kappa^2 b\, q_2 + \sigma^2 u_2,$$

$$\overline{C}(\lambda) \to 2\kappa^2 b\, q_2 - (\kappa b E\, q_2 + \kappa^2 b^2 \lambda\, q_3) + \sigma^2(u_2 - \lambda u_3),$$

$$\overline{R}_{\mathrm{pd}}(\lambda) \to 4\kappa^2 b\, q_2 - 2(2\kappa b E\, q_2 + 2\kappa^2 b^2 \lambda\, q_3) + (a_2 q_2 + a_3 q_3 + a_4 q_4) + \sigma^2(u_2 - 2\lambda u_3 + \lambda^2 u_4).$$

Finally, recall from the discussion in Section C.1 that the *full* in-distribution quantities in the main paper are obtained by adding $\sigma^2$ to each of $\overline{R}(\lambda), \overline{C}(\lambda), \overline{R}_{\mathrm{pd}}(\lambda)$. Therefore the full limits match exactly the statement of Theorem 3.2.  $\square$

## C.5. Proof of Lemma C.3

*Proof of Lemma C.3.* Fix $\lambda > 0$. Let $\varepsilon_i := y_i - x_i^\top \beta$ and $\varepsilon := (\varepsilon_i)_{i=1}^n$. By construction, $\mathbb{E}[\varepsilon_i] = 0$ and $\mathbb{E}[z_i \varepsilon_i] = 0$, and $\mathbb{E}[|\varepsilon_i|^{4+\nu}] < \infty$.

Write $X = Z\Sigma^{1/2}$, $\widehat{\Sigma} = X^\top X / n$, and

$$Q_\lambda := (\widehat{\Sigma} + \lambda I_p)^{-1}.$$

Define the dual (Gram) resolvent

$$\overline{Q}_\lambda := \Big(\frac{1}{n}XX^\top + \lambda I_n\Big)^{-1} = \Big(\frac{1}{n}Z\Sigma Z^\top + \lambda I_n\Big)^{-1}.$$

A standard push-through identity gives, for every integer $m \geq 1$,

$$Q_\lambda^m \frac{X^\top}{n} = \frac{X^\top}{n}\overline{Q}_\lambda^m, \qquad \text{and hence} \qquad \Sigma^{1/2}Q_\lambda^m \frac{X^\top}{n} = \frac{\Sigma Z^\top}{n}\overline{Q}_\lambda^m. \tag{43}$$

Define also

$$B_2 := \frac{1}{n}X\Sigma X^\top = \frac{1}{n}Z\Sigma^2 Z^\top.$$

All linear terms in Lemma C.2 can be reduced (using (43)) to forms

$$\frac{1}{n} a^\top \Sigma Z^\top \overline{Q}_\lambda^m \varepsilon, \qquad m \in \{1, 2\}, \tag{44}$$

where $a \in \mathbb{R}^p$ is deterministic (or independent of $Z$) with $\|a\|_2 = O(1)$. Likewise, the quadratic noise terms reduce to

$$\frac{1}{n} \varepsilon^\top \overline{Q}_\lambda^m B_2 \overline{Q}_\lambda^\ell \varepsilon, \qquad (m, \ell) \in \{(1,1), (1,2), (2,2)\}. \tag{45}$$

**Linear forms.** For $m = 1$, (44) is exactly Lemma D.2 of Patil & Du (2023) (apply it with $D = \Sigma$ and $g(z_i) = \varepsilon_i$), yielding convergence in probability to 0. For $m = 2$, use the exact identity (Lemma D.4 of Patil & Du (2023))

$$\overline{Q}_\lambda^2 = \frac{1}{t}\big(\overline{Q}_\lambda - \overline{Q}_{\lambda+t}\big) + t\,\overline{Q}_\lambda \overline{Q}_{\lambda+t}\overline{Q}_\lambda, \qquad t > 0.$$

Plugging into (44), the difference-quotient term reduces to a difference of two $m = 1$ linear forms at $\lambda$ and $\lambda+t$, which vanish in probability by Lemma D.2. The remainder is controlled by operator norms: $\|\overline{Q}_\lambda\|_{\mathrm{op}} \leq \lambda^{-1}$, $\|\overline{Q}_{\lambda+t}\|_{\mathrm{op}} \leq (\lambda + t)^{-1}$, $\|\Sigma Z^\top\|_{\mathrm{op}} = \mathcal{O}_\mathbb{P}(\sqrt{n})$, and $\|\varepsilon\|_2 = \mathcal{O}_\mathbb{P}(\sqrt{n})$. Choosing a deterministic $t = t_n \to 0$ (e.g. $t_n = n^{-1/4}$) makes the remainder $o_\mathbb{P}(1)$. Hence all linear terms are $o_\mathbb{P}(1)$.

**Quadratic forms.** Fix $(m, \ell)$ and set $M_{m,\ell} := \overline{Q}_\lambda^m B_2 \overline{Q}_\lambda^\ell$. Write

$$\varepsilon^\top M_{m,\ell}\varepsilon = \sum_i (M_{m,\ell})_{ii}\varepsilon_i^2 + \sum_{i \neq j}(M_{m,\ell})_{ij}\varepsilon_i \varepsilon_j.$$

The off-diagonal sum divided by $n$ converges to 0 in probability by Lemma D.3 of Patil & Du (2023) (after representing $M_{m,\ell}$ via resolvent identities as a finite linear combination of ridge-type resolvent kernels). The diagonal sum equals $\sigma^2 \operatorname{tr}(M_{m,\ell}) + o_{\mathbb{P}}(n)$ by the same leave-one-out decoupling (applied to $\varepsilon_i^2 - \sigma^2$). Therefore,

$$\frac{1}{n}\varepsilon^\top M_{m,\ell}\varepsilon = \frac{\sigma^2}{n}\operatorname{tr}(M_{m,\ell}) + o_{\mathbb{P}}(1).$$

Finally, use cyclicity of trace and commutativity of $Q_\lambda$ with $\widehat{\Sigma}$ to identify these traces with $U_2, U_3, U_4$ (as in the proof outline following Lemma C.3). Substituting back into Lemma C.2 proves Lemma C.3. □

### C.6. Proofs of Lemmas C.4 and C.5

C.6.1. BACKGROUND

We use an anisotropic (bilinear-form) notion of deterministic equivalent.

**Definition C.6** (Deterministic equivalent). *Let $A = A_p$ be a (possibly random) $p \times p$ matrix and $\overline{A} = \overline{A}_p$ a deterministic $p \times p$ matrix. We write $A \asymp \overline{A}$ if for every pair of deterministic vectors $u = u_p, v = v_p$ with $\|u\|_2, \|v\|_2 = O(1)$,*

$$u^\top (A - \overline{A})v \xrightarrow{\mathrm{P}} 0.$$

*If $A(\theta), \overline{A}(\theta)$ depend on a parameter $\theta$ in an open set $\Theta$, we write $A(\theta) \asymp \overline{A}(\theta)$ uniformly on compact subsets of $\Theta$ if the convergence above holds uniformly over $\theta$ in any compact $K \subset \Theta$.*

This notion implies trace convergence whenever operator norms are uniformly bounded. In particular, if $A \asymp \overline{A}$ and $\|A\|_{\mathrm{op}}, \|\overline{A}\|_{\mathrm{op}} = O_{\mathbb{P}}(1)$, then $\overline{\operatorname{tr}}(A) - \overline{\operatorname{tr}}(\overline{A}) \xrightarrow{\mathrm{P}} 0$.

We will also use that uniform deterministic equivalents can be differentiated.

**Lemma C.7** (Differentiate a deterministic equivalent). *Let $A(\theta)$ be random and $\overline{A}(\theta)$ deterministic, both entrywise differentiable in $\theta$ in a neighborhood of $\theta_0$. Assume $A(\theta) \asymp \overline{A}(\theta)$ uniformly in $\theta$ in that neighborhood. Then $A'(\theta_0) \asymp \overline{A}'(\theta_0)$.*

Recall $\widehat{\Sigma} = X^\top X/n$ and $Q_\lambda = (\widehat{\Sigma} + \lambda I_p)^{-1}$. A standard anisotropic local law for sample-covariance resolvents gives the following asymptotic equivalence; see, e.g., (Rubio & Mestre, 2011; Knowles & Yin, 2017; Patil & Du, 2023):

**Lemma C.8** (Scaled resolvent deterministic equivalent). *Under Assumption 3.1, for each fixed $\lambda > 0$,*

$$\lambda Q_\lambda \asymp \kappa(\lambda) (\Sigma + \kappa(\lambda)I_p)^{-1} = \kappa G, \tag{46}$$

*where $\kappa = \kappa(\lambda) > 0$ is the unique solution to*

$$\kappa = \lambda + \gamma \kappa \overline{\operatorname{tr}}(\Sigma(\Sigma + \kappa I_p)^{-1}), \tag{47}$$

*and $G = (\Sigma + \kappa I_p)^{-1}$. The equivalence holds uniformly for $\lambda$ in compact subsets of $(0, \infty)$.*

Define, as in the main paper,

$$t_k := \gamma \overline{\operatorname{tr}}(\Sigma^2 G^k), \quad k \in \{2, 3, 4\}, \qquad b := \frac{1}{1 - t_2}.$$

**Lemma C.9** (Derivative of the fixed-point solution). *The map $\lambda \mapsto \kappa(\lambda)$ is differentiable for $\lambda > 0$ and*

$$\kappa'(\lambda) = b(\lambda) = \frac{1}{1 - t_2(\lambda)}.$$

C.6.2. A TWO-POINT DETERMINISTIC EQUIVALENT VIA BLOCK LINEARIZATION

For $\lambda_1, \lambda_2 > 0$, define $R_i = (\widehat{\Sigma} + \lambda_i I_p)^{-1}$, $\kappa_i = \kappa(\lambda_i)$, $G_i = (\Sigma + \kappa_i I_p)^{-1}$, and

$$t_{12} := \gamma \overline{\operatorname{tr}}(\Sigma^2 G_1 G_2) = \gamma \overline{\operatorname{tr}}(\Sigma G_1 \Sigma G_2). \tag{48}$$

**Lemma C.10** (Two-point bias-resolvent deterministic equivalent). *For each fixed $\lambda_1, \lambda_2 > 0$,*

$$R_1 \Sigma R_2 \asymp \frac{\kappa_1 \kappa_2}{\lambda_1 \lambda_2} \cdot \frac{1}{1 - t_{12}} \, G_1 \Sigma G_2. \tag{49}$$

*Proof.* We use a block-resolvent generating function. For a scalar coupling $J \in \mathbb{R}$, define

$$\mathbb{H}(J) := \begin{pmatrix} \widehat{\Sigma} + \lambda_1 I_p & J\Sigma \\ J\Sigma & \widehat{\Sigma} + \lambda_2 I_p \end{pmatrix}, \qquad \mathbb{G}(J) := \mathbb{H}(J)^{-1}.$$

Differentiate $\mathbb{H}(J)\mathbb{G}(J) = I$:

$$\mathbb{G}'(J) = -\mathbb{G}(J)\, \mathbb{H}'(J)\, \mathbb{G}(J), \qquad \mathbb{H}'(0) = \begin{pmatrix} 0 & \Sigma \\ \Sigma & 0 \end{pmatrix}.$$

At $J = 0$, $\mathbb{G}(0) = \mathrm{diag}(R_1, R_2)$, hence the $(1,2)$ block satisfies the exact identity

$$(\mathbb{G}'(0))_{12} = -R_1 \Sigma R_2. \tag{50}$$

By anisotropic local laws for such deterministic $2 \times 2$ linearizations (uniformly for $J$ in a neighborhood of 0) and Lemma C.7, $(\mathbb{G}'(0))_{12}$ admits a deterministic equivalent obtained by differentiating the deterministic equivalent of $\mathbb{G}(J)$ at $J = 0$. On the diagonal blocks, Lemma C.8 gives $R_i \asymp (\kappa_i/\lambda_i)G_i$. The linearized equation for the off-diagonal block reduces (by commutativity of $G_1, G_2$ with $\Sigma$) to the scalar equivalence

$$(\mathbb{G}'(0))_{12} \asymp -\frac{\kappa_1 \kappa_2}{\lambda_1 \lambda_2} \, \rho_{12} \, G_1 \Sigma G_2,$$

together with the scalar self-consistency relation $\rho_{12} = 1 + t_{12}\rho_{12}$, where $t_{12}$ is given by (48). Solving yields $\rho_{12} = 1/(1 - t_{12})$, and combining with (50) gives (49). $\qquad \square$

### C.6.3. PROOF OF LEMMA C.4

Recall $Q_2, Q_3, Q_4$ from (32) and define $\mathcal{B}_k := G^k \Sigma$ for $k \in \{2, 3, 4\}$.

*Proof of Lemma C.4.* Fix $\lambda > 0$ and let $\kappa = \kappa(\lambda)$, $G = (\Sigma + \kappa I_p)^{-1}$.

**Equivalent for $Q_2$.** Apply Lemma C.10 with $\lambda_1 = \lambda_2 = \lambda$. Then $G_1 = G_2 = G$, $\kappa_1 = \kappa_2 = \kappa$, and $t_{12} = t_2$, so

$$Q_\lambda \Sigma Q_\lambda \asymp \frac{\kappa^2}{\lambda^2} \cdot \frac{1}{1 - t_2} G\Sigma G = \frac{\kappa^2 b}{\lambda^2} G^2 \Sigma.$$

Multiplying by $\lambda^2$ gives

$$Q_2(\lambda) = \lambda^2 Q_\lambda \Sigma Q_\lambda \asymp \kappa^2 b \, \mathcal{B}_2.$$

**Equivalent for $Q_3$.** Let $F(\lambda_1, \lambda_2) := R_{\lambda_1} \Sigma R_{\lambda_2}$. The exact identities

$$\partial_{\lambda_2} F(\lambda_1, \lambda_2) = -R_{\lambda_1} \Sigma R_{\lambda_2}^2, \qquad \partial_{\lambda_1} F(\lambda_1, \lambda_2) = -R_{\lambda_1}^2 \Sigma R_{\lambda_2}$$

imply

$$Q_3(\lambda) = \lambda^3 \big( Q_\lambda \Sigma Q_\lambda^2 + Q_\lambda^2 \Sigma Q_\lambda \big) = -\lambda^3 (\partial_{\lambda_1} + \partial_{\lambda_2}) F(\lambda_1, \lambda_2) \Big|_{\lambda_1 = \lambda_2 = \lambda}.$$

By Lemma C.10, $F \asymp \overline{F} := \alpha H$ where

$$\alpha(\lambda_1, \lambda_2) = \frac{\kappa_1 \kappa_2}{\lambda_1 \lambda_2} \cdot \frac{1}{1 - t_{12}}, \qquad H(\lambda_1, \lambda_2) = G_1 \Sigma G_2.$$

By Lemma C.7, $\partial_{\lambda_i} F \asymp \partial_{\lambda_i} \overline{F}$. A direct product-rule computation on the diagonal $\lambda_1 = \lambda_2 = \lambda$, using $\kappa'(\lambda) = b$ (Lemma C.9) and the trace derivative $\partial_{\lambda_2} t_{12}\big|_{\mathrm{diag}} = -b\, t_3$, yields

$$-\lambda^3(\partial_{\lambda_1} + \partial_{\lambda_2})\overline{F}\Big|_{\mathrm{diag}} = 2\kappa b E\, \mathcal{B}_2 + 2\kappa^2 b^2 \lambda\, \mathcal{B}_3,$$

where $E := \kappa - b\lambda + b^2 \kappa \lambda\, t_3$. Therefore

$$Q_3(\lambda) \asymp 2\kappa b E\, \mathcal{B}_2 + 2\kappa^2 b^2 \lambda\, \mathcal{B}_3.$$

**Equivalent for $Q_4$.** The exact mixed-derivative identity

$$\partial_{\lambda_1} \partial_{\lambda_2} F(\lambda_1, \lambda_2) = R_{\lambda_1}^2 \Sigma R_{\lambda_2}^2$$

implies

$$Q_4(\lambda) = \lambda^4 Q_\lambda^2 \Sigma Q_\lambda^2 = \lambda^4 \partial_{\lambda_1} \partial_{\lambda_2} F(\lambda_1, \lambda_2)\Big|_{\lambda_1 = \lambda_2 = \lambda}.$$

Again by Lemma C.7, we may replace $F$ by $\overline{F} = \alpha H$ and differentiate. On the diagonal, the derivatives satisfy

$$H = \mathcal{B}_2, \qquad \partial_{\lambda_1} H = \partial_{\lambda_2} H = -b\, \mathcal{B}_3, \qquad \partial_{\lambda_1} \partial_{\lambda_2} H = b^2\, \mathcal{B}_4,$$

and the overlap factor $b_{12} = (1 - t_{12})^{-1}$ satisfies

$$\partial_{\lambda_1} b_{12}\Big|_{\mathrm{diag}} = \partial_{\lambda_2} b_{12}\Big|_{\mathrm{diag}} = -b^3 t_3, \qquad \partial_{\lambda_1} \partial_{\lambda_2} b_{12}\Big|_{\mathrm{diag}} = b^4 t_4 + 2b^5 t_3^2,$$

with $t_4 = \gamma\, \overline{\mathrm{tr}}(\Sigma^2 G^4)$. Carrying out the product-rule expansion of $\partial_{\lambda_1} \partial_{\lambda_2} \overline{F}$ and collecting terms in $\mathcal{B}_2, \mathcal{B}_3, \mathcal{B}_4$ yields

$$\lambda^4 \partial_{\lambda_1} \partial_{\lambda_2} \overline{F}\Big|_{\mathrm{diag}} = a_2\, \mathcal{B}_2 + a_3\, \mathcal{B}_3 + a_4\, \mathcal{B}_4,$$

where

$$a_4 = b^3 \kappa^2 \lambda^2, \qquad a_3 = 2b^2 \kappa \lambda E, \qquad a_2 = bE^2 + b^4 \kappa^2 \lambda^2 t_4 + b^5 \kappa^2 \lambda^2 t_3^2.$$

Therefore $Q_4(\lambda) \asymp a_2\, \mathcal{B}_2 + a_3\, \mathcal{B}_3 + a_4\, \mathcal{B}_4$, completing the proof. $\qquad \square$

### C.6.4. PROOF OF LEMMA C.5

*Proof of Lemma C.5.* Fix $\lambda > 0$. Recall $U_k(\lambda) = \frac{1}{n} \mathrm{tr}(\Sigma \widehat{\Sigma} Q_\lambda^k)$ and $p/n \to \gamma$.

**Limit for $U_2$.** Use $\widehat{\Sigma} Q_\lambda = I_p - \lambda Q_\lambda$ to get the exact identity

$$\widehat{\Sigma} Q_\lambda^2 = (I_p - \lambda Q_\lambda) Q_\lambda = Q_\lambda - \lambda Q_\lambda^2,$$

hence

$$U_2(\lambda) = \frac{1}{n} \mathrm{tr}(\Sigma(Q_\lambda - \lambda Q_\lambda^2)) = \gamma\, \overline{\mathrm{tr}}(\Sigma Q_\lambda) - \lambda\gamma\, \overline{\mathrm{tr}}(\Sigma Q_\lambda^2). \tag{51}$$

Define $s(\lambda) := \gamma\, \overline{\mathrm{tr}}(\Sigma Q_\lambda)$. Since $Q_\lambda' = -Q_\lambda^2$,

$$s'(\lambda) = \gamma\, \overline{\mathrm{tr}}(\Sigma Q_\lambda') = -\gamma\, \overline{\mathrm{tr}}(\Sigma Q_\lambda^2),$$

so (51) becomes

$$U_2(\lambda) = s(\lambda) + \lambda s'(\lambda). \tag{52}$$

By Lemma C.8 and trace convergence,

$$s(\lambda) = \gamma\, \overline{\mathrm{tr}}(\Sigma Q_\lambda) \to \gamma\, \overline{\mathrm{tr}}\Big(\Sigma \cdot \frac{\kappa}{\lambda} G\Big) = \frac{\gamma\kappa}{\lambda}\, \overline{\mathrm{tr}}(\Sigma G).$$

Using the fixed-point equation (47), $\gamma\kappa\,\overline{\mathrm{tr}}(\Sigma G) = \kappa - \lambda$, hence

$$s(\lambda) \;\to\; \frac{\kappa - \lambda}{\lambda} = \frac{\kappa}{\lambda} - 1.$$

By uniformity in $\lambda$ locally and Lemma C.7,

$$s'(\lambda) \;\to\; \frac{\kappa'}{\lambda} - \frac{\kappa}{\lambda^2}.$$

Substitute into (52):

$$U_2(\lambda) \;\to\; \Big(\frac{\kappa}{\lambda} - 1\Big) + \lambda\Big(\frac{\kappa'}{\lambda} - \frac{\kappa}{\lambda^2}\Big) = \kappa'(\lambda) - 1 = b - 1 = \frac{t_2}{1 - t_2} = t_2 b =: u_2.$$

**Limits for $U_3$ and $U_4$.** Since $Q'_\lambda = -Q_\lambda^2$, we have the exact identities

$$\frac{d}{d\lambda}Q_\lambda^2 = -2Q_\lambda^3, \qquad \frac{d}{d\lambda}Q_\lambda^3 = -3Q_\lambda^4,$$

and therefore

$$U_2'(\lambda) = -2U_3(\lambda), \qquad U_3'(\lambda) = -3U_4(\lambda),$$

equivalently

$$U_3(\lambda) = -\tfrac{1}{2}U_2'(\lambda), \qquad U_4(\lambda) = \tfrac{1}{6}U_2''(\lambda). \tag{53}$$

From the $U_2$ step, $U_2(\lambda) \to u_2(\lambda) = t_2 b$. By uniformity and Lemma C.7, we may differentiate to obtain $U_2' \to u_2'$ and $U_2'' \to u_2''$. Thus by (53),

$$U_3(\lambda) \to -\tfrac{1}{2}u_2'(\lambda), \qquad U_4(\lambda) \to \tfrac{1}{6}u_2''(\lambda).$$

Finally, compute these derivatives. Since $t_2'(\kappa) = -2t_3$ and $\kappa' = b$, we have $t_2'(\lambda) = -2t_3 b$ and hence

$$u_2(\lambda) = \frac{t_2}{1 - t_2} \;\Rightarrow\; u_2'(\lambda) = \frac{t_2'(\lambda)}{(1 - t_2)^2} = (-2t_3 b)\, b^2 = -2t_3 b^3,$$

so $-\tfrac{1}{2}u_2' = t_3 b^3 =: u_3$. Similarly, using $t_3'(\kappa) = -3t_4$ and $b'(\lambda) = -2t_3 b^3$, we obtain

$$u_2''(\lambda) = \frac{d}{d\lambda}\big(-2t_3 b^3\big) = -2\big(t_3'(\lambda)b^3 + t_3 \cdot 3b^2 b'\big) = 6t_4 b^4 + 12t_3^2 b^5,$$

hence $\tfrac{1}{6}u_2'' = t_4 b^4 + 2t_3^2 b^5 =: u_4$. This proves $U_3 \to u_3$ and $U_4 \to u_4$. $\qquad\square$

### C.7. Proof of Corollary 3.3

*Proof of Corollary 3.3.* Fix $\lambda > 0$. Recall from Theorem 3.2 that the optimal mixing weight and optimal SD risk satisfy

$$\xi^\star(\lambda) \overset{\mathrm{p}}{\to} \frac{\mathscr{R}(\lambda) - \mathscr{C}(\lambda)}{\mathscr{R}(\lambda) + \mathscr{R}_{\mathsf{pd}}(\lambda) - 2\mathscr{C}(\lambda)}, \qquad R_{\mathsf{sd}}^\star(\lambda) \overset{\mathrm{p}}{\to} \mathscr{R}_{\mathsf{sd}}^\star(\lambda) := \mathscr{R}(\lambda) - \frac{(\mathscr{R}(\lambda) - \mathscr{C}(\lambda))^2}{\mathscr{R}(\lambda) + \mathscr{R}_{\mathsf{pd}}(\lambda) - 2\mathscr{C}(\lambda)}.$$

In particular, writing $\mathscr{D}(\lambda) := \mathscr{R}(\lambda) + \mathscr{R}_{\mathsf{pd}}(\lambda) - 2\mathscr{C}(\lambda) \geq 0$, we have $\mathscr{R}_{\mathsf{sd}}^\star(\lambda) < \mathscr{R}(\lambda)$ whenever $\mathscr{R}(\lambda) - \mathscr{C}(\lambda) \neq 0$ and $\mathscr{D}(\lambda) > 0$, and moreover $\mathrm{sign}(\xi^\star(\lambda)) = \mathrm{sign}(\mathscr{R}(\lambda) - \mathscr{C}(\lambda))$ whenever $\mathscr{D}(\lambda) > 0$. Thus it remains to characterize the sign of $\mathscr{R}(\lambda) - \mathscr{C}(\lambda)$ under the isotropic signal prior.

As before, let $\widehat{\Sigma} := X^\top X / n$ and $Q_\lambda := (\widehat{\Sigma} + \lambda I_p)^{-1}$. Under the isotropic prior $\beta \sim \mathcal{N}(0, (r^2/p)I_p)$, for any random matrix $M$ (measurable with respect to $X$), $\mathbb{E}[\beta^\top M\beta \mid X] = \mathrm{tr}(M\mathbb{E}[\beta\beta^\top]) = r^2\,\mathrm{tr}(M)/p$. Likewise, for a noise vector $\varepsilon$ independent of $X$ with $\mathbb{E}[\varepsilon] = 0$ and $\mathbb{E}[\varepsilon\varepsilon^\top] = \sigma^2 I_n$, for any random matrix $M$, $\mathbb{E}[\varepsilon^\top M\varepsilon \mid X] = \sigma^2\,\mathrm{tr}(M)$. Applying these $X$-conditional limits on the conditional risk and correlation expansions (39)–(41) yields (with $o_\mathbb{P}(1)$ remainders absorbed since they are mean-zero after conditioning on $X$):

$$\overline{R}_X(\lambda) := \mathbb{E}\big[\overline{R}(\lambda) \,\big|\, X\big] = \frac{r^2}{p}\,\mathrm{tr}\big(\Sigma\,\lambda^2 Q_\lambda^2\big) + \frac{\sigma^2}{n}\,\mathrm{tr}\big(\Sigma Q_\lambda(I_p - \lambda Q_\lambda)\big), \tag{54}$$

$$\overline{R}_{\mathsf{pd},X}(\lambda) := \mathbb{E}\big[\overline{R}_{\mathsf{pd}}(\lambda) \,\big|\, X\big] = \frac{r^2}{p}\,\mathrm{tr}\Big(\Sigma\,\lambda^2 Q_\lambda^2\,(2I_p - \lambda Q_\lambda)^2\Big) + \frac{\sigma^2}{n}\,\mathrm{tr}\big(\Sigma Q_\lambda (I_p - \lambda Q_\lambda)^3\big), \tag{55}$$

$$\overline{C}_X(\lambda) := \mathbb{E}\big[\overline{C}(\lambda) \,\big|\, X\big] = \frac{r^2}{p}\,\mathrm{tr}\Big(\Sigma\,\lambda^2 Q_\lambda^2\,(2I_p - \lambda Q_\lambda)\Big) + \frac{\sigma^2}{n}\,\mathrm{tr}\big(\Sigma Q_\lambda (I_p - \lambda Q_\lambda)^2\big). \tag{56}$$

Here $\overline{R}, \overline{R}_{\mathsf{pd}}, \overline{C}$ denote the corresponding excess components as above, i.e., $R(\lambda) = \sigma^2 + \overline{R}(\lambda)$, $R_{\mathsf{pd}}(\lambda) = \sigma^2 + \overline{R}_{\mathsf{pd}}(\lambda)$, and $C(\lambda) = \sigma^2 + \overline{C}(\lambda)$.

Define the nonasymptotic critical value $\lambda_{n,p}^\star := \frac{p}{n}\frac{\sigma^2}{r^2}$ so that $\lambda_{n,p}^\star \to \lambda^\star := \gamma\frac{\sigma^2}{r^2}$. A direct simplification of (54) and (56) gives

$$\begin{aligned}
\overline{R}_X(\lambda) - \overline{C}_X(\lambda) &= \lambda\Big(\frac{\sigma^2}{n} - \frac{r^2}{p}\lambda\Big)\Big(\,\mathrm{tr}(\Sigma Q_\lambda^2) - \lambda\,\mathrm{tr}(\Sigma Q_\lambda^3)\Big) \\
&= \lambda(\lambda_{n,p}^\star - \lambda)\,\frac{r^2}{p}\,\mathrm{tr}\big(\Sigma Q_\lambda^2(I_p - \lambda Q_\lambda)\big) \\
&= \lambda(\lambda_{n,p}^\star - \lambda)\,\frac{r^2}{p}\,\mathrm{tr}\big(\Sigma Q_\lambda^2\,\widehat{\Sigma} Q_\lambda\big),
\end{aligned}$$

where in the last step we used $I_p - \lambda Q_\lambda = \widehat{\Sigma} Q_\lambda$, which follows from $(\widehat{\Sigma} + \lambda I_p)Q_\lambda = I_p$. The trace factor is strictly positive for $\lambda > 0$ with probability one because $\Sigma \succ 0$ and $Q_\lambda \succ 0$ for $\lambda > 0$ and hence the trace vanishes if and only if $\widehat{\Sigma} = 0$ (which under under Assumption 3.1 is a measure zero event). Therefore, for each fixed $\lambda > 0$, $\mathrm{sign}(\overline{R}_X(\lambda) - \overline{C}_X(\lambda)) = \mathrm{sign}(\lambda_{n,p}^\star - \lambda)$ with probability one.

Now, by the same concentration arguments of Lemma C.3 used to prove Theorem 3.2, the random quantities $\overline{R}(\lambda) - \overline{C}(\lambda)$ concentrate around their conditional means, and $\overline{R}(\lambda) - \overline{C}(\lambda) - \big(\overline{R}_X(\lambda) - \overline{C}_X(\lambda)\big) \xrightarrow{\mathrm{P}} 0$. Moreover, $\lambda_{n,p}^\star \to \lambda^\star$ and $\overline{R}_X(\lambda) - \overline{C}_X(\lambda)$ converges to $\mathscr{R}(\lambda) - \mathscr{C}(\lambda)$ (since $\sigma^2$ cancels in the difference). Hence, $\mathscr{R}(\lambda) - \mathscr{C}(\lambda) = 0$ if and only if $\lambda = \lambda^\star$, and $\mathrm{sign}(\mathscr{R}(\lambda) - \mathscr{C}(\lambda)) = \mathrm{sign}(\lambda^\star - \lambda)$. Finally, for $\lambda > 0$, note that the limiting discrepancy $\mathscr{D}(\lambda) = \mathscr{R}(\lambda) + \mathscr{R}_{\mathsf{pd}}(\lambda) - 2\mathscr{C}(\lambda)$ is strictly positive with probability one in this case, as it is the limit of $\mathbb{E}[(f_\lambda(x_0) - f_{\mathsf{pd},\lambda}(x_0))^2 \mid \mathcal{D}]$, which is nondegenerate in this case (see also the proof of Theorem E.2 for a more direct argument). □

## C.8. Proof of Proposition 3.4

We provide two proofs. The first evaluates the needed trace functionals by diagonalizing $\widehat{\Sigma}$ and invoking the Marchenko–Pastur law (including negative moments) in the limits $\lambda \to 0$ and $\lambda \to \infty$. The second is a specialization of the general deterministic equivalent in Theorem 3.2 to $\Sigma = I_p$.

### C.8.1. FIRST PROOF

For the first approach, we characterize separately the extreme-$\lambda$ limits of the ridge risk $\mathscr{R}(\lambda)$, the ridge-optimal risk $\mathscr{R}^\star$, and the optimal SD risk $\mathscr{R}_{\mathsf{sd}}^\star(\lambda)$, and then take ratios as in Proposition 3.4.

We first recall some known results from the literature; see, e.g., (Dobriban & Wager, 2018).

**Lemma C.11.** *Assume $\Sigma = I_p$ and $\beta \sim \mathcal{N}(0, (r^2/p)I_p)$, and consider proportional asymptotics $p, n \to \infty$ with $p/n \to \gamma$. Then the asymptotic ridge prediction risk $\mathscr{R}(\lambda)$ satisfies:*

$$\lim_{\lambda \to 0} \mathscr{R}(\lambda) = \begin{cases} \frac{\sigma^2 \gamma}{1-\gamma} + \sigma^2, & \gamma \in (0,1), \\ \frac{r^2(\gamma-1)}{\gamma} + \frac{\sigma^2}{\gamma-1} + \sigma^2, & \gamma \in (1,\infty), \end{cases} \tag{57}$$

$$\lim_{\lambda \to \infty} \mathscr{R}(\lambda) = r^2 + \sigma^2, \quad \forall \gamma \in (0,\infty). \tag{58}$$

**Lemma C.12.** *Assume $\Sigma = I_p$. Then the (proportional) asymptotic ridge-optimal risk is:*

$$\mathscr{R}^\star = \sigma^2 + \frac{1}{2\gamma}\left(-\gamma\sigma^2 + r^2(\gamma-1) + \sqrt{4\gamma^2 r^2 \sigma^2 + (\gamma\sigma^2 - r^2(\gamma-1))^2}\right).$$

Next we characterize the extreme-$\lambda$ limits of $\mathscr{R}_{\mathsf{sd}}^\star(\lambda)$.

**Lemma C.13.** *Assume $\Sigma = I_p$ and $\beta \sim \mathcal{N}(0, (r^2/p)I_p)$, and consider proportional asymptotics $p, n \to \infty$ with $p/n \to \gamma$. Then the asymptotic optimal SD prediction risk $\mathscr{R}^\star_{\mathsf{sd}}(\lambda)$ satisfies*

$$\lim_{\lambda \to 0} \mathscr{R}^\star_{\mathsf{sd}}(\lambda) = \begin{cases} \dfrac{r^2\sigma^2\gamma(1-\gamma)^2 + \sigma^4\gamma^3}{r^2(1-\gamma)^3 + \sigma^2\gamma(1-\gamma^2)} + \sigma^2, & \gamma \in (0,1), \\[3mm] \dfrac{r^4(\gamma-1)^4 + r^2\sigma^2\gamma(\gamma+2)(\gamma-1)^2 + \sigma^4\gamma^2}{r^2\gamma(\gamma-1)^3 + \sigma^2\gamma^2(\gamma^2-1)} + \sigma^2, & \gamma \in (1,\infty), \end{cases} \tag{59}$$

$$\lim_{\lambda \to \infty} \mathscr{R}^\star_{\mathsf{sd}}(\lambda) = \frac{r^4\gamma + r^2\sigma^2\gamma}{r^2(\gamma+1) + \sigma^2\gamma} + \sigma^2, \quad \gamma \in (0,\infty). \tag{60}$$

*Proof of Lemma C.13.* As before, let $Q_\lambda := (\widehat{\Sigma} + \lambda I_p)^{-1}$ with $\widehat{\Sigma} := X^\top X/n$. Recall from (31) the closed-form expression of the optimal SD risk, in terms of $X$-conditional quantities:

$$\mathscr{R}^\star_{\mathsf{sd}}(\lambda) = \sigma^2 + \lim_{n,p \to \infty} \left( \frac{\overline{R}_X(\lambda)\overline{R}_{\mathsf{pd},X}(\lambda) - \overline{C}_X(\lambda)^2}{\overline{R}_X(\lambda) + \overline{R}_{\mathsf{pd},X}(\lambda) - 2\overline{C}_X(\lambda)} \right).$$

Recall the expressions for these terms from equations (54)–(56). Based on the assumption $\Sigma = I$, we have

$$\overline{R}_X(\lambda)\overline{R}_{\mathsf{pd},X}(\lambda) - \overline{C}_X(\lambda)^2$$

$$= \frac{r^4}{p^2}\left[ \operatorname{tr}\!\big(\lambda^2 Q_\lambda^2\big)\operatorname{tr}\!\big(\lambda^2 Q_\lambda^2(2I - \lambda Q_\lambda)^2\big) - \operatorname{tr}\big(\lambda^2 Q_\lambda^2(2I - \lambda Q_\lambda)\big)^2\right]$$

$$+ \frac{r^2\sigma^2}{pn\lambda}\left[ \operatorname{tr}\!\big(\lambda^2 Q_\lambda^2\big)\operatorname{tr}\!\big(\lambda Q_\lambda(I - \lambda Q_\lambda)^3\big) + \operatorname{tr}\!\big(\lambda Q_\lambda(I - \lambda Q_\lambda)\big)\operatorname{tr}\!\big(\lambda^2 Q_\lambda^2(2I - \lambda Q_\lambda)^2\big)\right.$$

$$\left. - 2\operatorname{tr}\big(\lambda^2 Q_\lambda^2(2I - \lambda Q_\lambda)\big)\operatorname{tr}\!\big(\lambda Q_\lambda(I - \lambda Q_\lambda)^2\big)\right]$$

$$+ \frac{\sigma^4}{n^2\lambda^2}\left[ \operatorname{tr}\!\big(\lambda Q_\lambda(I - \lambda Q_\lambda)\big)\operatorname{tr}\!\big(\lambda Q_\lambda(I - \lambda Q_\lambda)^3\big) - \operatorname{tr}\big(\lambda Q_\lambda(I - \lambda Q_\lambda)^2\big)^2\right],$$

$$:= \frac{r^4}{p^2}A_1 + \frac{r^2\sigma^2}{pn\lambda}A_2 + \frac{\sigma^4}{n^2\lambda^2}A_3,$$

$$\overline{R}_X(\lambda) + \overline{R}_{\mathsf{pd},X}(\lambda) - 2\overline{C}_X(\lambda) = \frac{r^2}{p}\operatorname{tr}\big(\lambda^2 Q_\lambda^2(I - \lambda Q_\lambda)^2\big) + \frac{\sigma^2}{n\lambda}\operatorname{tr}\big(\lambda^3 Q_\lambda^3(I - \lambda Q_\lambda)\big).$$

Denote the eigenvalues of the sample covariance matrix $\widehat{\Sigma}$ as $\{s_i\}_{i=1}^p$. Then we can rewritten the traces in the above as:

$$A_1 = \operatorname{tr}\big(\lambda^2 Q_\lambda^2\big)\operatorname{tr}\!\big(\lambda^2 Q_\lambda^2(2I - \lambda Q_\lambda)^2\big) - \operatorname{tr}\big(\lambda^2 Q_\lambda^2(2I - \lambda Q_\lambda)\big)^2$$

$$= \left(\sum_{i=1}^p \frac{\lambda^2}{(s_i + \lambda)^2}\right)\left(\sum_{i=1}^p \frac{\lambda^2}{(s_i + \lambda)^2}\left(2 - \frac{\lambda}{s_i + \lambda}\right)^2\right) - \left(\sum_{i=1}^p \frac{\lambda^2}{(s_i + \lambda)^2}\left(2 - \frac{\lambda}{s_i + \lambda}\right)\right)^2$$

$$\overset{(1)}{=} \frac{1}{2}\sum_{i,j=1}^p \left(\frac{\lambda}{s_i + \lambda}\frac{\lambda}{s_j + \lambda}\left(2 - \frac{\lambda}{s_j + \lambda}\right) - \frac{\lambda}{s_j + \lambda}\frac{\lambda}{s_i + \lambda}\left(2 - \frac{\lambda}{s_i + \lambda}\right)\right)^2$$

$$= \sum_{i,j=1}^p \frac{1}{2}\frac{\lambda^2}{(s_i + \lambda)^2(s_j + \lambda)^2}\left(\frac{\lambda}{s_i + \lambda} - \frac{\lambda}{s_j + \lambda}\right)^2$$

$$= \frac{1}{2}\sum_{i,j=1}^p \frac{\lambda^6(s_i - s_j)^2}{(s_i + \lambda)^4(s_j + \lambda)^4},$$

$$A_2 = \left(\sum_{i=1}^p \frac{\lambda^2}{(s_i + \lambda)^2}\right)\left(\sum_{i=1}^p \frac{\lambda}{s_i + \lambda}\frac{s_i^3}{(s_i + \lambda)^3}\right) + \left(\sum_{i=1}^p \frac{\lambda}{s_i + \lambda}\frac{s_i}{s_i + \lambda}\right)\left(\sum_{i=1}^p \frac{\lambda^2}{(s_i + \lambda)^2}\left(2 - \frac{\lambda}{s_i + \lambda}\right)^2\right)$$

$$- 2\left(\sum_{i=1}^p \frac{\lambda^2}{(s_i + \lambda)^2}\left(2 - \frac{\lambda}{s_i + \lambda}\right)\right)\left(\sum_{i=1}^p \frac{\lambda}{s_i + \lambda}\frac{s_i^2}{(s_i + \lambda)^2}\right)$$

$$\overset{(2)}{=} \sum_{i,j=1}^{p} \left( \frac{\lambda}{s_i + \lambda} \sqrt{\frac{\lambda}{s_j + \lambda}} \frac{s_j^{3/2}}{(s_j + \lambda)^{3/2}} - \frac{\lambda}{s_i + \lambda} \left( 2 - \frac{\lambda}{s_i + \lambda} \right) \sqrt{\frac{\lambda}{s_j + \lambda}} \frac{s_j}{s_j + \lambda} \right)^2$$

$$= \sum_{i,j=1}^{p} \frac{\lambda^2}{(s_i + \lambda)^2} \frac{\lambda}{s_j + \lambda} \frac{s_j}{s_j + \lambda} \left( \frac{s_j}{s_j + \lambda} - 2 + \frac{\lambda}{s_i + \lambda} \right)^2$$

$$= \sum_{i,j=1}^{p} \frac{\lambda^3 s_j}{(s_i + \lambda)^2 (s_j + \lambda)^2} \left( \frac{-\lambda}{s_j + \lambda} + \frac{\lambda}{s_i + \lambda} - 1 \right)^2,$$

$$A_3 = \left( \sum_{i=1}^{p} \frac{\lambda}{s_i + \lambda} \frac{s_i}{s_i + \lambda} \right) \left( \sum_{i=1}^{p} \frac{\lambda}{s_i + \lambda} \frac{s_i^3}{(s_i + \lambda)^3} \right) - \left( \sum_{i=1}^{p} \frac{\lambda}{s_i + \lambda} \frac{s_i^2}{(s_i + \lambda)^2} \right)^2$$

$$\overset{(1)}{=} \frac{1}{2} \sum_{i,j=1}^{p} \left( \sqrt{\frac{\lambda}{s_i + \lambda} \frac{s_i}{s_i + \lambda}} \sqrt{\frac{\lambda}{s_j + \lambda}} \frac{s_j^{3/2}}{(s_j + \lambda)^{3/2}} - \sqrt{\frac{\lambda}{s_j + \lambda} \frac{s_j}{s_j + \lambda}} \sqrt{\frac{\lambda}{s_i + \lambda}} \frac{s_i^{3/2}}{(s_i + \lambda)^{3/2}} \right)^2$$

$$= \frac{1}{2} \sum_{i,j=1}^{p} \frac{\lambda^2 s_i s_j}{(s_i + \lambda)^2 (s_j + \lambda)^2} \left( \frac{s_i}{s_i + \lambda} - \frac{s_j}{s_j + \lambda} \right)^2$$

$$= \frac{1}{2} \sum_{i,j=1}^{p} \frac{\lambda^4 s_i s_j (s_i - s_j)^2}{(s_i + \lambda)^4 (s_j + \lambda)^4},$$

and

$$\overline{R}_X(\lambda) + \overline{R}_{\mathsf{pd},X}(\lambda) - 2\overline{C}_X(\lambda) = \frac{r^2}{p} \sum_{i=1}^{p} \frac{\lambda^2 s_i^2}{(s_i + \lambda)^4} + \frac{\sigma^2}{n} \sum_{i=1}^{p} \frac{\lambda^2 s_i}{(s_i + \lambda)^4}, \tag{61}$$

where $(1)$ and $(2)$ are from the following equalities:

$$\left( \sum_{i=1}^{p} x_i^2 \right) \left( \sum_{i=1}^{p} y_i^2 \right) - \left( \sum_{i=1}^{p} x_i y_i \right)^2 = \frac{1}{2} \sum_{i,j=1}^{p} \left( x_i y_j - x_j y_i \right)^2,$$

$$\left( \sum_{i=1}^{p} x_i^2 \right) \left( \sum_{i=1}^{p} y_i^2 \right) + \left( \sum_{i=1}^{p} z_i^2 \right) \left( \sum_{i=1}^{p} t_i^2 \right) - 2 \left( \sum_{i=1}^{p} x_i t_i \right) \left( \sum_{i=1}^{p} y_i z_i \right) = \sum_{i,j=1}^{p} \left( x_i y_j - z_j t_i \right)^2.$$

**Case 1:** $\lambda \to \infty$ **and** $\gamma \in (0, \infty)$. For this case, we have

$$\lambda^2 \frac{r^4}{p^2} A_1 = \frac{r^4}{2p^2} \sum_{i,j=1}^{p} \frac{\lambda^8 (s_i - s_j)^2}{(s_i + \lambda)^4 (s_j + \lambda)^4} = \frac{r^4}{2p^2} \sum_{i,j=1}^{p} \frac{(s_i - s_j)^2}{(s_i/\lambda + 1)^4 (s_j/\lambda + 1)^4}$$

$$\xrightarrow{\lambda \to \infty} \frac{r^4}{2p^2} \sum_{i,j=1}^{p} (s_i - s_j)^2 \xrightarrow{n,p \to \infty} r^4 \gamma,$$

where we used the fact that $\sum_{i,j=1}^{p} (s_i - s_j)^2 / p^2 = 2 \sum_i s_i^2 / p - 2 (\sum_i s_i / p)^2 \to 2(1 + \gamma) - 2 = 2\gamma$ based on the Marchenko–Pastur law. Similarly,

$$\lambda^2 \frac{r^2 \sigma^2}{pn\lambda} A_2 = \frac{r^2 \sigma^2}{pn} \sum_{i,j=1}^{p} \frac{\lambda^4 s_j}{(s_i + \lambda)^2 (s_j + \lambda)^2} \left( \frac{-\lambda}{s_j + \lambda} + \frac{\lambda}{s_i + \lambda} - 1 \right)^2$$

$$= \frac{r^2 \sigma^2}{pn} \sum_{i,j=1}^{p} \frac{s_j}{(s_i/\lambda + 1)^2 (s_j/\lambda + 1)^2} \left( \frac{-1}{s_j/\lambda + 1} + \frac{1}{s_i/\lambda + 1} - 1 \right)^2$$

$$\xrightarrow{\lambda \to \infty} \frac{r^2 \sigma^2}{pn} \sum_{i,j=1}^{p} s_j = \frac{r^2 \sigma^2}{n} \sum_{j=1}^{p} s_j \xrightarrow{n,p \to \infty} r^2 \sigma^2 \gamma,$$

$$\lambda^2 \frac{\sigma^4}{n^2\lambda^2} A_3 = \frac{\sigma^2}{2n^2} \sum_{i,j=1}^{p} \frac{\lambda^4 s_i s_j (s_i - s_j)^2}{(s_i + \lambda)^4 (s_j + \lambda)^4} \xrightarrow{\lambda \to \infty} 0,$$

$$\lambda^2 (\overline{R}_X(\lambda) + \overline{R}_{\mathsf{pd},X}(\lambda) - 2\overline{C}_X(\lambda)) = \frac{r^2}{p} \sum_{i=1}^{p} \frac{\lambda^4 s_i^2}{(s_i + \lambda)^4} + \frac{\sigma^2}{n} \sum_{i=1}^{p} \frac{\lambda^4 s_i}{(s_i + \lambda)^4}$$

$$\xrightarrow{\lambda \to \infty} \frac{r^2}{p} \sum_{i=1}^{p} s_i^2 + \frac{\sigma^2}{n} \sum_{i=1}^{p} s_i \xrightarrow{n,p \to \infty} \frac{r^2}{p}(1 + \gamma) + \sigma^2 \gamma.$$

From the above limits, we have

$$\lim_{\lambda \to \infty} \mathscr{R}_{\mathsf{sd}}^{\star}(\lambda) = \sigma^2 + \lim_{\lambda \to \infty} \lim_{n,p \to \infty} \left( \frac{\lambda^2 (\overline{R}_X(\lambda) \overline{R}_{\mathsf{pd},X}(\lambda) - \overline{C}_X(\lambda)^2)}{\lambda^2 (\overline{R}_X(\lambda) + \overline{R}_{\mathsf{pd},X}(\lambda) - 2\overline{C}_X(\lambda))} \right) = \sigma^2 + \frac{r^4 \gamma + r^2 \sigma^2 \gamma}{r^2 (1 + \gamma) + \sigma^2 \gamma}.$$

**Case 2:** $\lambda \to 0$ **and** $\gamma < 1$**.** First, we state the following results for negative moments of Marchenko–Pastur law. Assume $X \sim \mathsf{MP}(\gamma)$ with $\gamma < 1$, then from Lemma C.14, we have

$$\mathbb{E}[X^{-1}] = \frac{1}{1 - \gamma}, \quad \mathbb{E}[X^{-2}] = \frac{1}{(1 - \gamma)^3}, \quad \mathbb{E}[X^{-3}] = \frac{1 + \gamma}{(1 - \gamma)^5}.$$

Back to this case, we have

$$\lambda^{-2} \frac{r^4}{p^2} A_1 = \frac{r^4}{2p^2} \sum_{i,j=1}^{p} \frac{\lambda^4 (s_i - s_j)^2}{(s_i + \lambda)^4 (s_j + \lambda)^4} \xrightarrow{\lambda \to 0} 0,$$

$$\lambda^{-2} \frac{r^2 \sigma^2}{pn\lambda} A_2 = \frac{r^2 \sigma^2}{pn} \sum_{i,j=1}^{p} \frac{s_j}{(s_i + \lambda)^2 (s_j + \lambda)^2} \left( \frac{-\lambda}{s_j + \lambda} + \frac{\lambda}{s_i + \lambda} - 1 \right)^2$$

$$\xrightarrow{\lambda \to 0} \frac{r^2 \sigma^2}{pn} \sum_{i,j=1}^{p} \frac{1}{s_i^2 s_j} = \frac{r^2 \sigma^2}{pn} \left( \sum_{i=1}^{p} \frac{1}{s_i^2} \right) \left( \sum_{i=1}^{p} \frac{1}{s_i} \right)$$

$$= \frac{r^2 \sigma^2 p}{n} \left( \frac{1}{p} \sum_{i=1}^{p} \frac{1}{s_i^2} \right) \left( \frac{1}{p} \sum_{i=1}^{p} \frac{1}{s_i} \right) \xrightarrow{n,p \to \infty} \frac{r^2 \sigma^2 \gamma}{(1 - \gamma)^4},$$

$$\lambda^{-2} \frac{\sigma^4}{n^2\lambda^2} A_3 = \frac{\sigma^4}{2n^2} \sum_{i,j=1}^{p} \frac{s_i s_j (s_i - s_j)^2}{(s_i + \lambda)^4 (s_j + \lambda)^4} \xrightarrow{\lambda \to 0} \frac{\sigma^4}{2n^2} \sum_{i,j=1}^{p} \frac{(s_i - s_j)^2}{s_i^3 s_j^3}$$

$$= \frac{\sigma^4}{n^2} \left( \left( \sum_{i=1}^{p} \frac{1}{s_i} \right) \left( \sum_{i=1}^{p} \frac{1}{s_i^3} \right) - \left( \sum_{i=1}^{p} \frac{1}{s_i^2} \right)^2 \right)$$

$$\xrightarrow{n,p \to \infty} \sigma^4 \gamma^2 \left( \frac{1 + \gamma}{(1 - \gamma)^6} - \frac{1}{(1 - \gamma)^6} \right) = \sigma^4 \gamma^2 \frac{\gamma}{(1 - \gamma)^6},$$

$$\lambda^{-2} (\overline{R}_X(\lambda) + \overline{R}_{\mathsf{pd},X}(\lambda) - 2\overline{C}_X(\lambda)) = \frac{r^2}{p} \sum_{i=1}^{p} \frac{s_i^2}{(s_i + \lambda)^4} + \frac{\sigma^2}{n} \sum_{i=1}^{p} \frac{s_i}{(s_i + \lambda)^4}$$

$$\xrightarrow{\lambda \to 0} \frac{r^2}{p} \sum_{i=1}^{p} \frac{1}{s_i^2} + \frac{\sigma^2}{n} \sum_{i=1}^{p} \frac{1}{s_i^3} \xrightarrow{n,p \to \infty} \frac{r^2}{(1 - \gamma)^3} + \frac{\sigma^2 \gamma (1 + \gamma)}{(1 - \gamma)^5}$$

From the above limits, we have

$$\lim_{\lambda \to 0} \mathscr{R}_{\mathsf{sd}}^{\star}(\lambda) = \sigma^2 + \lim_{\lambda \to 0} \lim_{n,p \to \infty} \left( \frac{\lambda^{-2} (\overline{R}_X(\lambda) \overline{R}_{\mathsf{pd},X}(\lambda) - \overline{C}_X(\lambda)^2)}{\lambda^{-2} (\overline{R}_X(\lambda) + \overline{R}_{\mathsf{pd},X}(\lambda) - 2\overline{C}_X(\lambda))} \right) = \sigma^2 + \frac{r^2 \sigma^2 \gamma (1 - \gamma)^2 + \sigma^4 \gamma^3}{r^2 (1 - \gamma)^3 + \sigma^2 \gamma (1 - \gamma^2)}.$$

**Case 3:** $\lambda \to 0$ **and** $\gamma > 1$**.** For this case, the sample covariance is at most rank $n < p$, so we have $s_i = 0$ for $i > n$. Thus

$$\lambda^{-2} \frac{r^4}{p^2} A_1 = \frac{r^4}{2p^2} \sum_{i,j=1}^{p} \frac{\lambda^4 (s_i - s_j)^2}{(s_i + \lambda)^4 (s_j + \lambda)^4}$$

$$= \frac{r^4}{2p^2} \left( \sum_{i=1}^n \sum_{j=1}^n \frac{\lambda^4 (s_i - s_j)^2}{(s_i + \lambda)^4 (s_j + \lambda)^4} + 2 \sum_{i=1}^n \sum_{j=n+1}^p \frac{\lambda^4 (s_i - s_j)^2}{(s_i + \lambda)^4 (s_j + \lambda)^4} \right)$$

$$= \frac{r^4}{2p^2} \left( \sum_{i=1}^n \sum_{j=1}^n \frac{\lambda^4 (s_i - s_j)^2}{(s_i + \lambda)^4 (s_j + \lambda)^4} + 2(p-n) \sum_{i=1}^n \frac{\lambda^4 s_i^2}{(s_i + \lambda)^4 \lambda^4} \right)$$

$$\xrightarrow{\lambda \to 0} \frac{r^4}{2p^2} \left( 0 + 2(p-n) \sum_{i=1}^n \frac{1}{s_i^2} \right) = r^4 \frac{p-n}{p} \left( \frac{1}{p} \sum_{i=1}^n \frac{1}{s_i^2} \right)$$

$$\xrightarrow{n,p \to \infty} r^4 \left( 1 - \frac{1}{\gamma} \right) \frac{1}{(\gamma - 1)^3} = \frac{r^4}{\gamma (\gamma - 1)^2},$$

$$\lambda^{-2} \frac{r^2 \sigma^2}{pn\lambda} A_2 = \frac{r^2 \sigma^2}{pn} \sum_{i,j=1}^p \frac{s_j}{(s_i + \lambda)^2 (s_j + \lambda)^2} \left( \frac{-\lambda}{s_j + \lambda} + \frac{\lambda}{s_i + \lambda} - 1 \right)^2$$

$$= \frac{r^2 \sigma^2}{pn} \left( \sum_{i=1}^n \sum_{j=1}^n \frac{s_j}{(s_i + \lambda)^2 (s_j + \lambda)^2} \left( \frac{-\lambda}{s_j + \lambda} + \frac{\lambda}{s_i + \lambda} - 1 \right)^2 + \sum_{i=n+1}^p \sum_{j=1}^n \frac{s_j}{\lambda^2 (s_j + \lambda)^2} \frac{\lambda^2}{(s_j + \lambda)^2} \right)$$

$$\xrightarrow{\lambda \to 0} \frac{r^2 \sigma^2}{pn} \left( \sum_{i=1}^n \sum_{j=1}^n \frac{1}{s_i^2 s_j} + (p-n) \sum_{j=1}^n \frac{1}{s_j^3} \right)$$

$$= \frac{r^2 \sigma^2 p}{n} \left[ \left( \frac{1}{p} \sum_{i=1}^n \frac{1}{s_i^2} \right) \left( \frac{1}{p} \sum_{i=1}^n \frac{1}{s_i} \right) + \frac{p-n}{p} \left( \frac{1}{p} \sum_{i=1}^n \frac{1}{s_i^3} \right) \right]$$

$$\xrightarrow{n,p \to \infty} r^2 \sigma^2 \gamma \left[ \frac{1}{\gamma (\gamma - 1)^4} + \left( 1 - \frac{1}{\gamma} \right) \frac{\gamma + 1}{(\gamma - 1)^5} \right] = \frac{r^2 \sigma^2 (\gamma + 2)}{(\gamma - 1)^4},$$

$$\lambda^{-2} \frac{\sigma^4}{n^2 \lambda^2} A_3 = \frac{\sigma^4}{2n^2} \sum_{i,j=1}^p \frac{s_i s_j (s_i - s_j)^2}{(s_i + \lambda)^4 (s_j + \lambda)^4} = \frac{\sigma^4}{2n^2} \sum_{i,j=1}^n \frac{s_i s_j (s_i - s_j)^2}{(s_i + \lambda)^4 (s_j + \lambda)^4}$$

$$\xrightarrow{\lambda \to 0} \frac{\sigma^4}{2n^2} \sum_{i,j=1}^n \frac{(s_i - s_j)^2}{s_i^3 s_j^3}$$

$$= \frac{\sigma^4 p^2}{n^2} \left( \left( \frac{1}{p} \sum_{i=1}^n \frac{1}{s_i} \right) \left( \frac{1}{p} \sum_{i=1}^n \frac{1}{s_i^3} \right) - \left( \frac{1}{p} \sum_{i=1}^n \frac{1}{s_i^2} \right)^2 \right)$$

$$\xrightarrow{n,p \to \infty} \sigma^4 \gamma^2 \left( \frac{\gamma + 1}{\gamma (\gamma - 1)^6} - \frac{1}{(\gamma - 1)^6} \right) = \frac{\sigma^4 \gamma}{(\gamma - 1)^6},$$

where we used the following facts about the MP law's negative moments when $\gamma > 1$:

$$\frac{1}{p} \sum_{i=1}^p \frac{1}{s_i} \to \frac{1}{\gamma (\gamma - 1)}, \quad \frac{1}{p} \sum_{i=1}^p \frac{1}{s_i^2} \to \frac{1}{(\gamma - 1)^3} \quad \frac{1}{p} \sum_{i=1}^p \frac{1}{s_i^3} \to \frac{\gamma + 1}{(\gamma - 1)^5}.$$

Similarly, we have

$$\lambda^{-2} (\overline{R}_X(\lambda) + \overline{R}_{\mathsf{pd},X}(\lambda) - 2\overline{C}_X(\lambda)) = \frac{r^2}{p} \sum_{i=1}^p \frac{s_i^2}{(s_i + \lambda)^4} + \frac{\sigma^2}{n} \sum_{i=1}^p \frac{s_i}{(s_i + \lambda)^4}$$

$$= \frac{r^2}{p} \sum_{i=1}^n \frac{s_i^2}{(s_i + \lambda)^4} + \frac{\sigma^2}{n} \sum_{i=1}^n \frac{s_i}{(s_i + \lambda)^4}$$

$$\xrightarrow{\lambda \to 0} \frac{r^2}{p} \sum_{i=1}^n \frac{1}{s_i^2} + \frac{\sigma^2}{n} \sum_{i=1}^n \frac{1}{s_i^3}$$

$$\xrightarrow{n,p \to \infty} \frac{r^2}{(\gamma - 1)^3} + \frac{\sigma^2 \gamma (\gamma + 1)}{(\gamma - 1)^5}.$$

Thus, for $\gamma > 1$, we have

$$\lim_{\lambda \to 0} \mathscr{R}^\star_{\mathsf{sd}}(\lambda) = \sigma^2 + \frac{r^4(\gamma - 1)^4 + r^2\sigma^2\gamma(\gamma + 2)(\gamma - 1)^2 + \sigma^4\gamma^2}{r^2\gamma(\gamma - 1)^3 + \sigma^2\gamma^2(\gamma^2 - 1)}.$$

Finally, the ratios in Proposition 3.4 are obtained by plugging in the asymptotic limits of the corresponding risks using Lemmas C.11–C.13. □

**Lemma C.14.** *Assume the random variable $X \sim MP(\gamma)$ with $\gamma < 1$. Then we have*

$$\mathbb{E}[X^{-1}] = \frac{1}{1 - \gamma}, \quad \mathbb{E}[X^{-2}] = \frac{1}{(1 - \gamma)^3}, \quad \mathbb{E}[X^{-3}] = \frac{1 + \gamma}{(1 - \gamma)^5}.$$

*Proof.* Denote $b = (1 + \sqrt{\gamma})^2$ and $a = (1 - \sqrt{\gamma})^2$. Let $R := \sqrt{(b - x)(x - a)}$, from Equation 2.265 of Gradshteyn & Ryzhik (2007), for $m \geq 2$ we have

$$\int_a^b \frac{\sqrt{R}}{x^m} dx = \frac{(2m - 5)(b + a)}{2(m - 1)ba} \int_a^b \frac{\sqrt{R}}{x^{m-1}} dx - \frac{m - 4}{(m - 1)ba} \int_a^b \frac{\sqrt{R}}{x^{m-2}} dx. \tag{62}$$

Recall the density of the MP law $f(x) = \frac{\sqrt{R}}{2\pi\gamma x}$ for $x \in (a, b)$. For $k \geq 1$, using the above identity, we have

$$\begin{aligned}
\mathbb{E}(X^{-k}) &= \int_a^b \frac{1}{x^k} f(x) dx = \frac{1}{2\pi\gamma} \int_a^b \frac{\sqrt{R}}{x^{k+1}} dx \\
&= \frac{(2k - 3)(1 + \gamma)}{k(1 - \gamma)^2} \mathbb{E}(X^{-k+1}) - \frac{k - 3}{k(1 - \gamma)^2} \mathbb{E}(X^{-k+2}).
\end{aligned}$$

Plug-in $k = 1, 2, 3$ and noting that $\mathbb{E}(X) = 1$, we obtain the desired results. □

### C.8.2. SECOND PROOF

For the second approach, we will specialize the general deterministic equivalent in Theorem 3.2 to the isotropic case.

*Proof of Proposition 3.4.* We specialize Theorem 3.2 to $\Sigma = I_p$. In this case, we have $G = (I_p + \kappa I_p)^{-1} = (1 + \kappa)^{-1} I_p$, and thus

$$t_k = \gamma \, \overline{\mathrm{tr}}(I_p^2 G^k) = \frac{\gamma}{(1 + \kappa)^k}, \qquad q_k = \beta^\top G^k I_p \beta = \frac{\|\beta\|_2^2}{(1 + \kappa)^k}.$$

With $\beta \sim \mathcal{N}(0, (r^2/p) I_p)$, $\|\beta\|_2^2 \to r^2$ in probability, so $q_k \to r^2/(1 + \kappa)^k$ in probability. The fixed-point equation becomes

$$\kappa = \lambda + \frac{\gamma\kappa}{1 + \kappa}, \tag{63}$$

whose unique nonnegative solution is

$$\kappa(\lambda) = \frac{1}{2}\left((\lambda + \gamma - 1) + \sqrt{(\lambda + \gamma - 1)^2 + 4\lambda}\right). \tag{64}$$

In particular,

$$\kappa(\lambda) \xrightarrow[\lambda \to \infty]{} \infty, \qquad \kappa(\lambda) \xrightarrow[\lambda \to 0]{} (\gamma - 1)_+.$$

Let $\overline{\mathscr{R}}^\star_{\mathsf{sd}}(\lambda) := \mathscr{R}^\star_{\mathsf{sd}}(\lambda) - \sigma^2$ denote the excess prediction risk. Substituting the isotropic specializations into the general formula and simplifying yields the explicit rational form (as a function of $\kappa = \kappa(\lambda)$):

$$\overline{\mathscr{R}}^\star_{\mathsf{sd}}(\lambda) = \frac{\gamma\left(r^4\kappa^4 + r^2\sigma^2(1 + \kappa)^4 + \gamma^2\sigma^2(r^2 + \sigma^2) - 2\gamma r^2\sigma^2(2\kappa + 1)\right)}{\left((1 + \kappa)^2 - \gamma\right)\left(r^2\left((1 + \gamma)(1 + \kappa)^2 - 4\gamma(1 + \kappa) + \gamma(1 + \gamma)\right) + \sigma^2\gamma\left((1 + \kappa)^2 + \gamma\right)\right)}. \tag{65}$$

**Case 1:** $\lambda \to \infty$. Using $\kappa(\lambda) \to \infty$, the leading terms in (65) give

$$\lim_{\lambda \to \infty} \overline{\mathscr{R}}_{\mathsf{sd}}^{\star}(\lambda) = \frac{\gamma r^2 (r^2 + \sigma^2)}{r^2(\gamma + 1) + \gamma \sigma^2},$$

and adding back $\sigma^2$ and writing in terms of SNR yields the stated $\lambda \to \infty$ limit.

**Case 2:** $\lambda \to 0$. If $\gamma \in (0, 1)$ then $\kappa(\lambda) \to 0$; substituting $\kappa = 0$ in (65) yields the stated $\gamma < 1$ limit. If $\gamma \in (1, \infty)$ then $\kappa(\lambda) \to \gamma - 1$; substituting $\kappa = \gamma - 1$ yields the stated $\gamma > 1$ limit. Adding back $\sigma^2$ and expressing in terms of SNR completes the proof. $\qquad\square$

## D. Proofs in Section 4

### D.1. Preliminaries

Fix $\lambda > 0$. Let $X \in \mathbb{R}^{n \times p}$ and $y \in \mathbb{R}^n$, and define the sample covariance $\widehat{\Sigma} := X^\top X/n$ and ridge resolvent $Q_\lambda := (\widehat{\Sigma} + \lambda I_p)^{-1}$. Define the ridge hat (smoother) matrix

$$H_\lambda := X Q_\lambda X^\top / n \in \mathbb{R}^{n \times n}.$$

The ridge teacher coefficient and fitted values are

$$\beta_\lambda := Q_\lambda X^\top y / n, \qquad \widehat{y}_\lambda := X \beta_\lambda = H_\lambda y.$$

Define the pure-distilled ridge coefficient (ridge refit on pseudo-labels $\widehat{y}_\lambda$)

$$\beta_{\mathsf{pd},\lambda} := Q_\lambda X^\top \widehat{y}_\lambda / n, \qquad \widehat{y}_{\mathsf{pd},\lambda} := X \beta_{\mathsf{pd},\lambda}.$$

Since $\widehat{y}_\lambda = H_\lambda y$ and $H_\lambda = X Q_\lambda X^\top / n$, a direct calculation yields the identity used repeatedly below:

$$\widehat{y}_{\mathsf{pd},\lambda} = H_\lambda^2 y. \tag{66}$$

Thus both the teacher and PD predictors are linear smoothers in $y$, with smoothing matrices $H_\lambda$ and $H_\lambda^2$, respectively. Consequently, their degrees of freedom equal the traces of these matrices:

$$\mathsf{df}_\lambda = \mathrm{tr}(H_\lambda), \qquad \mathsf{df}_{\mathsf{pd},\lambda} = \mathrm{tr}(H_\lambda^2).$$

Recall the GCV residuals and risk/correlation estimators defined in Equations (13) and (14):

$$\widehat{r}_\lambda := \frac{y - \widehat{y}_\lambda}{1 - \mathsf{df}_\lambda / n}, \qquad \widehat{r}_{\mathsf{pd},\lambda} := \frac{y - \widehat{y}_{\mathsf{pd},\lambda}}{1 - \mathsf{df}_{\mathsf{pd},\lambda} / n},$$

and

$$\widehat{R}(\lambda) := \frac{\|\widehat{r}_\lambda\|_2^2}{n}, \quad \widehat{R}_{\mathsf{pd}}(\lambda) := \frac{\|\widehat{r}_{\mathsf{pd},\lambda}\|_2^2}{n}, \quad \widehat{C}(\lambda) := \frac{\langle \widehat{r}_\lambda, \widehat{r}_{\mathsf{pd},\lambda} \rangle}{n}.$$

Let $(x_0, y_0)$ be an independent test pair distributed as in Assumption 3.1 and independent of the training data. Define the out-of-sample errors

$$e_\lambda := y_0 - x_0^\top \beta_\lambda, \qquad e_{\mathsf{pd},\lambda} := y_0 - x_0^\top \beta_{\mathsf{pd},\lambda},$$

and the corresponding conditional (oracle) quantities

$$R(\lambda) := \mathbb{E}[e_\lambda^2 \mid X, y], \qquad R_{\mathsf{pd}}(\lambda) := \mathbb{E}[e_{\mathsf{pd},\lambda}^2 \mid X, y], \qquad C(\lambda) := \mathbb{E}[e_\lambda e_{\mathsf{pd},\lambda} \mid X, y].$$

Finally define

$$D(\lambda) := R(\lambda) + R_{\mathsf{pd}}(\lambda) - 2C(\lambda) = \mathbb{E}\big[(e_\lambda - e_{\mathsf{pd},\lambda})^2 \mid X, y\big] \geq 0. \tag{67}$$

We will use the following ridge functional estimation result (Theorem 4 of Patil et al. (2022b)):

**Lemma D.1** (Ridge functional estimation via GCV/LOOCV). *Assume the conditions of Assumption 3.1 and $p/n \to \gamma$. Fix $\lambda > 0$. Let $t : \mathbb{R} \to \mathbb{R}$ be continuous and satisfy a quadratic growth condition. Define $T_\lambda := \mathbb{E}[t(e_\lambda) \mid X, y]$. Let $T_\lambda^{\mathrm{gcv}}$ and $T_\lambda^{\mathrm{loo}}$ be the plug-in estimators based on the usual ridge GCV and LOOCV residuals. Then*

$$T_\lambda^{\mathrm{gcv}} - T_\lambda \xrightarrow{\mathrm{p}} 0, \qquad T_\lambda^{\mathrm{loo}} - T_\lambda \xrightarrow{\mathrm{p}} 0.$$

### D.2. Helper Results (Risk and Correlation Components Consistency) for the Proof of Theorem 4.1

**Lemma D.2** (Consistency of the ridge-risk estimator). *Under Assumption 3.1 and $p/n \to \gamma$, for each fixed $\lambda > 0$,*

$$\widehat{R}(\lambda) - R(\lambda) \xrightarrow{\text{P}} 0.$$

*Proof.* This simply follows from Lemma D.1 with $t(z) = z^2$. Then $T_\lambda = \mathbb{E}[e_\lambda^2 \mid X, y] = R(\lambda)$, and the ridge GCV plug-in estimator is precisely $\widehat{R}(\lambda) = \|\widehat{r}_\lambda\|_2^2/n$. $\square$

**Lemma D.3** (Consistency of the pure-distilled risk and correlation estimators). *Under Assumption 3.1 and $p/n \to \gamma$, for each fixed $\lambda > 0$,*

$$\widehat{R}_{\mathsf{pd}}(\lambda) - R_{\mathsf{pd}}(\lambda) \xrightarrow{\text{P}} 0, \qquad \widehat{C}(\lambda) - C(\lambda) \xrightarrow{\text{P}} 0.$$

*Proof.* We follow the same three-part argument used by Patil et al. (2022b, Proof of Theorem 3): (i) relate the target risks/correlation to leave-one-out (LOO) errors, (ii) show a stability approximation that replaces true LOO errors by diagonal-corrected residuals computed from the full-sample smoother, and (iii) replace the diagonal correction by the GCV trace correction using diagonal–trace equivalence.

*Part (i): LOO concentration around $R_{\mathsf{pd}}(\lambda)$ and $C(\lambda)$.* Let $\beta_\lambda^{(-i)}$ and $\beta_{\mathsf{pd},\lambda}^{(-i)}$ be the teacher and PD coefficients trained without sample $i$, and define the corresponding LOO prediction errors

$$e_{i,\lambda}^{\mathrm{loo}} := y_i - x_i^\top \beta_\lambda^{(-i)}, \qquad e_{i,\mathsf{pd},\lambda}^{\mathrm{loo}} := y_i - x_i^\top \beta_{\mathsf{pd},\lambda}^{(-i)}.$$

Since $(x_i, y_i)$ is independent of $\sigma\{(x_j, y_j) : j \neq i\}$ and identically distributed to $(x_0, y_0)$, we have

$$\mathbb{E}[(e_{i,\mathsf{pd},\lambda}^{\mathrm{loo}})^2 \mid X^{(-i)}, y^{(-i)}] = R_{\mathsf{pd}}^{(-i)}(\lambda), \qquad \mathbb{E}[e_{i,\lambda}^{\mathrm{loo}} e_{i,\mathsf{pd},\lambda}^{\mathrm{loo}} \mid X^{(-i)}, y^{(-i)}] = C^{(-i)}(\lambda),$$

where $R_{\mathsf{pd}}^{(-i)}(\lambda)$ and $C^{(-i)}(\lambda)$ denote the analogues of $R_{\mathsf{pd}}(\lambda)$ and $C(\lambda)$ for the $(n-1)$-sample-trained predictors.

Using the same martingale-difference decomposition and Burkholder inequality arguments as in Patil et al. (2022b, Supplement S.1.2), together with the moment bounds in Assumption 3.1, one obtains

$$\left| \frac{1}{n} \sum_{i=1}^n (e_{i,\mathsf{pd},\lambda}^{\mathrm{loo}})^2 - \frac{1}{n} \sum_{i=1}^n R_{\mathsf{pd}}^{(-i)}(\lambda) \right| \xrightarrow{\text{P}} 0, \qquad \left| \frac{1}{n} \sum_{i=1}^n e_{i,\lambda}^{\mathrm{loo}} e_{i,\mathsf{pd},\lambda}^{\mathrm{loo}} - \frac{1}{n} \sum_{i=1}^n C^{(-i)}(\lambda) \right| \xrightarrow{\text{P}} 0.$$

Moreover, by LOO stability of ridge and the PD stability in Lemma D.5 (see Part (ii) below),

$$\frac{1}{n} \sum_{i=1}^n \left( R_{\mathsf{pd}}^{(-i)}(\lambda) - R_{\mathsf{pd}}(\lambda) \right) \xrightarrow{\text{P}} 0, \qquad \frac{1}{n} \sum_{i=1}^n \left( C^{(-i)}(\lambda) - C(\lambda) \right) \xrightarrow{\text{P}} 0,$$

because (conditionally on the training data) the test risks are Lipschitz in the coefficient vector: for instance,

$$\left| R_{\mathsf{pd}}^{(-i)}(\lambda) - R_{\mathsf{pd}}(\lambda) \right| = \left| \mathbb{E}[(x_0^\top (\beta_{\mathsf{pd},\lambda}^{(-i)} - \beta_{\mathsf{pd},\lambda}))^2 \mid X, y] \right| \leq \|\Sigma\|_{\mathrm{op}} \|\beta_{\mathsf{pd},\lambda}^{(-i)} - \beta_{\mathsf{pd},\lambda}\|_2^2.$$

Combining yields

$$\frac{1}{n} \sum_{i=1}^n (e_{i,\mathsf{pd},\lambda}^{\mathrm{loo}})^2 \xrightarrow{\text{P}} R_{\mathsf{pd}}(\lambda), \qquad \frac{1}{n} \sum_{i=1}^n e_{i,\lambda}^{\mathrm{loo}} e_{i,\mathsf{pd},\lambda}^{\mathrm{loo}} \xrightarrow{\text{P}} C(\lambda). \tag{68}$$

*Part (ii): Equivalence of diagonal-corrected residuals and LOO errors.* Define the diagonal-corrected residuals based on the full-sample smoothing matrices:

$$\bar{r}_{\lambda,i} := \frac{y_i - (H_\lambda y)_i}{1 - (H_\lambda)_{ii}}, \qquad \bar{r}_{\mathsf{pd},i} := \frac{y_i - (H_\lambda^2 y)_i}{1 - (H_\lambda^2)_{ii}}.$$

For ridge, the shortcut formula is exact: $e_{i,\lambda}^{\text{loo}} = \bar{r}_{\lambda,i}$ for all $i$. For the PD two-stage procedure, leaving out one sample changes both the teacher and student fits; however, by the stability statement in Lemma D.5 (together with ridge stability), these perturbations are negligible on average. In particular, one obtains

$$\frac{1}{n}\sum_{i=1}^{n}\left(e_{i,\text{pd},\lambda}^{\text{loo}} - \bar{r}_{\text{pd},i}\right)^2 \xrightarrow{\text{P}} 0. \tag{69}$$

Combining (68) and (69) yields

$$\frac{1}{n}\sum_{i=1}^{n}\bar{r}_{\text{pd},i}^2 \xrightarrow{\text{P}} R_{\text{pd}}(\lambda), \qquad \frac{1}{n}\sum_{i=1}^{n}\bar{r}_{\lambda,i}\bar{r}_{\text{pd},i} \xrightarrow{\text{P}} C(\lambda), \tag{70}$$

where for the cross term we use $\bar{r}_{\lambda,i} = e_{i,\lambda}^{\text{loo}}$ exactly and Cauchy–Schwarz.

*Part (iii): Equivalence of trace-corrected residuals to diagonal-corrected residuals.* By Lemma D.6,

$$\max_{1\leq i\leq n}\left|\frac{1}{1-\text{tr}(H_\lambda^2)/n} - \frac{1}{1-(H_\lambda^2)_{ii}}\right| \xrightarrow{\text{P}} 0,$$

and the analogous ridge diagonal–trace equivalence (see Patil et al. (2022b, Supplement S.1.3)) gives the same statement with $H_\lambda$ in place of $H_\lambda^2$. Since $\widehat{r}_{\text{pd},\lambda,i} = (y_i - (H_\lambda^2 y)_i)/(1-\text{tr}(H_\lambda^2)/n)$ and $\widehat{r}_{\lambda,i} = (y_i - (H_\lambda y)_i)/(1-\text{tr}(H_\lambda)/n)$, it follows that

$$\frac{1}{n}\sum_{i=1}^{n}(\widehat{r}_{\text{pd},\lambda,i} - \bar{r}_{\text{pd},i})^2 \xrightarrow{\text{P}} 0, \qquad \frac{1}{n}\sum_{i=1}^{n}(\widehat{r}_{\lambda,i} - \bar{r}_{\lambda,i})^2 \xrightarrow{\text{P}} 0.$$

Combining with (70) and using Cauchy–Schwarz yields

$$\frac{1}{n}\sum_{i=1}^{n}\widehat{r}_{\text{pd},\lambda,i}^2 \xrightarrow{\text{P}} R_{\text{pd}}(\lambda), \qquad \frac{1}{n}\sum_{i=1}^{n}\widehat{r}_{\lambda,i}\widehat{r}_{\text{pd},\lambda,i} \xrightarrow{\text{P}} C(\lambda),$$

which is exactly $\widehat{R}_{\text{pd}}(\lambda) \xrightarrow{\text{P}} R_{\text{pd}}(\lambda)$ and $\widehat{C}(\lambda) \xrightarrow{\text{P}} C(\lambda)$. $\square$

## D.3. Proof of Theorem 4.1

*Proof of Theorem 4.1.* By Lemmas D.2 and D.3,

$$\left(\widehat{R}(\lambda), \widehat{R}_{\text{pd}}(\lambda), \widehat{C}(\lambda)\right) \xrightarrow{\text{P}} \left(R(\lambda), R_{\text{pd}}(\lambda), C(\lambda)\right).$$

In particular,

$$\widehat{D}(\lambda) := \widehat{R}(\lambda) + \widehat{R}_{\text{pd}}(\lambda) - 2\widehat{C}(\lambda) \xrightarrow{\text{P}} D(\lambda) := R(\lambda) + R_{\text{pd}}(\lambda) - 2C(\lambda).$$

Under the proportional regime, Theorem 3.2 implies that $D(\lambda)$ converges in probability to a deterministic limit $\mathscr{D}(\lambda)$ (given by $\mathscr{R}(\lambda) + \mathscr{R}_{\text{pd}}(\lambda) - 2\mathscr{C}(\lambda)$). In all nondegenerate settings, $\mathscr{D}(\lambda) > 0$ (equivalently, the teacher and PD predictions do not coincide asymptotically), hence $D(\lambda)$ and $\widehat{D}(\lambda)$ are bounded away from $0$ in probability. Therefore, the plug-in maps

$$(a,b,c) \mapsto \frac{a-c}{a+b-2c}, \qquad (a,b,c) \mapsto a - \frac{(a-c)^2}{a+b-2c}$$

are continuous with high probability in a neighborhood of $(R(\lambda), R_{\text{pd}}(\lambda), C(\lambda))$, and the continuous mapping theorem yields

$$\widehat{\xi}^\star(\lambda) - \xi^\star(\lambda) \xrightarrow{\text{P}} 0, \qquad \widehat{R}_{\text{sd}}^\star(\lambda) - R_{\text{sd}}^\star(\lambda) \xrightarrow{\text{P}} 0.$$

(If $\mathscr{D}(\lambda) = 0$, then the oracle SD objective is asymptotically flat in $\xi$; in this case $R_{\text{sd}}^\star(\lambda) = R(\lambda)$ and the risk-consistency conclusion remains valid, while $\xi^\star(\lambda)$ is not identifiable.) $\square$

## D.4. Technical Lemmas

We establish two technical ingredients for the pure-distilled smoother $H_\lambda^2$: (i) leave-one-out stability and (ii) diagonal-trace equivalence.

### D.4.1. LEAVE-ONE-OUT STABILITY

For $i \in \{1, \ldots, n\}$, let $(X^{(-i)}, y^{(-i)})$ denote the dataset with observation $i$ removed and $\widehat{\Sigma}^{(-i)} := (X^{(-i)})^\top X^{(-i)}/(n-1)$. Let $Q_\lambda^{(-i)} := (\widehat{\Sigma}^{(-i)} + \lambda I_p)^{-1}$ and $\beta_\lambda^{(-i)} := Q_\lambda^{(-i)}(X^{(-i)})^\top y^{(-i)}/(n-1)$. Define the PD coefficient trained without $i$ by

$$\beta_{\mathsf{pd},\lambda}^{(-i)} := Q_\lambda^{(-i)}(X^{(-i)})^\top \widehat{y}_\lambda^{(-i)}/(n-1), \qquad \widehat{y}_\lambda^{(-i)} := X^{(-i)}\beta_\lambda^{(-i)}.$$

**Lemma D.4** (Operator-Lipschitz property of $M_\lambda(\cdot)$)**.** *For $\lambda > 0$, define $M_\lambda(A) := A(A + \lambda I)^{-1} = I - \lambda(A + \lambda I)^{-1}$. Then for any symmetric $A, B \succeq 0$,*

$$\|M_\lambda(A) - M_\lambda(B)\|_{\mathrm{op}} \leq \frac{1}{\lambda} \|A - B\|_{\mathrm{op}}.$$

*Proof.* Using $M_\lambda(A) = I - \lambda(A + \lambda I)^{-1}$ and the resolvent identity,

$$M_\lambda(A) - M_\lambda(B) = -\lambda\big[(A + \lambda I)^{-1} - (B + \lambda I)^{-1}\big] = -\lambda(A + \lambda I)^{-1}(B - A)(B + \lambda I)^{-1}.$$

Taking operator norms and using $\|(A + \lambda I)^{-1}\|_{\mathrm{op}} \leq 1/\lambda$ and $\|(B + \lambda I)^{-1}\|_{\mathrm{op}} \leq 1/\lambda$ yields the claim. □

**Lemma D.5** (Average stability of the pure-distilled coefficient)**.** *Under Assumption 3.1 and $p/n \to \gamma$, for each fixed $\lambda > 0$,*

$$\frac{1}{n} \sum_{i=1}^n \big\|\beta_{\mathsf{pd},\lambda} - \beta_{\mathsf{pd},\lambda}^{(-i)}\big\|_2^2 \xrightarrow{\mathrm{p}} 0.$$

*Proof.* Write $\beta_{\mathsf{pd},\lambda} = M_\lambda(\widehat{\Sigma})\beta_\lambda$ and $\beta_{\mathsf{pd},\lambda}^{(-i)} = M_\lambda(\widehat{\Sigma}^{(-i)})\beta_\lambda^{(-i)}$. Then

$$\beta_{\mathsf{pd},\lambda} - \beta_{\mathsf{pd},\lambda}^{(-i)} = M_\lambda(\widehat{\Sigma})\big(\beta_\lambda - \beta_\lambda^{(-i)}\big) + \big(M_\lambda(\widehat{\Sigma}) - M_\lambda(\widehat{\Sigma}^{(-i)})\big)\beta_\lambda^{(-i)}.$$

Using $(a + b)^2 \leq 2a^2 + 2b^2$ and $\|Au\|_2 \leq \|A\|_{\mathrm{op}}\|u\|_2$, we obtain

$$\frac{1}{n} \sum_{i=1}^n \|\beta_{\mathsf{pd},\lambda} - \beta_{\mathsf{pd},\lambda}^{(-i)}\|_2^2 \leq \frac{2}{n} \sum_{i=1}^n \|M_\lambda(\widehat{\Sigma})\|_{\mathrm{op}}^2 \|\beta_\lambda - \beta_\lambda^{(-i)}\|_2^2 + \frac{2}{n} \sum_{i=1}^n \|(M_\lambda(\widehat{\Sigma}) - M_\lambda(\widehat{\Sigma}^{(-i)}))\beta_\lambda^{(-i)}\|_2^2. \tag{71}$$

Since $\|M_\lambda(\cdot)\|_{\mathrm{op}} \leq 1$, the first term in (71) is bounded by $\frac{2}{n} \sum_i \|\beta_\lambda - \beta_\lambda^{(-i)}\|_2^2$, which converges to 0 in probability by the ridge stability result of Patil et al. (2022b) (used there to control LOOCV/GCV).

For the second term, use the exact identity $M_\lambda(A) = I - \lambda(A + \lambda I)^{-1}$ to write

$$M_\lambda(\widehat{\Sigma}) - M_\lambda(\widehat{\Sigma}^{(-i)}) = -\lambda\big(Q_\lambda - Q_\lambda^{(-i)}\big).$$

By the resolvent identity,

$$Q_\lambda - Q_\lambda^{(-i)} = Q_\lambda(\widehat{\Sigma}^{(-i)} - \widehat{\Sigma})Q_\lambda^{(-i)}.$$

Moreover, $\widehat{\Sigma} = \frac{n-1}{n}\widehat{\Sigma}^{(-i)} + \frac{1}{n}x_i x_i^\top$, so $\widehat{\Sigma}^{(-i)} - \widehat{\Sigma} = \frac{1}{n}\big(\widehat{\Sigma}^{(-i)} - x_i x_i^\top\big)$. Define $v_i := Q_\lambda^{(-i)}\beta_\lambda^{(-i)}$. Then

$$(M_\lambda(\widehat{\Sigma}) - M_\lambda(\widehat{\Sigma}^{(-i)}))\beta_\lambda^{(-i)} = \frac{\lambda}{n} Q_\lambda\big(x_i x_i^\top - \widehat{\Sigma}^{(-i)}\big)v_i.$$

Since $\|Q_\lambda\|_{\mathrm{op}} \leq 1/\lambda$, it suffices to control $n^{-1} \sum_i \|(x_i x_i^\top - \widehat{\Sigma}^{(-i)})v_i\|_2^2/n^2$. Condition on $\sigma\{(x_j, y_j) : j \neq i\}$, under which $v_i$ is deterministic and independent of $x_i$. Using $\|x_i x_i^\top v_i\|_2^2 = \|x_i\|_2^2(x_i^\top v_i)^2$ and bounded-spectrum $\Sigma$ from Assumption 3.1,

$$\mathbb{E}\Big[\|x_i x_i^\top v_i\|_2^2 \,\Big|\, X^{(-i)}, y^{(-i)}\Big] = \mathbb{E}[\|x_i\|_2^2(x_i^\top v_i)^2 \mid X^{(-i)}, y^{(-i)}] \lesssim p\|v_i\|_2^2,$$

while $\|\widehat{\Sigma}^{(-i)}v_i\|_2^2 \lesssim \|v_i\|_2^2$ since $\|\widehat{\Sigma}^{(-i)}\|_{\mathrm{op}} = O_p(1)$ for proportional random design. Therefore,

$$\mathbb{E}\Big[\|(x_i x_i^\top - \widehat{\Sigma}^{(-i)})v_i\|_2^2 \,\Big|\, X^{(-i)}, y^{(-i)}\Big] \lesssim p\|v_i\|_2^2,$$

and hence

$$\mathbb{E}\left[\|(M_\lambda(\widehat{\Sigma}) - M_\lambda(\widehat{\Sigma}^{(-i)}))\beta_\lambda^{(-i)}\|_2^2\right] \lesssim \frac{p}{n^2}\,\mathbb{E}[\|v_i\|_2^2].$$

Finally, $\|v_i\|_2 = \|Q_\lambda^{(-i)}\beta_\lambda^{(-i)}\|_2 \leq \|Q_\lambda^{(-i)}\|_{\mathrm{op}}\|\beta_\lambda^{(-i)}\|_2 \leq \lambda^{-1}\|\beta_\lambda^{(-i)}\|_2$, and for fixed $\lambda > 0$ we have $\|\beta_\lambda^{(-i)}\|_2 = O_p(1)$ uniformly in $i$ under Assumption 3.1. Since $p/n \to \gamma$, we have $p/n^2 = O(1/n)$, and averaging over $i$ yields

$$\frac{1}{n}\sum_{i=1}^n \|(M_\lambda(\widehat{\Sigma}) - M_\lambda(\widehat{\Sigma}^{(-i)}))\beta_\lambda^{(-i)}\|_2^2 \xrightarrow{\mathrm{P}} 0.$$

Together with the first term in (71), this proves the claim. $\qquad\square$

### D.4.2. DIAGONAL-TRACE EQUIVALENCE

Let $S := XX^\top/n \in \mathbb{R}^{n\times n}$ and $G_\lambda := (S + \lambda I_n)^{-1}$. Recall the standard identity

$$H_\lambda = S(S + \lambda I_n)^{-1} = I_n - \lambda G_\lambda,$$

so

$$H_\lambda^2 = (I_n - \lambda G_\lambda)^2 = I_n - 2\lambda G_\lambda + \lambda^2 G_\lambda^2. \tag{72}$$

**Lemma D.6** (Diagonal-trace equivalence for $H_\lambda^2$). *Fix $\lambda > 0$. Under Assumption 3.1 and $p/n \to \gamma$,*

$$\max_{1\leq i\leq n}\left|(H_\lambda^2)_{ii} - \frac{1}{n}\operatorname{tr}(H_\lambda^2)\right| \xrightarrow{p} 0,$$

*and hence*

$$\max_{1\leq i\leq n}\left|\frac{1}{1 - \operatorname{tr}(H_\lambda^2)/n} - \frac{1}{1 - (H_\lambda^2)_{ii}}\right| \xrightarrow{p} 0.$$

*Proof.* By (72), for each $i$,

$$(H_\lambda^2)_{ii} - \frac{1}{n}\operatorname{tr}(H_\lambda^2) = -2\lambda\left((G_\lambda)_{ii} - \frac{1}{n}\operatorname{tr}(G_\lambda)\right) + \lambda^2\left((G_\lambda^2)_{ii} - \frac{1}{n}\operatorname{tr}(G_\lambda^2)\right).$$

Thus it suffices to prove

$$\max_i\left|(G_\lambda)_{ii} - \frac{1}{n}\operatorname{tr}(G_\lambda)\right| \xrightarrow{p} 0, \qquad \max_i\left|(G_\lambda^2)_{ii} - \frac{1}{n}\operatorname{tr}(G_\lambda^2)\right| \xrightarrow{p} 0.$$

1. *Control for $G_\lambda$:* Since $1 - (H_\lambda)_{ii} = \lambda(G_\lambda)_{ii}$ and $1 - \operatorname{tr}(H_\lambda)/n = \lambda\operatorname{tr}(G_\lambda)/n$, the diagonal–trace equivalence for ridge (see Patil et al. (2022b, Supplement S.1.3)) implies $\max_i|(G_\lambda)_{ii} - \operatorname{tr}(G_\lambda)/n| \xrightarrow{\mathrm{P}} 0$.

2. *Control for $G_\lambda^2$:* Fix any $t > 0$ and define $G_{\lambda+t} := (S + (\lambda + t)I_n)^{-1}$. Since $G_\lambda$ and $G_{\lambda+t}$ are both functions of $S$, they commute, and we have the exact identity

$$G_\lambda^2 = \frac{G_\lambda - G_{\lambda+t}}{t} + t\,G_{\lambda+t}\,G_\lambda^2. \tag{73}$$

Taking diagonal–trace differences and maxima gives

$$\max_i\left|(G_\lambda^2)_{ii} - \tfrac{1}{n}\operatorname{tr}(G_\lambda^2)\right| \leq \frac{1}{t}\max_i\left|(G_\lambda)_{ii} - \tfrac{1}{n}\operatorname{tr}(G_\lambda)\right| + \frac{1}{t}\max_i\left|(G_{\lambda+t})_{ii} - \tfrac{1}{n}\operatorname{tr}(G_{\lambda+t})\right| + 2t\,\|G_{\lambda+t}G_\lambda^2\|_{\mathrm{op}}.$$

Since $\|G_\lambda\|_{\mathrm{op}} \leq 1/\lambda$ and $\|G_{\lambda+t}\|_{\mathrm{op}} \leq 1/(\lambda + t)$, the last term is bounded by $2t/\lambda^3$. Letting $n \to \infty$ and using Step 1 at $\lambda$ and at $\lambda + t$ yields

$$\limsup_{n\to\infty}\max_i\left|(G_\lambda^2)_{ii} - \tfrac{1}{n}\operatorname{tr}(G_\lambda^2)\right| \leq \frac{2t}{\lambda^3} \qquad \text{in probability.}$$

Since $t > 0$ was arbitrary, sending $t \to 0$ gives the desired diagonal–trace control for $G_\lambda^2$.

This proves the first claim. For the second claim, note that $0 \preceq H_\lambda^2 \preceq I_n$, so $1 - (H_\lambda^2)_{ii} > 0$ and $1 - \operatorname{tr}(H_\lambda^2)/n > 0$. Moreover, for fixed $\lambda > 0$ these denominators are bounded away from 0 with high probability. Hence

$$\left|\frac{1}{1 - \operatorname{tr}(H_\lambda^2)/n} - \frac{1}{1 - (H_\lambda^2)_{ii}}\right| = \frac{\left|(H_\lambda^2)_{ii} - \operatorname{tr}(H_\lambda^2)/n\right|}{\left(1 - \operatorname{tr}(H_\lambda^2)/n\right)\left(1 - (H_\lambda^2)_{ii}\right)},$$

and the second claim follows from the first. $\qquad\square$

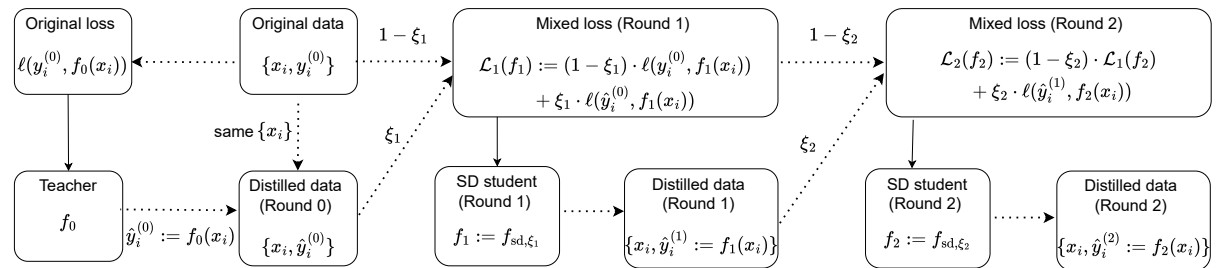

*Figure 11.* Visual illustration of the recursive multi-round self-distillation process for ridge regression.

# E. Further Details in Section 5

## E.1. Results and Proofs in Section 5.1

This appendix states and proves Proposition E.1 and records a simple structural reason why the anchored update need not be monotone. All arguments are purely geometric/structural. We condition on the observed training sample $\mathcal{D} := (X, y)$ and do not assume any (parametric) model. We recall the setup in more details and state the main result below.

Fix $\lambda > 0$ and set $y^{(0)} := y$. A natural *recursive* multi-round scheme iteratively self-distills using the previous round's ridge predictions. For each $k \geq 0$, let $f_\lambda^{(k)}$ denote the teacher ridge regression trained on $(X, y^{(k)})$ with penalty $\lambda$. and write $\widehat{y}_\lambda^{(k)} := f_\lambda^{(k)}(X) \in \mathbb{R}^n$ for its fitted values. Let $f_{\mathsf{pd},\lambda}^{(k)}$ denote the corresponding PD refit of round $k$, i.e., ridge trained on $(X, \widehat{y}_\lambda^{(k)})$ with the same penalty $\lambda$. Given a mixing weight $\xi_{k+1} \in \mathbb{R}$, define the round-$(k+1)$ SD predictor $f_{\mathsf{sd},\lambda,\xi_{k+1}}^{(k+1)}$ as the ridge fit obtained from the same mixed-loss construction as in (2), but with base labels $y^{(k)}$ and teacher pseudo-labels $\widehat{y}_\lambda^{(k)}$. A simplification shows this is equivalent to fitting ridge regression on the mixed labels:

$$y_{\lambda,\xi_{k+1}}^{(k+1)} := (1 - \xi_{k+1})\, y^{(k)} + \xi_{k+1}\, \widehat{y}_\lambda^{(k)}. \tag{74}$$

Because ridge is linear in the response, the round-$(k+1)$ predictor $f_{\mathsf{sd},\lambda,\xi_{k+1}}^{(k+1)}$ lies on the affine path:

$$f_{\mathsf{sd},\lambda,\xi_{k+1}}^{(k+1)} = (1 - \xi_{k+1})\, f_\lambda^{(k)} + \xi_{k+1}\, f_{\mathsf{pd},\lambda}^{(k)}. \tag{75}$$

*Proof of Equation* (75). Fix $k \geq 0$ and $\xi \in \mathbb{R}$. Recall that the round-$(k+1)$ SD predictor $f_{\mathsf{sd},\lambda,\xi_{k+1}}^{(k+1)}$ is defined as:

$$f_{\mathsf{sd},\lambda,\xi}^{(k+1)}(x) := x^\top \underset{\beta \in \mathbb{R}^p}{\arg\min} \left\{ (1 - \xi)\|y^{(k)} - X\beta\|_2^2/n + \xi\|\widehat{y}_\lambda^{(k)} - X\beta\|_2^2/n + \lambda\|\beta\|_2^2 \right\}. \tag{76}$$

Expand the objective in (76) (dropping constants independent of $\beta$):

$$(1 - \xi)\|y^{(k)} - X\beta\|_2^2/n + \xi\|\widehat{y}_\lambda^{(k)} - X\beta\|_2^2/n + \lambda\|\beta\|_2^2$$
$$\equiv \|(1 - \xi)y^{(k)} + \xi\widehat{y}_\lambda^{(k)} - X\beta\|_2^2/n + \lambda\|\beta\|_2^2 \;=\; \|y_{\lambda,\xi}^{(k+1)} - X\beta\|_2^2/n + \lambda\|\beta\|_2^2,$$

where $y_{\lambda,\xi}^{(k+1)} = (1 - \xi)y^{(k)} + \xi\widehat{y}_\lambda^{(k)}$ is exactly the mixed-label vector in (74). Thus $f_{\mathsf{sd},\lambda,\xi}^{(k+1)}$ coincides with ridge regression trained on $(X, y_{\lambda,\xi}^{(k+1)})$ with penalty $\lambda$. Since ridge is linear in its response vector, we have

$$f_{\mathsf{sd},\lambda,\xi}^{(k+1)} = (1 - \xi)\, f_\lambda^{(k)} + \xi\, f_{\mathsf{pd},\lambda}^{(k)},$$

because $f_\lambda^{(k)}$ is ridge on response $y^{(k)}$ and $f_{\mathsf{pd},\lambda}^{(k)}$ is ridge on response $\widehat{y}_\lambda^{(k)} = f_\lambda^{(k)}(X)$. This is (75). $\qquad\square$

As in Section 2, we evaluate any predictor $f$ by the conditional squared prediction risk $R(f) := \mathbb{E}[(y_0 - f(x_0))^2 \mid \mathcal{D}]$ for a (possibly out-of-distribution) test pair $(x_0, y_0)$ with finite conditional second moments. Define the round-$k$ risk:

$R_k(\lambda) := R(f_\lambda^{(k)})$, round-$k$ correlation: $C_k(\lambda) := \mathbb{E}[(y_0 - f_\lambda^{(k)}(x_0))(y_0 - f_{\mathsf{pd},\lambda}^{(k)}(x_0)) \mid \mathcal{D}]$, and round-$k$ discrepancy: $D_k(\lambda) := \mathbb{E}[(f_\lambda^{(k)}(x_0) - f_{\mathsf{pd},\lambda}^{(k)}(x_0))^2 \mid \mathcal{D}] \geq 0$. For each round $k \geq 0$, let $\xi_{k+1}^\star(\lambda) \in \arg\min_{\xi_{k+1} \in \mathbb{R}} R(f_{\mathsf{sd},\lambda,\xi_{k+1}}^{(k+1)})$ denote the optimal (unconstrained) mixing weight. Write $y^{(k+1)} = y_{\lambda,\xi_{k+1}^\star}^{(k+1)}$, $f_\lambda^{(k+1)} := f_{\mathsf{sd},\lambda,\xi_{k+1}^\star}^{(k+1)}$, and $R_{k+1}(\lambda) := R(f_\lambda^{(k+1)})$.

All the one-round structural identities from Section 2 apply at each round $k$ by viewing $y^{(k)}$ as the "base" labels and $(f_\lambda^{(k)}, f_{\mathsf{pd},\lambda}^{(k)})$ as the round-$k$ (teacher, PD) pair. In particular, whenever $D_k(\lambda) > 0$, the round-wise optimizer and improvement satisfy the same formulas as Proposition 2.1:

$$\xi_{k+1}^\star(\lambda) = \frac{R_k(\lambda) - C_k(\lambda)}{D_k(\lambda)}, \quad R_{k+1}(\lambda) = R_k(\lambda) - \frac{(R_k(\lambda) - C_k(\lambda))^2}{D_k(\lambda)}.$$

This leads to the following risk monotonicity result for optimal multi-round SD:

**Proposition E.1** (Monotonicity of optimal recursive multi-round self-distillation). *Fix $\lambda > 0$. The optimal SD risks are monotone (weakly) decreasing in the number of rounds $k$:*

$$R_{k+1}(\lambda) \leq R_k(\lambda) \quad \text{for all } k \geq 0.$$

*Assuming $D_k(\lambda) > 0$, the optimal SD mixing weights and risks admit the closed forms:*

$$\xi_{k+1}^\star(\lambda) = -\frac{\lambda}{2}\frac{R_k'(\lambda)}{D_k(\lambda)}, \quad R_{k+1}(\lambda) = R_k(\lambda) - \frac{\lambda^2}{4}\frac{(R_k'(\lambda))^2}{D_k(\lambda)}, \tag{77}$$

*where $R_k'(\lambda)$ is the derivative along the ridge path for the round-$k$ teacher with $y^{(k)}$ held fixed. Thus, if $R_k'(\lambda) \neq 0$, then $R_{k+1}(\lambda) < R_k(\lambda)$, and $\operatorname{sign}(\xi_{k+1}^\star(\lambda)) = -\operatorname{sign}(R_k'(\lambda))$.*

*Proof of Proposition E.1.* Fix $k \geq 0$. By Equation (75), the family $\{f_{\mathsf{sd},\lambda,\xi}^{(k+1)} : \xi \in \mathbb{R}\}$ is the affine path between $f_\lambda^{(k)}$ and $f_{\mathsf{pd},\lambda}^{(k)}$.

*Monotonicity.* Since $\xi = 0$ is feasible and $f_{\mathsf{sd},\lambda,0}^{(k+1)} = f_\lambda^{(k)}$, we have

$$R_{k+1}(\lambda) = \min_{\xi \in \mathbb{R}} R(f_{\mathsf{sd},\lambda,\xi}^{(k+1)}) \leq R(f_{\mathsf{sd},\lambda,0}^{(k+1)}) = R(f_\lambda^{(k)}) = R_k(\lambda).$$

*Closed forms.* When $D_k(\lambda) > 0$, applying Proposition 2.1 to the affine path between $f_\lambda^{(k)}$ and $f_{\mathsf{pd},\lambda}^{(k)}$ yields the displayed round-wise closed forms (with $(R, C, D)$ replaced by $(R_k, C_k, D_k)$). Likewise, (77) follows by applying Theorem 2.2 at round $k$, again with $(f_\lambda, f_{\mathsf{pd},\lambda}, R, C, D)$ replaced by $(f_\lambda^{(k)}, f_{\mathsf{pd},\lambda}^{(k)}, R_k, C_k, D_k)$, and interpreting $R_k'(\lambda)$ as the derivative with respect to the ridge penalty while holding the base labels $y^{(k)}$ fixed. □

As with all results in Section 2, Proposition E.1 is purely structural: it holds conditionally on $\mathcal{D} = (X, y)$ with probability one and for any squared prediction risk (including out-of-distribution).

### E.2. Results and Proofs in Section 5.2

We compare the same-$X$ self-distillation geometry of the main paper to a common pseudo-labeling variant in which the refit uses an *independent* design matrix. We state our theorem below.

**Theorem E.2** (Same-$X$ dominates fresh-$X$). *Under Assumption 3.1 with $\Sigma = I_p$ and $\beta \sim \mathcal{N}(0, (r^2/p)I_p)$, as $m, n, p \to \infty$ with $p/m, p/n \to \gamma \in (0, \infty)$, for every fixed $\lambda > 0$, we have*

$$\mathscr{R}_{\mathsf{sd}}^{\star,\mathrm{same}}(\lambda) \leq \mathscr{R}_{\mathsf{sd}}^{\star,\mathrm{fraff}}(\lambda),$$

*where $\mathscr{R}_{\mathsf{sd}}^{\star,\mathrm{same}}(\lambda)$ and $\mathscr{R}_{\mathsf{sd}}^{\star,\mathrm{fraff}}(\lambda)$ denote the limiting (deterministic) optimal SD risks for the same-$X$ SD predictor (3) and the affine fresh-$X$ SD predictor (18), respectively*

This result is somewhat counterintuitive as one might expect that additional unlabeled data would help self-distillation. It highlights that the gains characterized in the main paper are intimately tied to self-distillation on the same data.

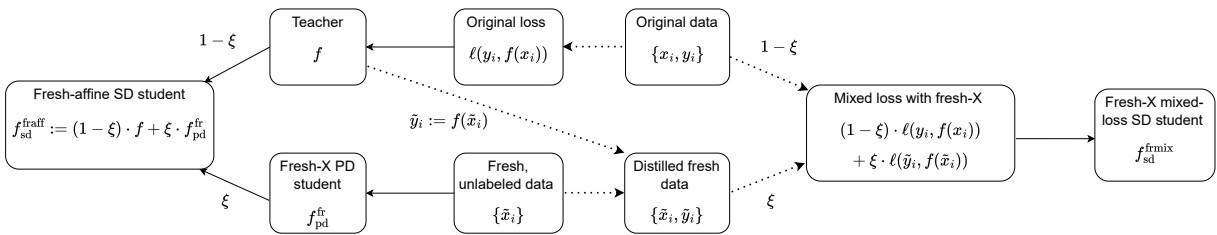

*Figure 12.* Visual illustration of self-distillation with fresh unlabeled features.

### E.2.1. FRESH-X MIXED-LOSS STUDENT REPRESENTATION

Recall the fresh-$X$ mixed-loss SD predictor $f_{\text{sd},\lambda,\xi}^{\text{frmix}}$ from (17). Writing $f_{\text{sd},\lambda,\xi}^{\text{frmix}}(x) = x^\top \beta_{\text{sd},\lambda}^{\text{frmix}}(\xi)$, we derive an explicit closed form for $\beta_{\text{sd},\lambda}^{\text{frmix}}(\xi)$ and show that, unlike the same-$X$ setting, the map $\xi \mapsto f_{\text{sd},\lambda,\xi}^{\text{frmix}}$ is generally not an affine path.

Let $\widehat{\Sigma} := X^\top X/n$, $\widetilde{\Sigma} := \widetilde{X}^\top \widetilde{X}/m$, $\widehat{v} := X^\top y/n$, and define the ridge resolvents:

$$Q_\lambda := (\widehat{\Sigma} + \lambda I_p)^{-1}, \qquad \widetilde{Q}_\lambda := (\widetilde{\Sigma} + \lambda I_p)^{-1}.$$

The ridge teacher coefficient is $\beta_\lambda = Q_\lambda \widehat{v}$. The fresh-$X$ PD refit trained on $(\widetilde{X}, \widetilde{y}_\lambda)$ with $\widetilde{y}_\lambda = \widetilde{X}\beta_\lambda$ has coefficient

$$\beta_{\text{pd},\lambda}^{\text{fr}} := (\widetilde{\Sigma} + \lambda I_p)^{-1}\frac{1}{m}\widetilde{X}^\top \widetilde{y}_\lambda = (\widetilde{\Sigma} + \lambda I_p)^{-1}\widetilde{\Sigma}\beta_\lambda = \widetilde{Q}_\lambda \widetilde{\Sigma}\beta_\lambda.$$

(Here $\widetilde{Q}_\lambda \widetilde{\Sigma} = \widetilde{\Sigma}\widetilde{Q}_\lambda$ since both are functions of $\widetilde{\Sigma}$.)

**Lemma E.3** (Fresh-$X$ mixed-loss student representation). *Fix $\lambda > 0$ and let $\xi \in \mathbb{R}$ be such that the mixed-loss objective* (17) *is strictly convex in $\beta$, equivalently $A_\xi := (1-\xi)\widehat{\Sigma} + \xi\widetilde{\Sigma} + \lambda I_p \succ 0$. (In particular, this holds for all $\xi \in [0,1]$.) Then the coefficient of the fresh-$X$ mixed-loss student satisfies*

$$\beta_{\text{sd},\lambda}^{\text{frmix}}(\xi) = A_\xi^{-1}\big((1-\xi)\widehat{v} + \xi\,\widetilde{\Sigma}\beta_\lambda\big) = \big((1-\xi)Q_\lambda^{-1} + \xi\widetilde{Q}_\lambda^{-1}\big)^{-1}\big((1-\xi)\widehat{v} + \xi\,\widetilde{\Sigma}\beta_\lambda\big). \tag{78}$$

*Moreover, for $\xi \notin \{0,1\}$, writing $S := Q_\lambda \widetilde{Q}_\lambda^{-1}$ (so $S^{-1} = \widetilde{Q}_\lambda Q_\lambda^{-1}$), we have the matrix-weighted decomposition*

$$\beta_{\text{sd},\lambda}^{\text{frmix}}(\xi) = \Big(I + \tfrac{\xi}{1-\xi}S\Big)^{-1}\beta_\lambda + \Big(I + \tfrac{1-\xi}{\xi}S^{-1}\Big)^{-1}\beta_{\text{pd},\lambda}^{\text{fr}}, \tag{79}$$

*with the boundary cases $\xi \in \{0,1\}$ obtained by continuity.*

*Proof.* Expanding (17) and setting the gradient with respect to $\beta$ to zero yields the normal equations

$$\big((1-\xi)\widehat{\Sigma} + \xi\widetilde{\Sigma} + \lambda I_p\big)\beta = (1-\xi)\widehat{v} + \xi\,\widetilde{\Sigma}\beta_\lambda,$$

which gives the first identity in (78); the second follows from $Q_\lambda^{-1} = \widehat{\Sigma} + \lambda I_p$ and $\widetilde{Q}_\lambda^{-1} = \widetilde{\Sigma} + \lambda I_p$. For (79), substitute $\widehat{v} = Q_\lambda^{-1}\beta_\lambda$ and $\widetilde{\Sigma}\beta_\lambda = \widetilde{Q}_\lambda^{-1}\beta_{\text{pd},\lambda}^{\text{fr}}$ into (78), then split the two terms and factor $Q_\lambda^{-1}$ (respectively $\widetilde{Q}_\lambda^{-1}$) from the left. $\square$

The upshot of Lemma E.3 is that, unless $S = I_p$ (equivalently $Q_\lambda = \widetilde{Q}_\lambda$), the map $\xi \mapsto \beta_{\text{sd},\lambda}^{\text{frmix}}(\xi)$ is not a scalar affine combination $(1-\xi)\beta_\lambda + \xi\beta_{\text{pd},\lambda}^{\text{fr}}$. In particular, $\xi \mapsto f_{\text{sd},\lambda,\xi}^{\text{frmix}}$ is generally not an affine path, unlike the same-$X$ case in (3).

### E.2.2. SAME-$X$ SD UNDER ISOTROPIC SETTING

Here we specialize the general deterministic equivalents from Theorem 3.2 to the isotropic setting $\Sigma = I_p$ with isotropic signal $\beta \sim \mathcal{N}(0, (r^2/p)I_p)$.

Define the companion Stieltjes transform $v = v(\lambda) > 0$ as the unique solution of

$$\frac{1}{v} = \lambda + \frac{\gamma}{1+v}, \qquad \lambda > 0, \tag{80}$$

and set $\kappa(\lambda) := 1/v(\lambda)$. In the isotropic case, these admit the explicit closed forms

$$\kappa(\lambda) = \frac{(\lambda + \gamma - 1) + \sqrt{(\lambda + \gamma - 1)^2 + 4\lambda}}{2}, \qquad v(\lambda) = \frac{\sqrt{(\lambda + \gamma - 1)^2 + 4\lambda} - (\lambda + \gamma - 1)}{2\lambda}.$$

Since $\Sigma = I_p$, we have $G = (\Sigma + \kappa I_p)^{-1} = (1 + \kappa)^{-1} I_p$. Hence for $k \in \{2, 3, 4\}$,

$$
\begin{aligned}
t_k &:= \gamma \frac{\mathrm{tr}(\Sigma^2 G^k)}{p} = \gamma \Big(\frac{1}{1 + \kappa}\Big)^k = \gamma \Big(\frac{v}{1 + v}\Big)^k, \\
q_k &:= \mathbb{E}[\beta^\top G^k \Sigma \beta] = r^2 \Big(\frac{1}{1 + \kappa}\Big)^k = r^2 \Big(\frac{v}{1 + v}\Big)^k.
\end{aligned}
\tag{81}
$$

(Equivalently, $q_k = \lim_{p \to \infty} \beta^\top G^k \Sigma \beta$ in probability, by isotropy of $\beta$.)

Define

$$b := (1 - t_2)^{-1}, \qquad E := \kappa - b\lambda + b^2 \kappa \lambda t_3,$$

the variance-trace limits

$$u_2 := t_2 b, \qquad u_3 := t_3 b^3, \qquad u_4 := t_4 b^4 + 2t_3^2 b^5,$$

and

$$a_2 := bE^2 + b^4 \kappa^2 \lambda^2 t_4 + b^5 \kappa^2 \lambda^2 t_3^2, \qquad a_3 := 2b^2 \kappa \lambda E, \qquad a_4 := b^3 \kappa^2 \lambda^2.$$

**Lemma E.4** (Isotropic limits for same-$X$ SD). *Fix $\lambda > 0$. Under Assumption 3.1 with $\Sigma = I_p$ and $\beta \sim \mathcal{N}(0, (r^2/p)I_p)$, as $n, p \to \infty$ with $p/n \to \gamma \in (0, \infty)$, the same-$X$ optimal SD risk satisfies*

$$R_{\mathsf{sd}}^{\star,\mathrm{same}}(\lambda) \xrightarrow{\mathrm{p}} \mathscr{R}_{\mathsf{sd}}^{\star,\mathrm{same}}(\lambda) := \mathscr{R}(\lambda) - \frac{\big(\mathscr{R}(\lambda) - \mathscr{C}^{\mathrm{same}}(\lambda)\big)^2}{\mathscr{R}(\lambda) + \mathscr{R}_{\mathsf{pd}}^{\mathrm{same}}(\lambda) - 2\mathscr{C}^{\mathrm{same}}(\lambda)}.$$

*Here $\mathscr{R}(\lambda)$, $\mathscr{R}_{\mathsf{pd}}^{\mathrm{same}}(\lambda)$, and $\mathscr{C}^{\mathrm{same}}(\lambda)$ are the deterministic limits of the (one-round) teacher risk $R(\lambda)$, same-$X$ PD risk $R_{\mathsf{pd}}^{\mathrm{same}}(\lambda)$, and residual correlation $C^{\mathrm{same}}(\lambda)$, respectively (in the same-$X$ setting of Section 2, these coincide with $R(\lambda)$, $R_{\mathsf{pd}}(\lambda)$, and $C(\lambda)$), given by:*

$$
\begin{aligned}
\mathscr{R}(\lambda) &= \sigma^2 + \kappa^2 b\, q_2 + \sigma^2 u_2, \\
\mathscr{C}^{\mathrm{same}}(\lambda) &= \sigma^2 + 2\kappa^2 b\, q_2 - \big(\kappa bE\, q_2 + \kappa^2 b^2 \lambda\, q_3\big) + \sigma^2 (u_2 - \lambda u_3), \\
\mathscr{R}_{\mathsf{pd}}^{\mathrm{same}}(\lambda) &= \sigma^2 + 4\kappa^2 b\, q_2 - 2\big(2\kappa bE\, q_2 + 2\kappa^2 b^2 \lambda\, q_3\big) + (a_2 q_2 + a_3 q_3 + a_4 q_4) + \sigma^2 (u_2 - 2\lambda u_3 + \lambda^2 u_4).
\end{aligned}
$$

*Proof.* This follows by specializing Theorem 3.2 to $\Sigma = I_p$ and isotropic $\beta$. $\qquad\square$

For an illustration of the empirical versus theoretical risks in Lemma E.4, see the second panel of Figure 13.

### E.2.3. FRESH-$X$ (AFFINE) SD UNDER ISOTROPIC SETTING

Throughout, we work in the balanced proportional regime $m, n, p \to \infty$ with $p/m \to \gamma$ and $p/n \to \gamma$, and under the isotropic assumptions $\Sigma = I_p$ and $\beta \sim \mathcal{N}(0, (r^2/p)I_p)$ from Section E.2. Recall the companion Stieltjes transform $v(\lambda)$ and $\kappa(\lambda) = 1/v(\lambda)$ from Section E.2.2, as well as $b(\lambda) = \kappa'(\lambda) = (1 - t_2(\lambda))^{-1}$.

Let $\widetilde{\Sigma} := \widetilde{X}^\top \widetilde{X}/m$ and define the (fresh) shrinkage matrix $M_\lambda := \widetilde{\Sigma}(\widetilde{\Sigma} + \lambda I_p)^{-1}$. Let

$$s(\lambda) := \lim_{m,p \to \infty} \frac{1}{p} \mathrm{tr}(M_\lambda), \qquad s_2(\lambda) := \lim_{m,p \to \infty} \frac{1}{p} \mathrm{tr}(M_\lambda^2).$$

Writing $\widetilde{S} := \widetilde{X}\widetilde{X}^\top/m$ and $\widetilde{G}_\lambda := (\widetilde{S} + \lambda I_m)^{-1}$, we have $\widetilde{S}(\widetilde{S} + \lambda I_m)^{-1} = I_m - \lambda \widetilde{G}_\lambda$, and therefore

$$\frac{1}{p} \mathrm{tr}(M_\lambda) = \frac{m}{p} \cdot \frac{1}{m} \mathrm{tr}\big(\widetilde{S}(\widetilde{S} + \lambda I_m)^{-1}\big) = \frac{1}{\gamma}\Big(1 - \lambda \frac{1}{m} \mathrm{tr}(\widetilde{G}_\lambda)\Big).$$

Since $p/m \to \gamma$, we have $\frac{1}{m}\operatorname{tr}(\widetilde{G}_\lambda) \to v(\lambda)$, where $v(\lambda)$ solves (80). Hence

$$s(\lambda) = \frac{1 - \lambda v(\lambda)}{\gamma} = \frac{1 - \lambda/\kappa(\lambda)}{\gamma}. \tag{82}$$

Similarly, using $M_\lambda = I_p - \lambda(\widetilde{\Sigma} + \lambda I_p)^{-1}$ and the identity $\frac{1}{m}\operatorname{tr}(\widetilde{G}_\lambda^2) = -v'(\lambda)$, we obtain

$$s_2(\lambda) = \frac{1 - 2\lambda v(\lambda) - \lambda^2 v'(\lambda)}{\gamma} = \frac{1 - 2\lambda v(\lambda) + \lambda^2 b(\lambda)v(\lambda)^2}{\gamma} = \frac{1 - 2\lambda/\kappa(\lambda) + \lambda^2 b(\lambda)/\kappa(\lambda)^2}{\gamma}. \tag{83}$$

**Lemma E.5** (Isotropic limits for fresh-$X$ (affine) SD)**.** *Fix $\lambda > 0$. Under Assumption 3.1 with $\Sigma = I_p$ and $\beta \sim \mathcal{N}(0, (r^2/p)I_p)$, as $m, n, p \to \infty$ with $p/m, p/n \to \gamma \in (0, \infty)$, the optimal fresh-$X$ SD risk satisfies*

$$R_{\mathsf{sd}}^{\star,\mathrm{fraff}}(\lambda) \overset{\mathrm{p}}{\to} \mathscr{R}_{\mathsf{sd}}^{\star,\mathrm{fraff}}(\lambda) := \mathscr{R}(\lambda) - \frac{\big(\mathscr{R}(\lambda) - \mathscr{C}^{\mathrm{fr}}(\lambda)\big)^2}{\mathscr{R}(\lambda) + \mathscr{R}_{\mathsf{pd}}^{\mathrm{fr}}(\lambda) - 2\mathscr{C}^{\mathrm{fr}}(\lambda)}.$$

*Here $\mathscr{R}(\lambda)$ is the same teacher risk limit from Lemma E.4, and $\mathscr{R}_{\mathsf{pd}}^{\mathrm{fr}}(\lambda)$ and $\mathscr{C}^{\mathrm{fr}}(\lambda)$ are the deterministic limits of the fresh-$X$ PD risk $R_{\mathsf{pd}}^{\mathrm{fr}}(\lambda)$ and the corresponding residual correlation $C^{\mathrm{fr}}(\lambda)$, given by:*

$$\mathscr{C}^{\mathrm{fr}}(\lambda) = \sigma^2 + s(\lambda)\big(\mathscr{R}(\lambda) - \sigma^2\big) + r^2\big(1 - s(\lambda)\big)^2,$$
$$\mathscr{R}_{\mathsf{pd}}^{\mathrm{fr}}(\lambda) = \sigma^2 + s_2(\lambda)\big(\mathscr{R}(\lambda) - \sigma^2\big) + r^2\big(1 - 2s(\lambda)^2 + (2s(\lambda) - 1)s_2(\lambda)\big).$$

*Proof.* Let $\beta_{\mathsf{pd},\lambda}^{\mathrm{fr}} := M_\lambda \beta_\lambda$ be the coefficient of the fresh-$X$ PD refit (trained on $(\widetilde{X}, \widetilde{y}_\lambda)$ with $\widetilde{y}_\lambda = \widetilde{X}\beta_\lambda$), and write $\Delta_\lambda := \beta_\lambda - \beta$, and $\Delta_{\mathsf{pd},\lambda}^{\mathrm{fr}} := \beta_{\mathsf{pd},\lambda}^{\mathrm{fr}} - \beta$. In the isotropic in-distribution setting, the teacher and PD risks and their residual correlation satisfy the identities:

$$R(\lambda) = \sigma^2 + \|\Delta_\lambda\|_2^2, \qquad R_{\mathsf{pd}}^{\mathrm{fr}}(\lambda) = \sigma^2 + \|\Delta_{\mathsf{pd},\lambda}^{\mathrm{fr}}\|_2^2, \qquad C^{\mathrm{fr}}(\lambda) = \sigma^2 + \langle \Delta_\lambda, \Delta_{\mathsf{pd},\lambda}^{\mathrm{fr}} \rangle.$$

Since $\Delta_{\mathsf{pd},\lambda}^{\mathrm{fr}} = M_\lambda \beta_\lambda - \beta = M_\lambda \Delta_\lambda + (M_\lambda - I_p)\beta$, we have

$$\langle \Delta_\lambda, \Delta_{\mathsf{pd},\lambda}^{\mathrm{fr}} \rangle = \Delta_\lambda^\top M_\lambda \Delta_\lambda + \Delta_\lambda^\top (M_\lambda - I_p)\beta, \tag{84}$$

and

$$\|\Delta_{\mathsf{pd},\lambda}^{\mathrm{fr}}\|_2^2 = \Delta_\lambda^\top M_\lambda^2 \Delta_\lambda + \beta^\top (M_\lambda - I_p)^2 \beta + 2\,\Delta_\lambda^\top M_\lambda (M_\lambda - I_p)\beta. \tag{85}$$

Thus, we need asymptotics for linear and quadratic forms involving $\Delta_\lambda$, which we obtain below.

*Linear and quadratic forms of $\Delta_\lambda$.* Write the ridge teacher error as

$$\Delta_\lambda = -\lambda(\widehat{\Sigma} + \lambda I_p)^{-1}\beta + (\widehat{\Sigma} + \lambda I_p)^{-1}\frac{X^\top \varepsilon}{n} =: \Delta_\lambda^{\mathrm{bias}} + \Delta_\lambda^{\mathrm{var}},$$

where $\widehat{\Sigma} = X^\top X/n$ and $\varepsilon := y - X\beta$. As in the proof of Theorem 3.2, the cross term $\langle \Delta_\lambda^{\mathrm{bias}}, \Delta_\lambda^{\mathrm{var}} \rangle$ is $o_p(1)$, and $\|\Delta_\lambda\|_2^2 = \|\Delta_\lambda^{\mathrm{bias}}\|_2^2 + \|\Delta_\lambda^{\mathrm{var}}\|_2^2 + o_p(1)$. Moreover, by Lemma E.4, $\|\Delta_\lambda\|_2^2 \overset{\mathrm{P}}{\to} \mathscr{R}(\lambda) - \sigma^2$.

Because $\widetilde{X}$ is independent of $(X, \beta, \varepsilon)$, the matrix $M_\lambda$ is independent of $(\Delta_\lambda, \beta)$ and is orthogonally invariant, with $\|M_\lambda\|_{\mathrm{op}} \le 1$. By standard concentration for quadratic forms of an independent orthogonally invariant matrix, similar to Lemma C.3 in the proof of Theorem 3.2, we have

$$\Delta_\lambda^\top M_\lambda \Delta_\lambda = \left(\frac{1}{p}\operatorname{tr}(M_\lambda)\right)\|\Delta_\lambda\|_2^2 + o_p(1), \tag{86}$$

$$\Delta_\lambda^\top M_\lambda^2 \Delta_\lambda = \left(\frac{1}{p}\operatorname{tr}(M_\lambda^2)\right)\|\Delta_\lambda\|_2^2 + o_p(1), \tag{87}$$

$$\Delta_\lambda^\top M_\lambda \beta = \left(\frac{1}{p}\operatorname{tr}(M_\lambda)\right)\langle \Delta_\lambda, \beta \rangle + o_p(1), \tag{88}$$

$$\Delta_\lambda^\top M_\lambda^2 \beta = \left(\frac{1}{p}\operatorname{tr}(M_\lambda^2)\right)\langle\Delta_\lambda, \beta\rangle + o_p(1), \tag{89}$$

and similarly,

$$\beta^\top (M_\lambda - I_p)^2 \beta = \left(\frac{1}{p}\operatorname{tr}\left((M_\lambda - I_p)^2\right)\right)\|\beta\|_2^2 + o_p(1). \tag{90}$$

Now, note that

$$\langle\Delta_\lambda, \beta\rangle = -\lambda\,\beta^\top(\widehat{\Sigma} + \lambda I_p)^{-1}\beta + \beta^\top(\widehat{\Sigma} + \lambda I_p)^{-1}\frac{X^\top\varepsilon}{n}.$$

The second term is mean-zero conditional on $(X, \beta)$ and is $o_p(1)$ by Lemma C.3. For the first term, isotropy of $\beta$ and standard quadratic-form concentration give

$$\beta^\top(\widehat{\Sigma} + \lambda I_p)^{-1}\beta = \frac{\|\beta\|_2^2}{p}\operatorname{tr}(\widehat{\Sigma} + \lambda I_p)^{-1} + o_p(1).$$

Using $\|\beta\|_2^2 \xrightarrow{\mathrm{P}} r^2$ and $\widehat{\Sigma}(\widehat{\Sigma} + \lambda I_p)^{-1} = I_p - \lambda(\widehat{\Sigma} + \lambda I_p)^{-1}$, we obtain

$$\lambda\cdot\frac{1}{p}\operatorname{tr}(\widehat{\Sigma} + \lambda I_p)^{-1} = 1 - \frac{1}{p}\operatorname{tr}\left(\widehat{\Sigma}(\widehat{\Sigma} + \lambda I_p)^{-1}\right).$$

In the isotropic proportional regime with $p/n \to \gamma$, the term $\frac{1}{p}\operatorname{tr}(\widehat{\Sigma}(\widehat{\Sigma} + \lambda I_p)^{-1})$ converges to the same deterministic limit as $\frac{1}{p}\operatorname{tr}(M_\lambda)$ (since $\widehat{\Sigma}$ and $\widetilde{\Sigma}$ are independent Wishart matrices with the same aspect ratio), namely $s(\lambda)$ from (82). Therefore,

$$\langle\Delta_\lambda, \beta\rangle \xrightarrow{\mathrm{P}} -r^2\left(1 - s(\lambda)\right). \tag{91}$$

We are now ready to obtain the asymptotics for the fresh-$X$ PD risk and residual correlation.

*Residual correlation asymptotics.* Combining (84) with (86) and (88) gives

$$\langle\Delta_\lambda, \Delta_{\mathsf{pd},\lambda}^{\mathrm{fr}}\rangle = \left(\frac{1}{p}\operatorname{tr}(M_\lambda)\right)\|\Delta_\lambda\|_2^2 + \left(\frac{1}{p}\operatorname{tr}(M_\lambda) - 1\right)\langle\Delta_\lambda, \beta\rangle + o_p(1).$$

Using $\frac{1}{p}\operatorname{tr}(M_\lambda) \to s(\lambda)$, $\|\Delta_\lambda\|_2^2 \xrightarrow{\mathrm{P}} \mathscr{R}(\lambda) - \sigma^2$, and (91), we obtain

$$\langle\Delta_\lambda, \Delta_{\mathsf{pd},\lambda}^{\mathrm{fr}}\rangle \xrightarrow{\mathrm{P}} s(\lambda)\left(\mathscr{R}(\lambda) - \sigma^2\right) + r^2\left(1 - s(\lambda)\right)^2.$$

Therefore,

$$C^{\mathrm{fr}}(\lambda) = \sigma^2 + \langle\Delta_\lambda, \Delta_{\mathsf{pd},\lambda}^{\mathrm{fr}}\rangle \xrightarrow{\mathrm{P}} \sigma^2 + s(\lambda)\left(\mathscr{R}(\lambda) - \sigma^2\right) + r^2\left(1 - s(\lambda)\right)^2 = \mathscr{C}^{\mathrm{fr}}(\lambda).$$

*PD risk asymptotics.* Similarly, combining (85) with (87), (90), and (89) yields:

$$\|\Delta_{\mathsf{pd},\lambda}^{\mathrm{fr}}\|_2^2 = \left(\frac{1}{p}\operatorname{tr}(M_\lambda^2)\right)\|\Delta_\lambda\|_2^2 + \left(\frac{1}{p}\operatorname{tr}((M_\lambda - I_p)^2)\right)\|\beta\|_2^2 + 2\left(\frac{1}{p}\operatorname{tr}(M_\lambda^2) - \frac{1}{p}\operatorname{tr}(M_\lambda)\right)\langle\Delta_\lambda, \beta\rangle + o_p(1).$$

Using $\frac{1}{p}\operatorname{tr}(M_\lambda^2) \to s_2(\lambda)$, $\frac{1}{p}\operatorname{tr}((M_\lambda - I_p)^2) \to 1 - 2s(\lambda) + s_2(\lambda)$, $\|\beta\|_2^2 \xrightarrow{\mathrm{P}} r^2$, $\|\Delta_\lambda\|_2^2 \xrightarrow{\mathrm{P}} \mathscr{R}(\lambda) - \sigma^2$, and (91), we get

$$\|\Delta_{\mathsf{pd},\lambda}^{\mathrm{fr}}\|_2^2 \xrightarrow{\mathrm{P}} s_2(\lambda)\left(\mathscr{R}(\lambda) - \sigma^2\right) + r^2\left(1 - 2s(\lambda)^2 + (2s(\lambda) - 1)s_2(\lambda)\right).$$

Hence

$$R_{\mathsf{pd}}^{\mathrm{fr}}(\lambda) = \sigma^2 + \|\Delta_{\mathsf{pd},\lambda}^{\mathrm{fr}}\|_2^2 \xrightarrow{\mathrm{P}} \sigma^2 + s_2(\lambda)\left(\mathscr{R}(\lambda) - \sigma^2\right) + r^2\left(1 - 2s(\lambda)^2 + (2s(\lambda) - 1)s_2(\lambda)\right) = \mathscr{R}_{\mathsf{pd}}^{\mathrm{fr}}(\lambda).$$

Finally, because $f_{\mathsf{sd},\lambda,\xi}^{\mathrm{fraff}} = (1 - \xi)f_\lambda + \xi f_{\mathsf{pd},\lambda}^{\mathrm{fr}}$ is a two-predictor affine path, combining the oracle formula in Proposition 2.1 with the component convergences proved above finishes the proof. $\qquad\square$

For an illustration of the empirical versus theoretical risks in Lemma E.5, see the second panel of Figure 13.

### E.2.4. PROOF OF THEOREM E.2

*Proof of Theorem E.2.* As before, let $v = v(\lambda) > 0$ denote the (unique) solution of (80), and define the signal-to-noise ratio $\text{SNR} := r^2/\sigma^2 > 0$. Set $D_0 := (1+v)^2 - \gamma v^2 = (1+v)^2(1 - t_2)$, where $t_2 = \gamma(v/(1+v))^2$ as in (81). Note that $D_0 > 0$. This is because differentiating (80) shows that

$$v'(\lambda) = -\frac{v(\lambda)^2}{1 - \gamma(v(\lambda)/(1+v(\lambda)))^2} = -\frac{v(\lambda)^2}{1 - t_2(\lambda)} < 0,$$

so $1 - t_2(\lambda) > 0$ and hence $D_0 > 0$ for all $\lambda > 0$.

Using the explicit isotropic expressions from Lemmas E.4 and E.5 and simplifying, one obtains the factorization

$$\mathscr{R}_{\mathsf{sd}}^{\star,\mathrm{fraff}}(\lambda) - \mathscr{R}_{\mathsf{sd}}^{\star,\mathrm{same}}(\lambda) = \sigma^2 \frac{\gamma v \, \Delta(v)^2 \, \Xi(v)}{(1+v)^3 \, D_0 \, \Pi_1(v) \, \Pi_2(v)}, \tag{92}$$

where

$$\Delta(v) := \gamma \, \text{SNR} \, v + \gamma v^2 + \gamma v - \text{SNR} \, v - \text{SNR}, \tag{93}$$

$$\Pi_1(v) := -\gamma \, \text{SNR} \, v + \gamma \, \text{SNR} + \gamma v + \gamma + \text{SNR} \, v + \text{SNR}, \tag{94}$$

$$\Pi_2(v) := \gamma^2 \, \text{SNR} \, v^2 + \gamma^2 v^2 - 2\gamma \, \text{SNR} \, v^2 - 2\gamma \, \text{SNR} \, v + \gamma \, \text{SNR}$$

$$+ \gamma v^2 + 2\gamma v + \gamma + \text{SNR} \, v^2 + 2 \, \text{SNR} \, v + \text{SNR}, \tag{95}$$

$$\Xi(v) := \gamma^2 v^3 + \text{SNR}\Big(\frac{(D_0 - 1)^2}{v} + D_0 \, v + v + 2\Big). \tag{96}$$

All factors on the right-hand side of (92) are nonnegative: $\sigma^2 > 0$, $\gamma > 0$, $v > 0$, and $\Delta(v)^2 \geq 0$. Moreover, $D_0 > 0$ as shown above, and $\Xi(v) > 0$ since it is a sum of strictly positive terms.

It remains to note that $\Pi_1(v)$ and $\Pi_2(v)$ are strictly positive. Indeed, in both the same-$X$ and fresh-$X$ settings, the two-predictor oracle formula Proposition 2.1 involves the discrepancy denominator

$$\mathscr{D}^\bullet(\lambda) := \mathscr{R}(\lambda) + \mathscr{R}_{\mathsf{pd}}^\bullet(\lambda) - 2\mathscr{C}^\bullet(\lambda) \qquad \bullet \in \{\mathrm{same}, \mathrm{fr}\},$$

which is strictly positive for every $\lambda > 0$ (the teacher and the corresponding PD refit do not coincide in this isotropic model). A direct identification shows that $\Pi_1(v)$ and $\Pi_2(v)$ are proportional to $\mathscr{D}^{\mathrm{fr}}(\lambda)$ and $\mathscr{D}^{\mathrm{same}}(\lambda)$, respectively, with positive proportionality constants; hence $\Pi_1(v), \Pi_2(v) > 0$.

Therefore the right-hand side of (92) is nonnegative for all $\lambda > 0$, showing the desired domination. $\square$

For a visual illustration of the risk asymptotics of the same-$X$ versus (affine) fresh-$X$ optimal SD risks, see the first panel of Figure 13. Consistent with Theorem E.2, the same-$X$ optimal SD risk uniformly dominates the fresh-$X$ optimal SD risk across $\lambda$.

## E.3. Results and Proofs in Section 5.3

This appendix formalizes the extension discussion in Section 5.3. The structural results of Section 2 extend beyond ordinary ridge regression to a class of (ridge) resolvent-based smoothers. At a high level, two properties drive the extension: (i) the teacher is a *linear smoother* in the labels, and (ii) the student refit applies the same *resolvent-based* smoothing family (at the same regularization level $\lambda$). Under (i), the SD family is again an affine path in the mixing weight $\xi$, so the two-predictor identities (e.g., Proposition 2.1) apply under any squared prediction risk. Under (ii), the teacher–PD gap admits a (surprising) derivative representation, resulting in strict improvement and sign rule properties analogous to Theorem 2.2. We state the generic result first, then highlight two representative examples.

Fix training inputs $X = (x_1^\top, \ldots, x_n^\top)^\top$ and labels $y \in \mathbb{R}^n$. Let $\{f_\lambda\}_{\lambda>0}$ be a teacher predictor family such that for each $\lambda$ there exists a vector-valued map $s_\lambda(\cdot) \in \mathbb{R}^n$ (depending on $X, \lambda$ but not on $y$) satisfying $f_\lambda(x) = s_\lambda(x)^\top y$ and

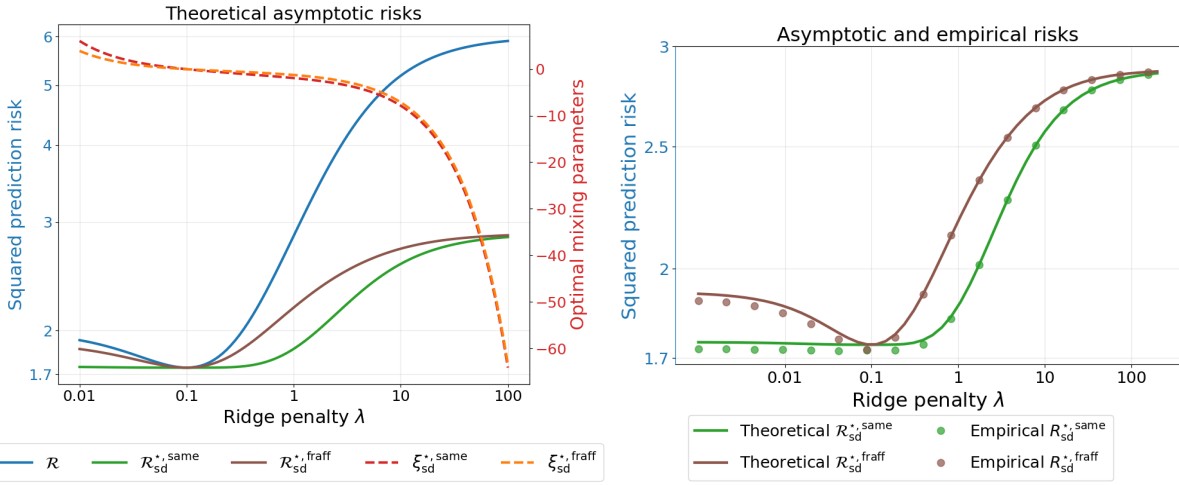

*Figure 13.* **Same-$X$ versus fresh-$X$ self-distillation risks.** Theoretical asymptotic (Theorem E.2) and empirical risks (averaged over 20 simulations) and optimal mixing weights for an isotropic setting with $n = 400$, $p = 200$, $r^2 = 5$, and $\sigma^2 = 1$.

$\widehat{y}_\lambda := f_\lambda(X) = S_\lambda y$ for some (smoothing) matrix $S_\lambda \in \mathbb{R}^{n \times n}$ depending on $X, \lambda$ but not on $y$. Define the (same-$\lambda$) PD student refit by reapplying the same smoother to the teacher fitted values: $f_{\mathsf{pd},\lambda}(x) := s_\lambda(x)^\top \widehat{y}_\lambda$. Define the SD predictor by the same mixed-label construction as in (2) (now with teacher pseudo-labels $\widehat{y}_\lambda$), which is equivalent to applying the smoother to the mixed labels $y(\xi) := (1 - \xi)y + \xi\widehat{y}_\lambda$. Hence, the SD predictor can be expressed as:

$$f_{\mathsf{sd},\lambda,\xi}(x) = s_\lambda(x)^\top y(\xi) = (1 - \xi)f_\lambda(x) + \xi f_{\mathsf{pd},\lambda}(x). \tag{97}$$

*Proof of Equation* (97). Recall that $f_\lambda(x) = s_\lambda(x)^\top y$ and $\widehat{y}_\lambda = S_\lambda y$, and that the SD refit applies the same smoother to the mixed labels $y^{(\xi)} := (1 - \xi)y + \xi\widehat{y}_\lambda$. Then, for every $x$,

$$f_{\mathsf{sd},\lambda,\xi}(x) = s_\lambda(x)^\top y^{(\xi)} = (1 - \xi)s_\lambda(x)^\top y + \xi s_\lambda(x)^\top \widehat{y}_\lambda = (1 - \xi)f_\lambda(x) + \xi f_{\mathsf{pd},\lambda}(x),$$

where $f_{\mathsf{pd},\lambda}(x) := s_\lambda(x)^\top \widehat{y}_\lambda$ by definition. □

For any squared prediction risk $R(\cdot)$ as in (4), write $R(\lambda) := R(f_\lambda)$, $R_{\mathsf{pd}}(\lambda) := R(f_{\mathsf{pd},\lambda})$, and define:

$$C(\lambda) := \mathbb{E}\big[(y_0 - f_\lambda(x_0))(y_0 - f_{\mathsf{pd},\lambda}(x_0)) \mid \mathcal{D}\big], \quad D(\lambda) := \mathbb{E}\big[(f_\lambda(x_0) - f_{\mathsf{pd},\lambda}(x_0))^2 \mid \mathcal{D}\big].$$

(As in Section 2, it is easy to check that $D(\lambda) = R(\lambda) + R_{\mathsf{pd}}(\lambda) - 2C(\lambda)$.) Assume in addition that the ridge-smoother family satisfies the derivative (tangent) identity

$$f_\lambda(x) - f_{\mathsf{pd},\lambda}(x) = -\lambda\,\partial_\lambda f_\lambda(x) \qquad \text{for all } x, \tag{98}$$

and that $\lambda \mapsto R(\lambda)$ is differentiable. Then the derivative-based characterization from Theorem 2.2 extends as follows.

**Theorem E.6** (Ridge-smoother strict improvement and sign rule). *Fix $\lambda > 0$ and assume $D(\lambda) > 0$. Under* (98)*, the optimal SD mixing weight and risk along the regularization path satisfy*

$$\xi^\star(\lambda) = -\frac{\lambda}{2}\frac{R'(\lambda)}{D(\lambda)}, \qquad R_{\mathsf{sd}}^\star(\lambda) = R(\lambda) - \frac{\lambda^2}{4}\frac{(R'(\lambda))^2}{D(\lambda)}.$$

*In particular, if $R'(\lambda) \neq 0$, then $R_{\mathsf{sd}}^\star(\lambda) < R(\lambda)$ and $\operatorname{sign}(\xi^\star(\lambda)) = -\operatorname{sign}(R'(\lambda))$.*

*Proof of Theorem E.6.* Fix $\lambda > 0$. By (97), the SD family is the affine path $f_{\mathsf{sd},\lambda,\xi} = (1 - \xi)f_\lambda + \xi f_{\mathsf{pd},\lambda}$. Let $e_\lambda := y_0 - f_\lambda(x_0)$ and $e_{\mathsf{pd}} := y_0 - f_{\mathsf{pd},\lambda}(x_0)$. Expanding the square gives, for every $\xi \in \mathbb{R}$,

$$R_{\mathsf{sd}}(\lambda, \xi) = \mathbb{E}\Big[\big((1 - \xi)e_\lambda + \xi e_{\mathsf{pd}}\big)^2 \,\Big|\, \mathcal{D}\Big]$$

$$= (1 - \xi)^2 R(\lambda) + \xi^2 R_{\mathsf{pd}}(\lambda) + 2\xi(1 - \xi)C(\lambda)$$
$$= R(\lambda) - 2\xi\big(R(\lambda) - C(\lambda)\big) + \xi^2 D(\lambda),$$

where $D(\lambda) = R(\lambda) + R_{\mathsf{pd}}(\lambda) - 2C(\lambda)$. Under $D(\lambda) > 0$, this is a strictly convex quadratic in $\xi$, hence the unique minimizer is

$$\xi^\star(\lambda) = \frac{R(\lambda) - C(\lambda)}{D(\lambda)}, \qquad R_{\mathsf{sd}}^\star(\lambda) = R(\lambda) - \frac{(R(\lambda) - C(\lambda))^2}{D(\lambda)}.$$

Next, assume (98) holds. Since $\lambda \mapsto R(\lambda)$ is differentiable, we may differentiate inside the conditional expectation to get

$$R'(\lambda) = \partial_\lambda \mathbb{E}[e_\lambda^2 \mid \mathcal{D}] = \mathbb{E}\big[2e_\lambda \, \partial_\lambda e_\lambda \mid \mathcal{D}\big] = -2\,\mathbb{E}\big[e_\lambda \, \partial_\lambda f_\lambda(x_0) \mid \mathcal{D}\big].$$

On the other hand,

$$R(\lambda) - C(\lambda) = \mathbb{E}\big[e_\lambda(e_\lambda - e_{\mathsf{pd}}) \mid \mathcal{D}\big] = \mathbb{E}\big[e_\lambda \, (f_{\mathsf{pd},\lambda}(x_0) - f_\lambda(x_0)) \mid \mathcal{D}\big]$$
$$= -\mathbb{E}\big[e_\lambda \, (f_\lambda(x_0) - f_{\mathsf{pd},\lambda}(x_0)) \mid \mathcal{D}\big] = -\mathbb{E}\big[e_\lambda \, (-\lambda \, \partial_\lambda f_\lambda(x_0)) \mid \mathcal{D}\big]$$
$$= \lambda \, \mathbb{E}\big[e_\lambda \, \partial_\lambda f_\lambda(x_0) \mid \mathcal{D}\big] = -\frac{\lambda}{2} \, R'(\lambda).$$

Substituting $R(\lambda) - C(\lambda) = -(\lambda/2)R'(\lambda)$ into the closed forms above yields

$$\xi^\star(\lambda) = -\frac{\lambda}{2} \frac{R'(\lambda)}{D(\lambda)}, \qquad R_{\mathsf{sd}}^\star(\lambda) = R(\lambda) - \frac{\lambda^2}{4} \frac{(R'(\lambda))^2}{D(\lambda)}.$$

Finally, if $R'(\lambda) \neq 0$ then $R_{\mathsf{sd}}^\star(\lambda) < R(\lambda)$ and $\mathrm{sign}(\xi^\star(\lambda)) = -\mathrm{sign}(R'(\lambda))$. $\qquad\square$

The derivative identity (98) surprisingly holds for many resolvent-based ridge estimators, including generalized or Tikhonov ridge (Lemma E.7) and kernel ridge regression (Lemma E.8):

- *Generalized ridge.* For a fixed penalty operator $\Omega \succ 0$ (independent of $y$), generalized ridge is a linear smoother with

$$s_\lambda^\Omega(x) = X \, (X^\top X + n\lambda\,\Omega)^{-1} x, \qquad S_\lambda^\Omega = X \, (X^\top X + n\lambda\,\Omega)^{-1} X^\top.$$

  This includes feature-wise shrinkage (diagonal $\Omega$), graph or Laplacian regularization, spline-type penalties, and other quadratic Tikhonov penalties.
- *Kernel ridge.* For a PSD kernel with kernel matrix $K \in \mathbb{R}^{n \times n}$ and $k_x := (k(x, x_1), \ldots, k(x, x_n))^\top$, kernel ridge is a linear smoother with
$$s_\lambda^{\mathrm{kern}}(x) = (K + n\lambda I_n)^{-1} k_x, \qquad S_\lambda^{\mathrm{kern}} = K \, (K + n\lambda I_n)^{-1}.$$

  This covers standard kernel ridge estimators (e.g., Gaussian and polynomial kernels) and their variants obtained by changing $k$. See Figure 9 for empirical illustrations with a Gaussian kernel.

Both examples can be shown to satisfy (98) (see Section E.3.1), hence inherit Theorem E.6 under any squared prediction risk (including out-of-distribution).

### E.3.1. VERIFYING DERIVATIVE PROPERTY (98) FOR COMMON RIDGE VARIANTS

**Lemma E.7** (Generalized ridge satisfies (98))**.** *Fix $\Omega \succ 0$ and $\lambda > 0$, and let*

$$f_\lambda^\Omega(x) := x^\top (X^\top X + n\lambda\,\Omega)^{-1} X^\top y.$$

*Let $f_{\mathsf{pd},\lambda}^\Omega$ denote the PD refit obtained by training generalized ridge at the same $(X, \Omega, \lambda)$ on pseudo-labels $\widehat{y}_\lambda^\Omega = f_\lambda^\Omega(X)$. Then, for all $x$,*

$$f_\lambda^\Omega(x) - f_{\mathsf{pd},\lambda}^\Omega(x) = -\lambda \, \partial_\lambda f_\lambda^\Omega(x).$$

*Proof.* Let $A_\lambda := X^\top X + n\lambda\,\Omega$. Then $f_\lambda^\Omega(x) = x^\top A_\lambda^{-1} X^\top y$ and $f_{\mathsf{pd},\lambda}^\Omega(x) = x^\top A_\lambda^{-1} X^\top X A_\lambda^{-1} X^\top y$. Hence

$$f_\lambda^\Omega(x) - f_{\mathsf{pd},\lambda}^\Omega(x) = x^\top A_\lambda^{-1}(A_\lambda - X^\top X)A_\lambda^{-1} X^\top y = n\lambda \, x^\top A_\lambda^{-1}\Omega A_\lambda^{-1} X^\top y.$$

Moreover, $\partial_\lambda A_\lambda^{-1} = -A_\lambda^{-1}(\partial_\lambda A_\lambda)A_\lambda^{-1} = -nA_\lambda^{-1}\Omega A_\lambda^{-1}$, so

$$\partial_\lambda f_\lambda^\Omega(x) = x^\top(\partial_\lambda A_\lambda^{-1})X^\top y = -n\,x^\top A_\lambda^{-1}\Omega A_\lambda^{-1}X^\top y,$$

and multiplying by $-\lambda$ gives the claim. $\qquad\square$

**Lemma E.8** (Kernel ridge satisfies (98))**.** *Fix $\lambda > 0$ and a PSD kernel with kernel matrix $K \in \mathbb{R}^{n \times n}$ and $k_x := (k(x, x_1), \ldots, k(x, x_n))^\top$. Let*

$$f_\lambda^{\mathrm{kern}}(x) := k_x^\top(K + n\lambda I_n)^{-1}y,$$

*and let $f_{\mathsf{pd},\lambda}^{\mathrm{kern}}$ denote the PD refit trained at the same $\lambda$ on pseudo-labels $\widehat{y}_\lambda^{\mathrm{kern}} = f_\lambda^{\mathrm{kern}}(X)$. Then, for all $x$,*

$$f_\lambda^{\mathrm{kern}}(x) - f_{\mathsf{pd},\lambda}^{\mathrm{kern}}(x) = -\lambda\,\partial_\lambda f_\lambda^{\mathrm{kern}}(x).$$

*Proof.* Let $B_\lambda := K + n\lambda I_n$. Then $f_\lambda^{\mathrm{kern}}(x) = k_x^\top B_\lambda^{-1}y$ and $f_{\mathsf{pd},\lambda}^{\mathrm{kern}}(x) = k_x^\top B_\lambda^{-1}KB_\lambda^{-1}y$. Thus

$$f_\lambda^{\mathrm{kern}}(x) - f_{\mathsf{pd},\lambda}^{\mathrm{kern}}(x) = k_x^\top B_\lambda^{-1}(B_\lambda - K)B_\lambda^{-1}y = n\lambda\,k_x^\top B_\lambda^{-2}y.$$

Since $\partial_\lambda B_\lambda^{-1} = -B_\lambda^{-1}(\partial_\lambda B_\lambda)B_\lambda^{-1} = -nB_\lambda^{-2}$, we have

$$\partial_\lambda f_\lambda^{\mathrm{kern}}(x) = k_x^\top(\partial_\lambda B_\lambda^{-1})y = -n\,k_x^\top B_\lambda^{-2}y,$$

and multiplying by $-\lambda$ yields the claim. $\qquad\square$

# F. Additional Numerical Illustrations

The source code for reproducing the results of this paper can be found at the following location: https://github.com/hhd357/optimal_self_distillation_ridge

## F.1. Additional Experiments on CIFAR datasets

### F.1.1. RESULT ON CIFAR100 DATASET USING PRE-TRAINED RESNET-34 FEATURES

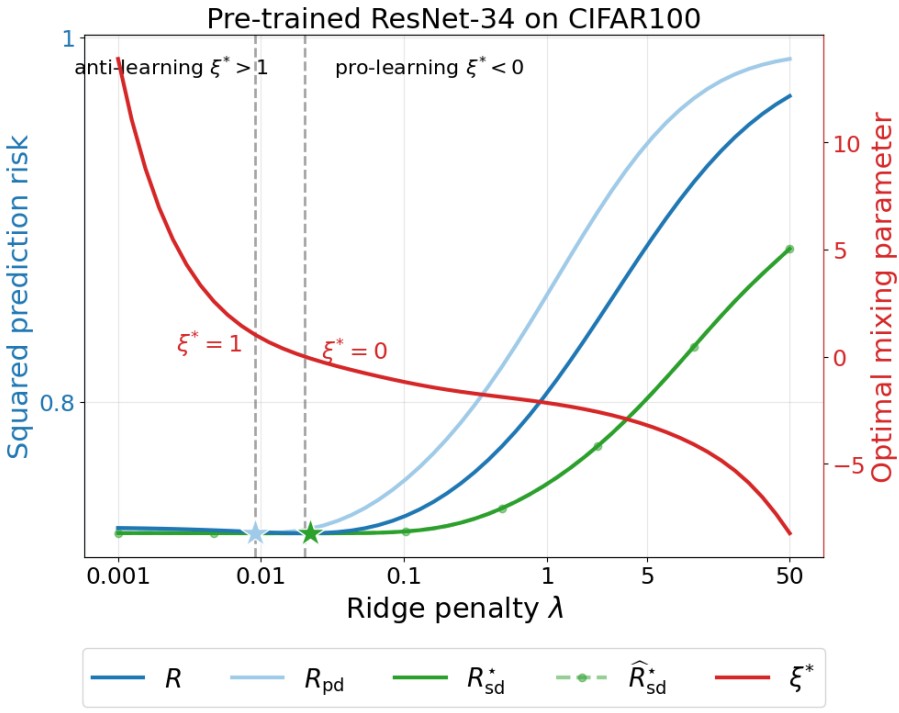

*Figure 14.* Squared prediction risks and optimal mixing parameter for ridge, pure-distilled and self-distilled ridge on pre-trained ResNet-34 features on CIFAR100 dataset.

### F.1.2. CIFAR10 AND CIFAR100 TEST ACCURACY OF THE RIDGE AND SELF-DISTILL RIDGE PREDICTORS

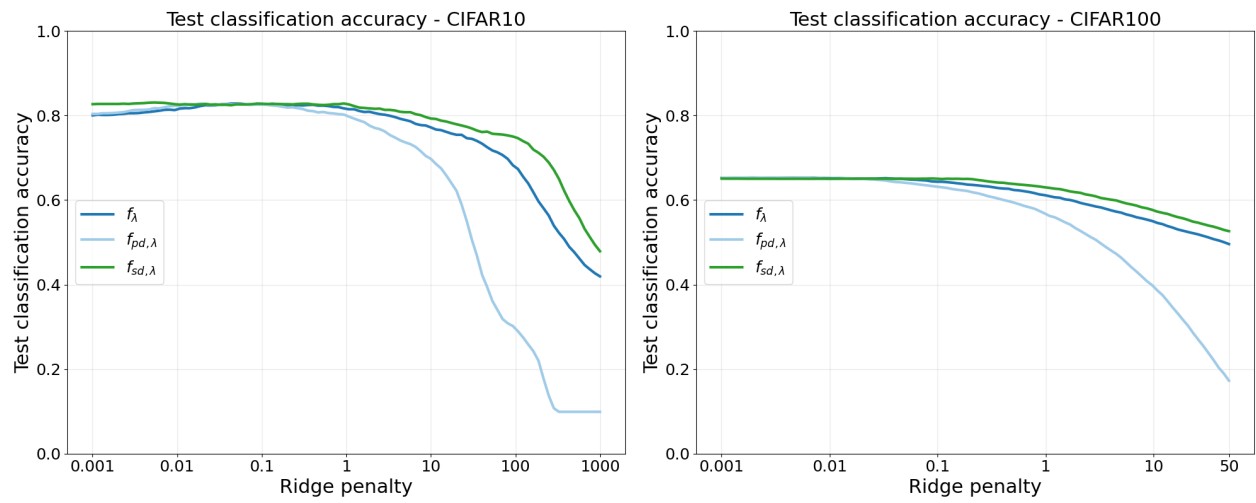

*Figure 15.* Test classification accuracy of the ridge and self-distilled ridges on CIFAR10 and CIFAR100 experiments. Although the optimal mixing is chosen to minimize the squared prediction risk, the strict improvement property still somewhat holds for test accuracy as well.

## F.2. Additional Illustrations Real-World Regression Tasks

### F.2.1. WITH CONSTRAINED $\xi \in [0, 1]$

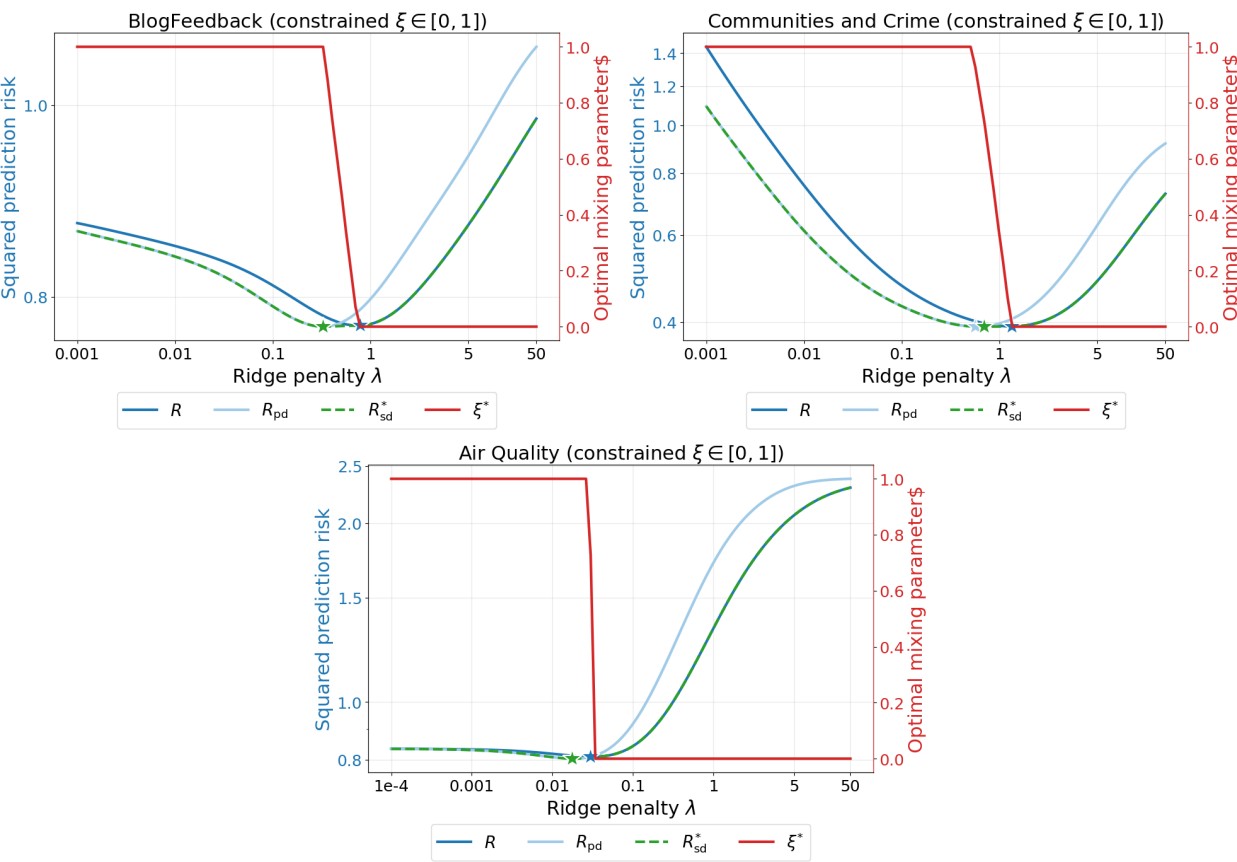

*Figure 16.* Test-set squared prediction risk and optimal mixing parameter, being constrained in $[0, 1]$. We choose the optimal $\xi$ over a grid of 100 values in $[0, 1]$ and pick the one with the lowest SD risk. The linestyle for the SD risk is changed only for this figure since it mostly overlaps with either the original ridge or the pure-distilled ridge curves. When $\xi = 1$, the SD risk exactly matches the PD risk and when $\xi = 0$, it exactly matches the teacher ridge risk.

## F.2.2. WITH DIFFERENT TRAIN/TEST SPLIT RATIOS AND GAINS

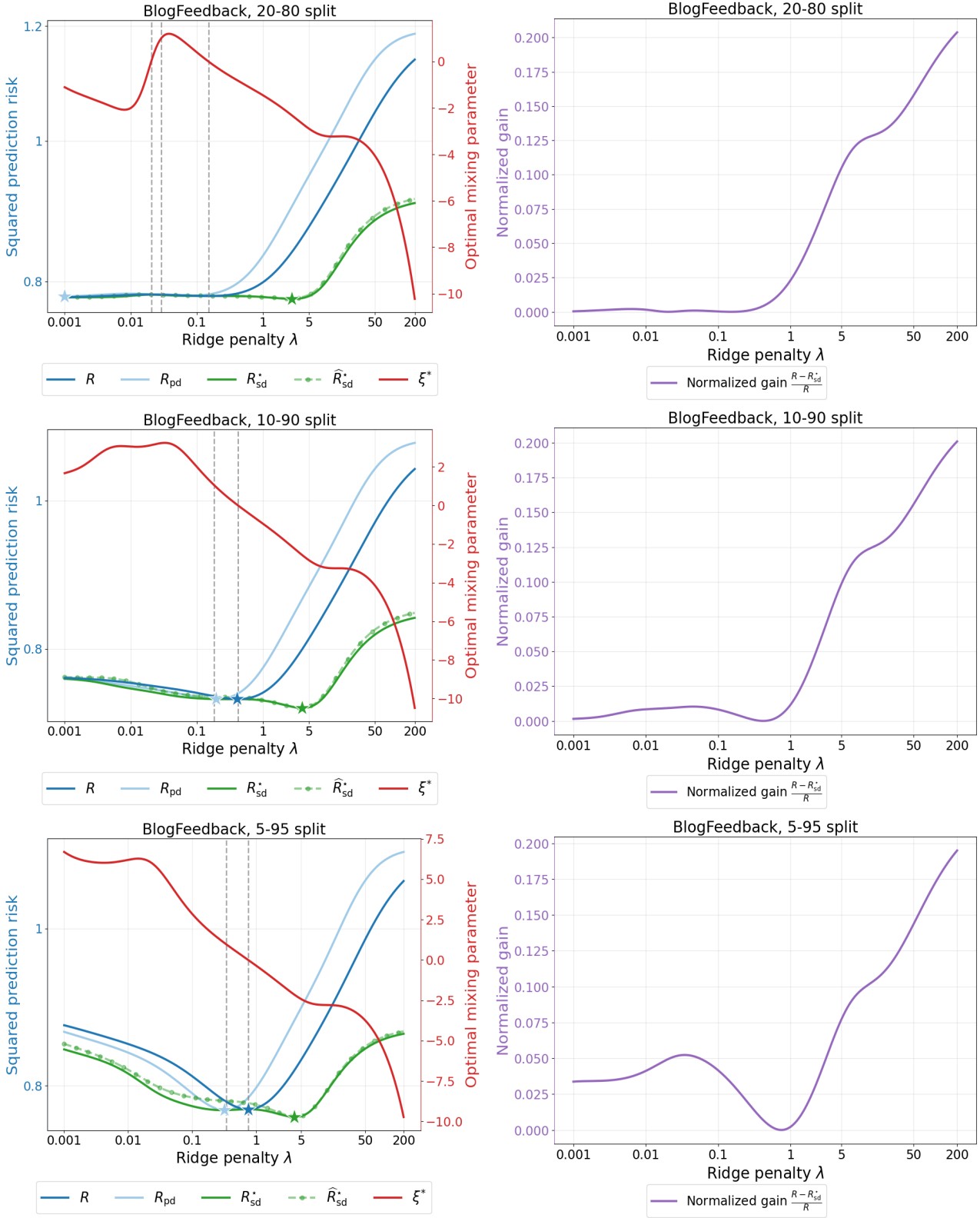

*Figure 17.* Squared prediction risks and gain curve on BlogFeedback dataset with different split ratios.

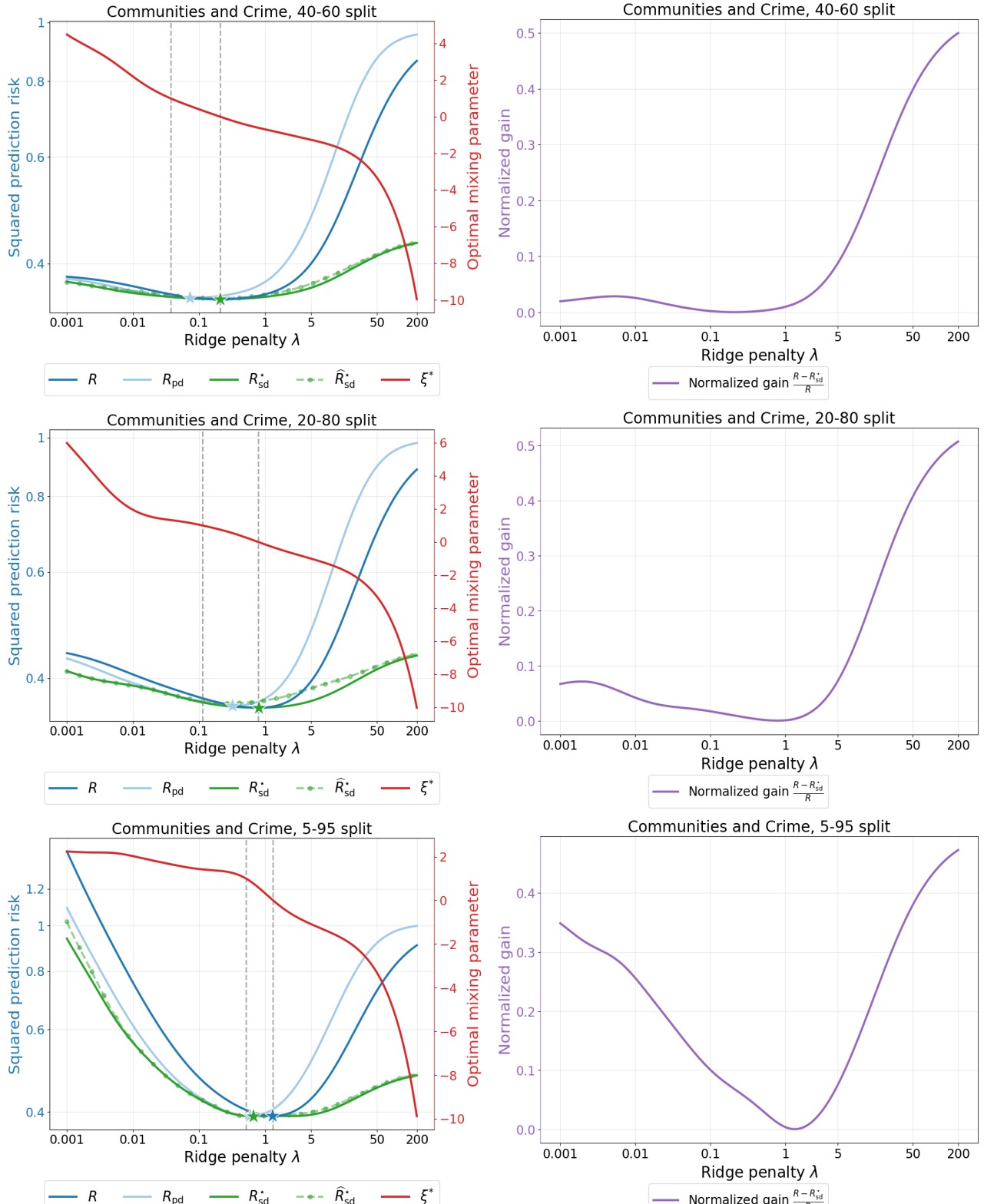

*Figure 18.* Squared prediction risks and gain curve on Communities and Crime dataset with different split ratios.

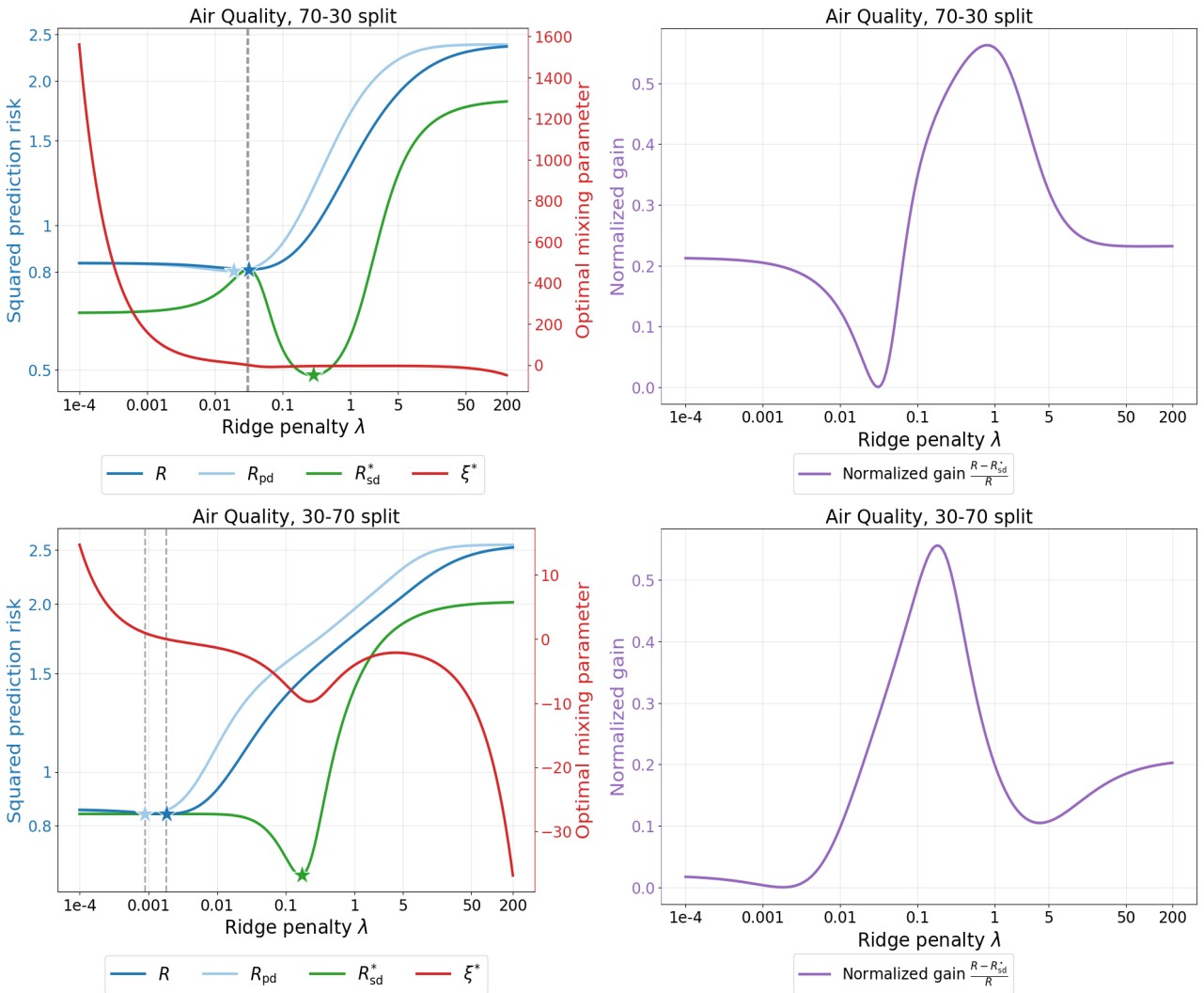

*Figure 19.* Squared prediction risks and gain curve on Air Quality dataset with different split ratios.

## F.3. Additional Illustrations on Proportional Asymptotic Risks

### F.3.1. WITH DIFFERENT SIGNAL-TO-NOISE RATIOS

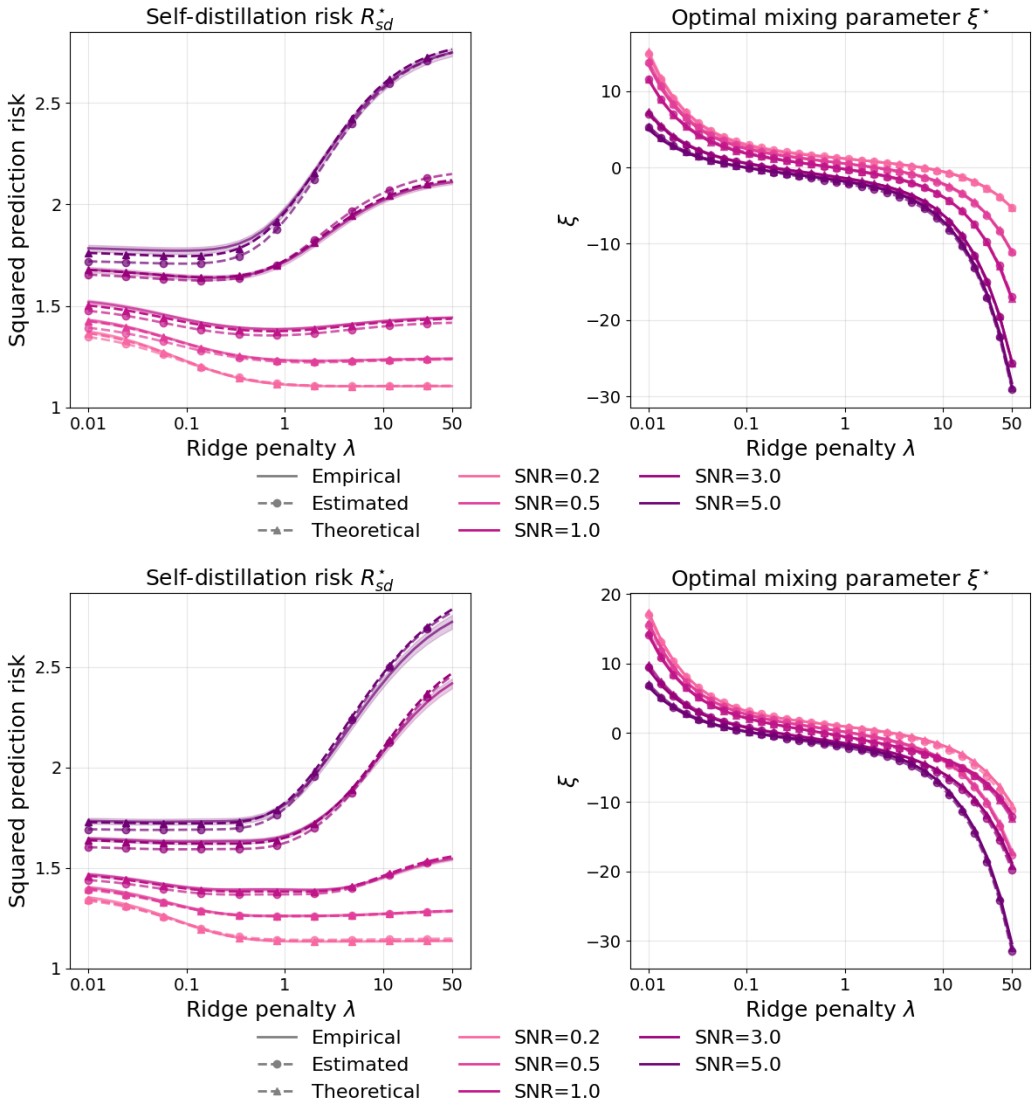

*Figure 20.* Asymptotic SD risk and optimal mixing parameter over various SNR ratios. Empirical curves are averaged over 30 numerical simulations, and the shaded band represents one standard deviation around the mean. The estimated are obtained using proposed tuning method (Section 4, averaged over 30 runs) and theoretical curves are from Theorem 3.2. $n = 400$, $p = 200$, $\sigma^2 = 1$ and $r^2 = \sigma^2$SNR. *Top row:* Data covariance $\Sigma$ is AR1, deterministic ground-truth signal is aligned with the bottom $10\%$ eigenvalues of $\Sigma$, with alignment factor 0.9. *Bottom row:* Data covariance $\Sigma$ is a spiked covariance matrix, deterministic ground-truth signal is aligned with the top $10\%$ eigenvalues of $\Sigma$, with alignment factor 0.9.

### F.3.2. WITH DIFFERENT ASPECT RATIOS

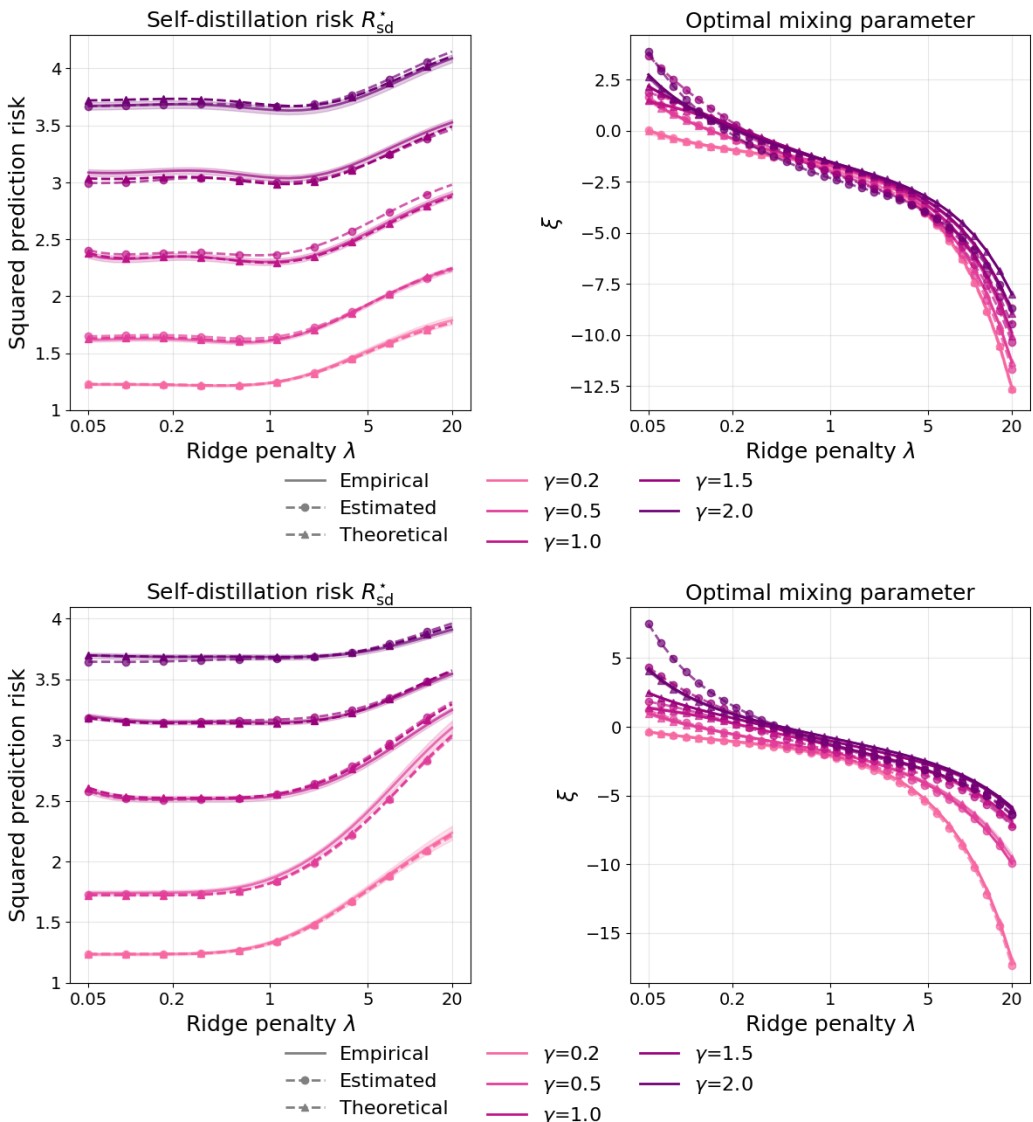

*Figure 21.* Asymptotic SD risk and optimal mixing parameter over different aspect ratios $\gamma = p/n$. Empirical curves are averaged over 30 numerical simulations, and the shaded band represents one standard deviation around the mean. The estimated are obtained using proposed tuning method (Section 4, averaged over 30 runs) and theoretical curves are from Theorem 3.2. $n = 300$, $p = n\gamma$, $\sigma^2 = 1$ and $r^2 = 5$. *Top row:* Data covariance $\Sigma$ is AR1, deterministic ground-truth signal is aligned with the top 10% eigenvalues of $\Sigma$, with alignment factor 0.9. *Bottom row:* Data covariance $\Sigma$ is spiked covariance matrix with $\Sigma = I + 5vv^\top$ with $v$ is a random isotropic Gaussian vector, deterministic ground-truth signal is aligned with the top 10% eigenvalues of $\Sigma$, with alignment factor 0.9.

F.3.3. MORE THEORETICAL RISK CURVES AND GAIN CURVES

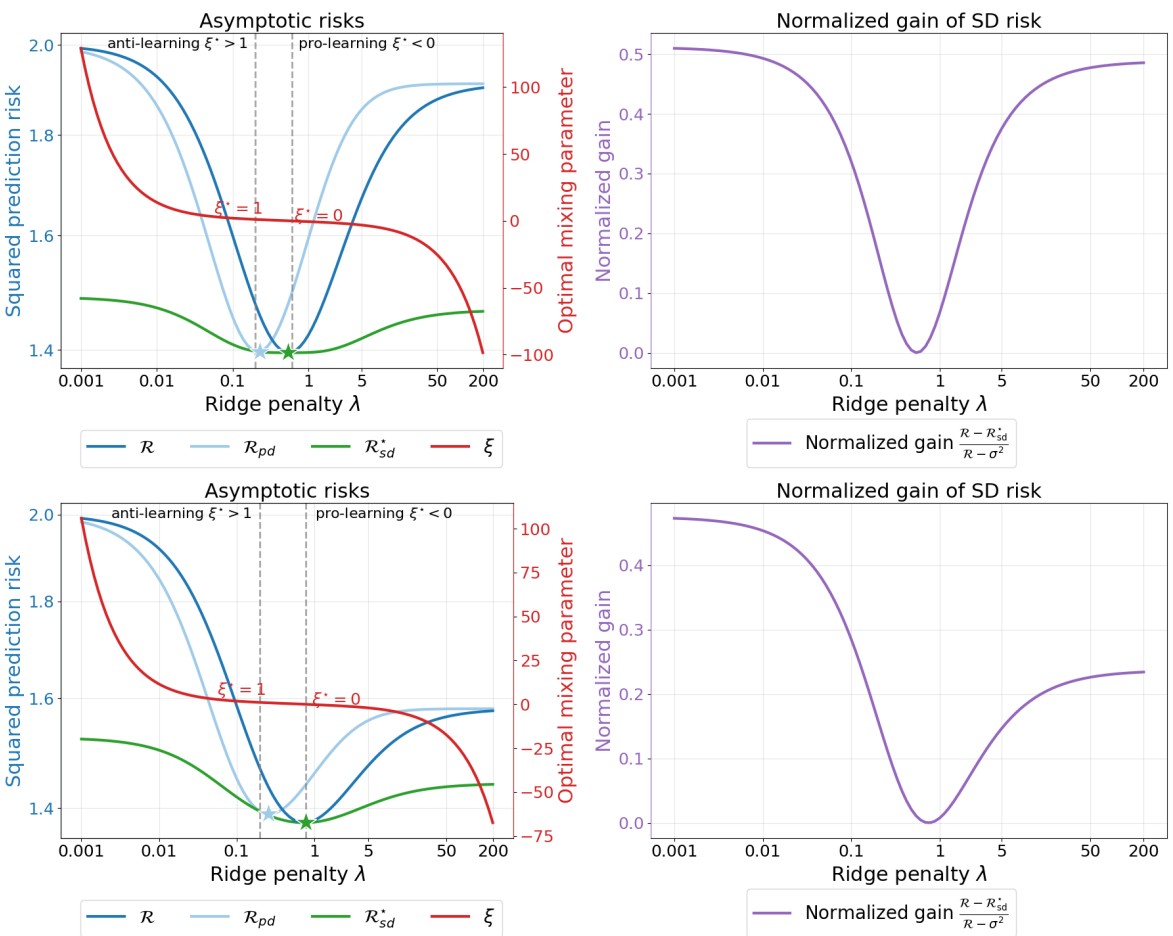

*Figure 22.* Asymptotic risk curves, asymptotic optimal mixing parameter and gain curves over the penalty $\lambda$. *Top row:* Data covariance $\Sigma$ is isotropic, isotropic signal, $n = 400, p = 200, r^2 = \sigma^2 = 1$. *Bottom row:* Data covariance $\Sigma$ is AR-1(0.25), deterministic ground-truth signal is aligned with the bottom 10% eigenvalues of $\Sigma$, with alignment factor 0.9, $n = 400, p = 200, r^2 = \sigma^2 = 1$.

## F.4. Additional Illustrations on Extreme Regularized Risks

### F.4.1. ISOTROPIC COVARIANCE, ISOTROPIC SIGNAL

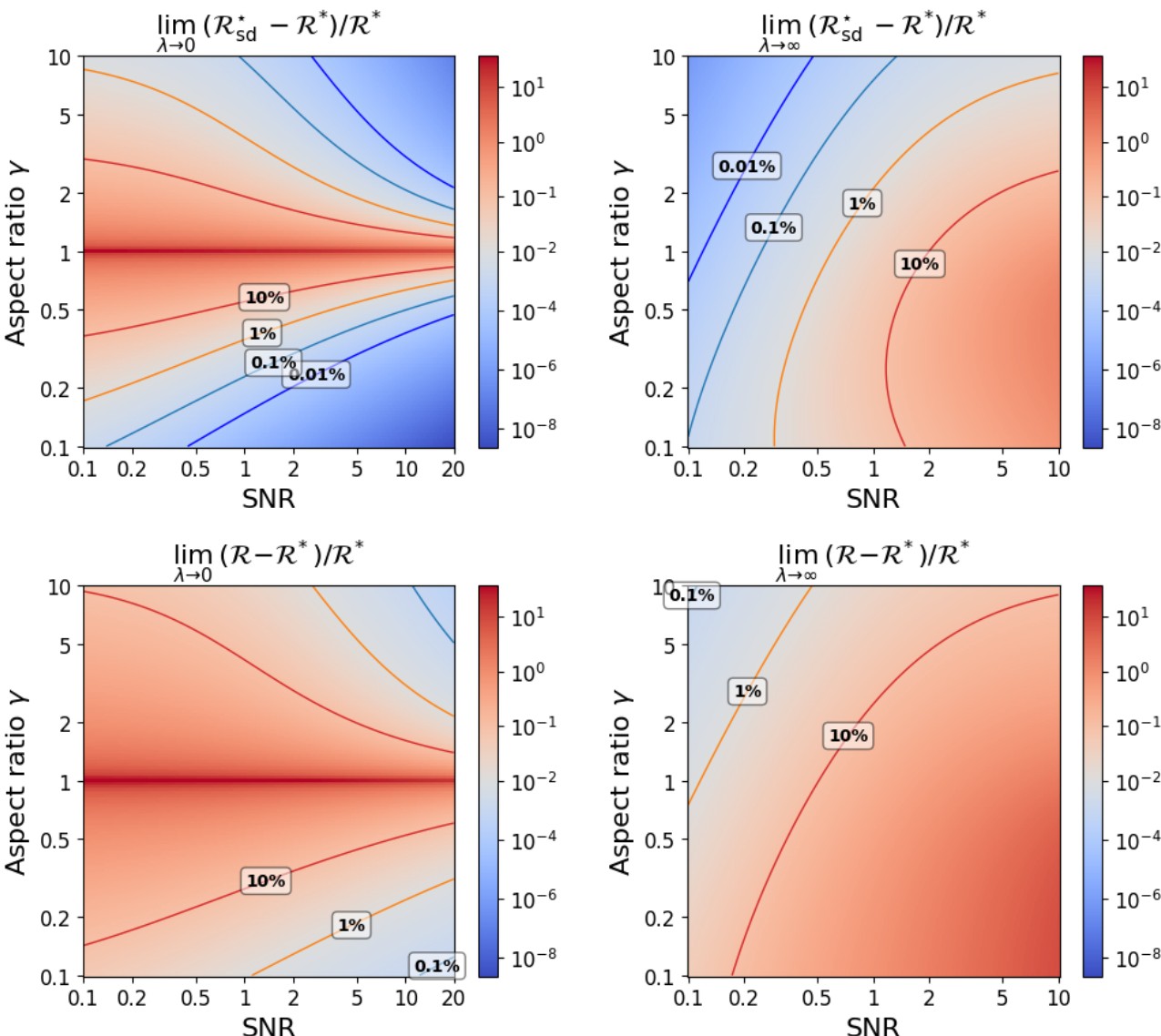

*Figure 23.* Percentage difference between $\mathscr{R}(\mathrm{SNR}, \gamma)$ and $\mathscr{R}_{sd}^{\star}(\mathrm{SNR}, \gamma)$ to the optimal ridge $\mathscr{R}^{\star}(\mathrm{SNR}, \gamma)$ (also the best predictor in this setting), and with isotropic design $\Sigma = I_p$, random signal follow an isotropic Gaussian distribution. The values plotted are calculated from Theorem 3.4.

## F.4.2. AR1 COVARIANCE, ISOTROPIC SIGNAL

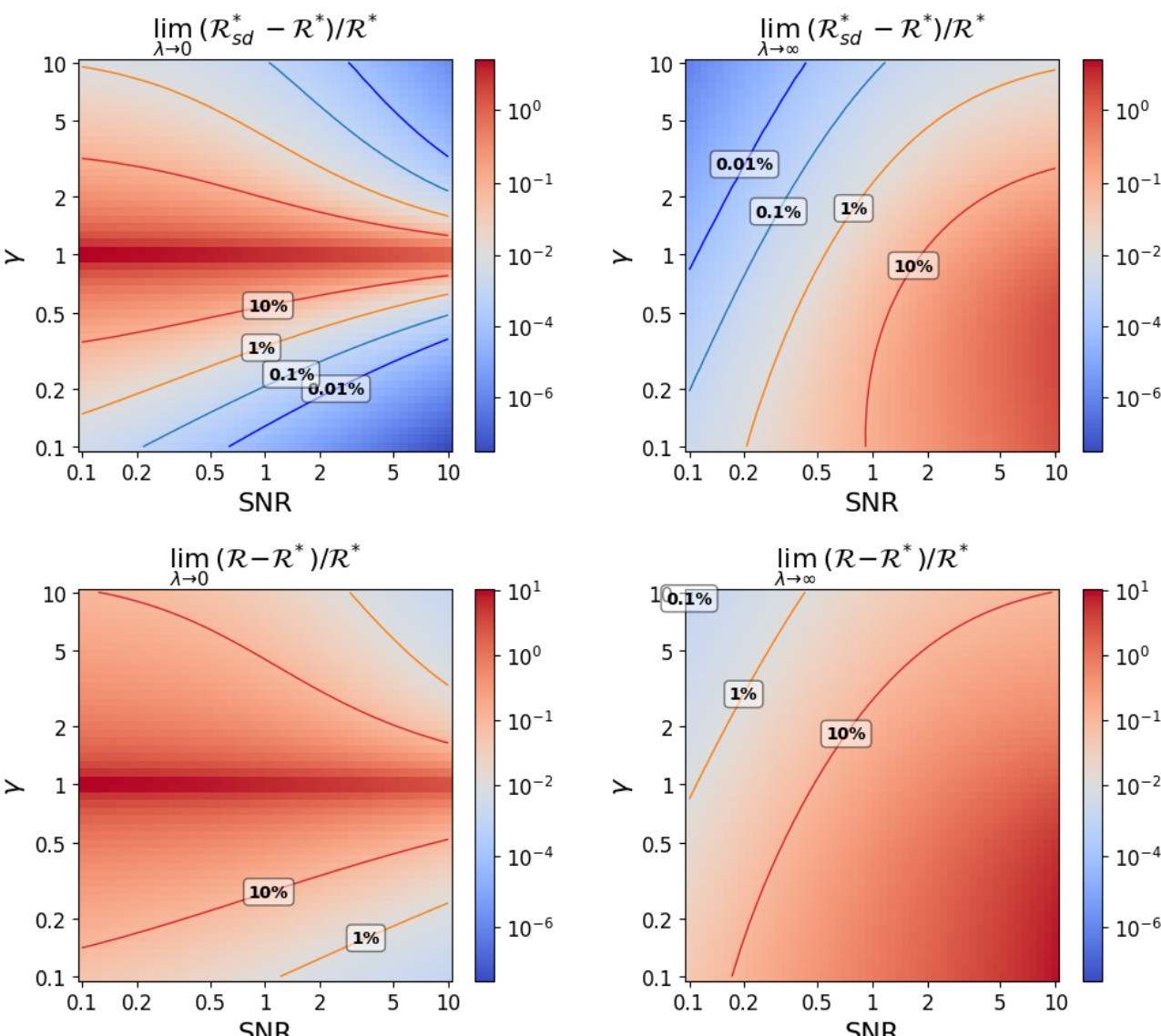

*Figure 24.* Percentage difference between $\mathcal{R}(\mathrm{SNR}, \gamma)$ and $\mathcal{R}_{sd}^{\star}(\mathrm{SNR}, \gamma)$ to the optimal ridge $\mathcal{R}^{\star}(\mathrm{SNR}, \gamma)$ (also the best predictor in this setting), with AR1 covariance design and random signal follows an isotropic Gaussian distribution. The ratios plotted are calculated using risk formulations at Theorem 3.2 at $\lambda = 10^{-3}$ and $\lambda = 10^6$.

### F.4.3. SPIKED COVARIANCE, ISOTROPIC SIGNAL

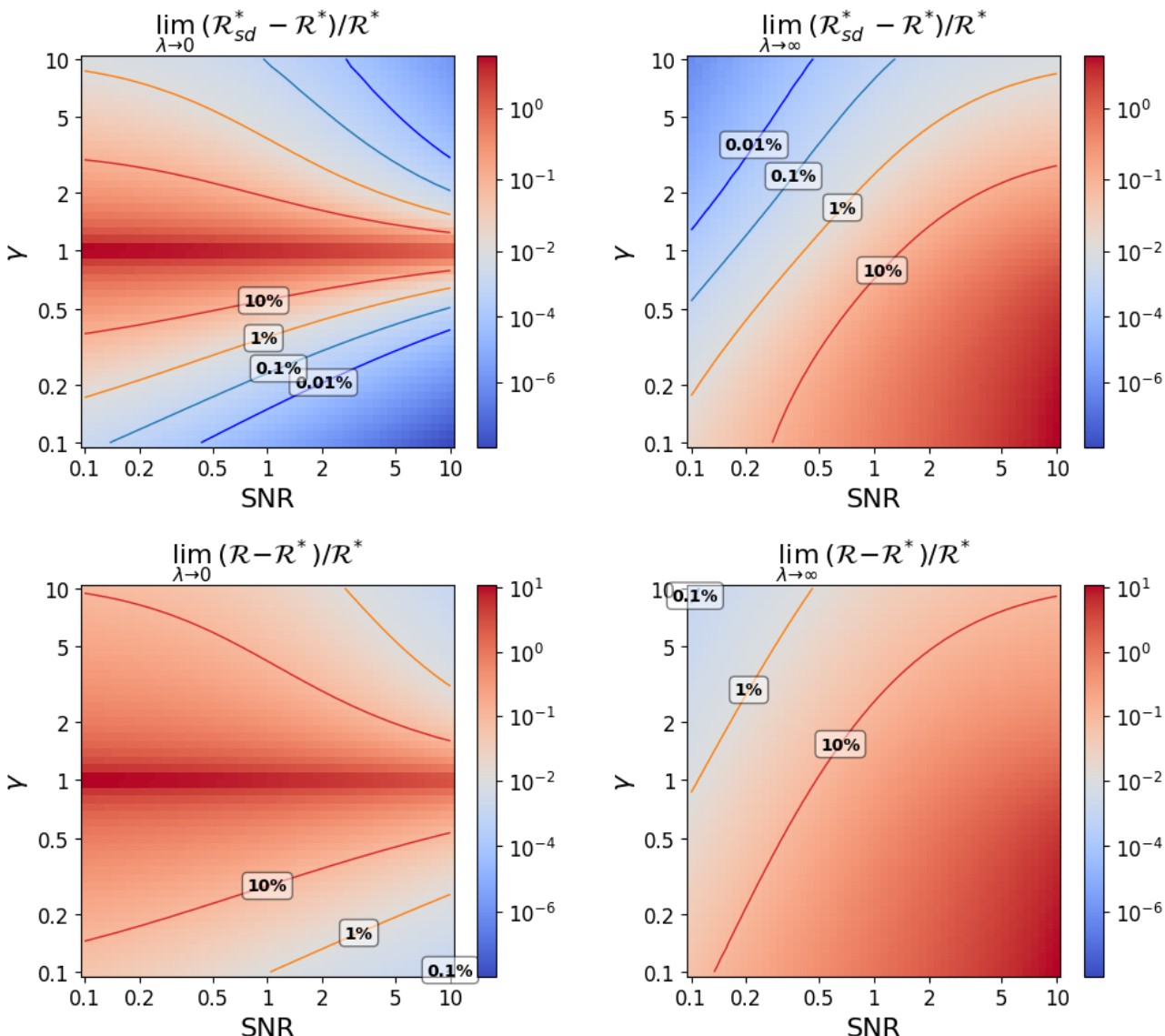

*Figure 25.* Percentage difference between $\mathcal{R}(\text{SNR}, \gamma)$ and $\mathcal{R}_{sd}^\star(\text{SNR}, \gamma)$ to the optimal ridge $\mathcal{R}^\star(\text{SNR}, \gamma)$ (also the best predictor in this setting), with spiked covariance design and random signal follow an isotropic Gaussian distribution. The ratios plotted are calculated using risk formulations at Theorem 3.2 at $\lambda = 10^{-3}$ and $\lambda = 10^6$.

## G. Experiment Details

### G.1. Real-World Regression Tasks: Datasets Description, Splitting and Pre-processing

For the experiments on real-world regression task, including UCI Blog Feedback (Buza, 2014), UCI Communities and Crime (Redmond, 2002), UCI Air Quality (Vito, 2008) datasets, we split the data into the training and test set (detailed below). Then, we process the data by removing records with missing values and we centered and standardized all the variables using the training set's mean and standard deviation. All the variables in the test set are also centered and standardized using the information from training set only.

All the risk curves shown at Figures 2 and 3 are the squared risk measured on the test set. The optimal mixing parameter is computed using the formula from Equation (6) where $R$, $R_{\mathsf{pd}}$ and $C$ are calculated using the test data.

**UCI Blogfeedback.** The UCI Blogfeedback dataset (Buza, 2014) originates from blog posts, where the raw HTML-documents of the blog posts were crawled and processed. We use only the training set of this dataset, which contains 52,397 samples and perform a random 5-95 split (5% for training and 95% for test set). The dataset consists of 280 features that capture many aspects of blog content and metadata, such as post length, number of links, number of comments in the first 24 hours after the publication of the blog post. The target variable is the number of comments in the next 24 hours (relative to base time). Thus, the training set in our experiment contains $p = 280$ covariates and $n = 2{,}619$ samples. The additional results of different split ratios are shown at Figure 17.

**UCI Communities and Crime.** The UCI Communities and Crime dataset (Redmond, 2002) are authentic data that combines socio-economic data from the 1990 US Census, law enforcement data from the 1990 US LEMAS survey, and crime data from the 1995 FBI UCR. The dataset consists of 1994 samples with 127 covariates. The goal is to predict the value of "ViolentCrimesPerPop", which is the rate of violent crimes per 100,000 population. For this dataset, we removed 5 non-predictive identifiers and 23 covariates from the LEMAS survey that contain 1,675 missing values out of 1,994 instances. The remaining 100 covariates have no missing values and used in our experiments. We perform a random 20-80 split (20% for training and 80% for test set). Thus, the training set in our experiment contains $p = 99$ covariates and $n = 398$ samples. The additional results of different split ratios are shown at Figure 18.

**UCI Air Quality.** The UCI Air Quality dataset (Vito, 2008) contains records of hourly averaged responses from an array of 5 metal oxide chemical sensors. Data were recorded at an Italy city from March 2004 to February 2005. Similar as (Pareek et al., 2024), we use $p = 8$ covariates for this task, which include 5 metal oxide chemical readings `PT08.S1(CO)`, `PT08.S2(NMHC)`, `PT08.S3(NOx)`, `PT08.S4(NO2)`, `PT08.S5(O3)` and 3 other covariates including Temperature `T`, Relative Humidity `RH`, and Absolute Humidity `AH`. The goal is to predict the Nitrogen Dioxide `NO2(GT)`. After removing missing records for these variables, we are left with $n = 7{,}393$ samples. We perform a *sequential* 70-30 split since the data is heavily time-dependent. Thus, the training set in our experiment contains $p = 8$ covariates and $n = 5{,}175$ samples. The additional results of different split ratios are shown at Figure 19.

For the kernel ridge regression experiment shown in Figure 9, we use a Gaussian kernel with bandwidth estimated as the median of the $\ell_2$ distances between covariates in the training set.

### G.2. CIFAR10 and CIFAR100 Experiments

For the experiment on CIFAR10 and CIFAR100 datasets, we extract the pre-trained ResNet-18 and ResNet-34 features (He et al., 2016), that trained on ImageNet dataset (available in Pytorch). For CIFAR10, we randomly sample 2,000 samples for training and 2,000 samples for the test set. For CIFAR100, we randomly sample 20,000 samples for training and 10,000 samples for the test set. Let $K$ is the number of classes. We then perform ridge regression on the last-layer features of the pretrained models. The predictor is now defined as the vector-valued function $f : \mathbb{R}^{512} \to \mathbb{R}^K$. To aggregate the risk, for an one-hot label vector $y \in \mathbb{R}^K$ where , we simply sum the mean squared error over the $K$ input dimensions,

$$R(f) = \mathbb{E}_{(x,y)} \left[ \sum_{k=1}^{K} (y_k - f(x)_k)^2 \right], \tag{99}$$

where $f_k : \mathbb{R}^{512} \to \mathbb{R}$ is a ridge predictor that predict the probability that the input belong to class $k$. The optimal mixing parameter is calculated from Equation (6)

## G.3. Synthetic Asymptotic Experiments

We give more details here about the data covariance $\Sigma$ and signal $\beta$ (defined in Assumption 3.1) used in the proportional asymptotic experiments in Sections 3, F.3 and F.4.

- Isotropic covariance: $\Sigma = I_p$
- AR1 covariance: $\rho$-autoregressive covariance with $\rho = 0.25$, $\Sigma_{ij} = \rho^{|i-j|}$ for all $i, j$.
- Spiked covariance: $\Sigma = I_p + 5vv^\top$ where $v \in \mathbb{R}^p$ is a random isotropic Gaussian vector.
- Isotropic signal: $\beta \sim \mathcal{N}(0, (r^2/p)I_p)$.
- Top-aligned signal with alignment ratio $m\%$ and alignment factor of $a$: let $k = \frac{m}{100} \cdot p$, then $\beta \sim \mathcal{N}(0, \Sigma_\beta)$ where $\Sigma_\beta = pV \operatorname{diag}(\frac{a}{k}, \frac{a}{k}, \ldots, \frac{a}{k}, \frac{1-a}{p-k}, \ldots, \frac{1-a}{p-k})V^\top$ where $V = [v_1, \ldots, v_p]$ contain the eigenvectors of $\Sigma$ with $v_k$ corresponds to $k$-th largest eigenvalue.
- Bottom-aligned signal with alignment ratio $m\%$ and alignment factor of $a$: let $k = \frac{m}{100} \cdot p$, then $\beta \sim \mathcal{N}(0, \Sigma_\beta)$ where $\Sigma_\beta = pV \operatorname{diag}(\frac{1-a}{p-k}, \ldots, \frac{1-a}{p-k}, \frac{a}{k}, \ldots, \frac{a}{k})V^\top$ where $V = [v_1, \ldots, v_p]$ contain the eigenvectors of $\Sigma$ with $v_k$ corresponds to $k$-th largest eigenvalue.

