# OpenReview forum: "Optimal Unconstrained Self-Distillation in Ridge Regression: Strict Improvements, Precise Asymptotics, and One-Shot Tuning"
_ICML.cc/2026/Conference — ICML 2026 regular_

### Official Review · Reviewer_6Yzx · 2026-02-25

**Soundness:** 4
**Presentation:** 4
**Significance:** 3
**Originality:** 3
**Overall Recommendation:** 5
**Confidence:** 3

**Summary:**

The authors study self-distillation, in the regularized case, and unconstrained setting for the self-distillation mixing weights. They first give non-asymptotic expressions, with no modeling assumption, for the optimal SD weights as a function of expected square risk, deducing therefrom some insightful structural results. Leveraging these results and prior ideas in GCV, they propose a one-shot estimate for the optimal weights. They then conduct an analysis in the proportional asymptotic regime, under weaker assumptions than prior works, and derive tight asymptotic characterizations for the risk and optimal SD weights. All theoretical claims are bolstered through numerical experiments on both synthetic and real data.

**Compliance With Llm Reviewing Policy:**

Affirmed.

**Final Justification:**

The rebuttal phase confirmed my first positive evaluation of this work, which makes a solid theoretical contribution to the study of self-distillation, and nicely complements existing works. The rebuttal addressed all my concerns and questions. I particularly appreciate the thoroughness of the analysis. I recommend acceptance of this manuscript.

**Key Questions For Authors:**

I have very few questions, most of which are not crucial to my evaluation.

- The authors missed some earlier works, in particular [a] (and possibly follow-up works), who similarly study a general setting which encompasses PD and SD, and logistic loss, and conjecture proportional asymptotics characterizations for the test error.

[a] Saglietti and Zdeborova, Solvable Model for Inheriting the Regularization through Knowledge Distillation, 2020

- Is there a particular reason the regularization for the SD training is the same as the one used to train the teacher? I understand it is required for the affine identity (2), but would the authors anticipate the analysis to go through also in the case where another value $\lambda^\prime$ is taken (I believe the asymptotic part would go through at least)? How would the structural picture change if $\lambda,\lambda^\prime,\xi$ are jointly optimized over ?

**Limitations:**

All assumptions are clearly stated.

**Strengths And Weaknesses:**

I overall have a positive evaluation of this work.

- Contributions : The results extends some prior results in many directions, summarized clearly and in detail in a "novelty" paragraph and Table 1. The GCV estimator for SD weights is to the best of my knowledge novel. Taken together, I believe these results make a good contribution in the theoretical analysis of SD, refining the understanding of it, and could be of interest to the community. This said, my own knowledge of this line of work is far from exhaustive, so it is not impossible there are some contributions I mis-evaluated. I appreciated the thoroughness of the work in addressing a breadth of points (e.g. the effect of data repetition versus fresh samples, comparison to related settings, to optimal ridge) that answered all questions I had during my first read.

- Presentation : The presentation is clear, and all claims are satisfyingly illustrated through numerical experiments, making the paper enjoyable to read. The supplementary material is also well-organized.

- Soundness : I have read several sections of the technical derivations, but did not carefully check all derivations in total detail. Overall, the claims seem sound, and are convincingly supported by numerical experiments.

I therefore would tend to support this paper for acceptation.

---

> ### Author Rebuttal · Authors · 2026-03-28
>
> Thank you for your positive and encouraging feedback! We are very glad to hear that you found our paper clear and enjoyable to read. We also appreciate the insightful questions you raised and have addressed them below.
>
> **Question [Q1]**
> > The authors missed some earlier works, in particular [a] (and possibly follow-up works), who similarly study a general setting which encompasses PD and SD, and logistic loss, and conjecture proportional asymptotics characterizations for the test error.
>
> Thank you for pointing us to the relevant reference [SZ20]. It contains several insightful qualitative observations and experiments on knowledge distillation in a solvable Gaussian-mixture classification model. We note that in this work, the mixing weight is constrained to lie in [0,1], thus, allowing an unconstrained mixing may further improve the student’s performance. The setting in which only half of the student’s parameters are trained to induce a weaker student is also interesting. We agree that this is all relevant context and we will cite and discuss it in the revision.
>
> **Question [Q2]**
> > Is there a particular reason the regularization for the SD training is the same as the one used to train the teacher? I understand it is required for the affine identity (2), but would the authors anticipate the analysis to go through also in the case where another value $\lambda^{'}$ is taken (I believe the asymptotic part would go through at least)? How would the structural picture change if $\lambda, \lambda^{'}, \xi$ are jointly optimized over ?
>
> Thank you for this great question! As a natural theoretical setting to learn about the SD benefit with optimal mixing weight between the hard labels and teacher's labels, we use the same $\lambda$ for both to emphasize the benefit of mixing weight $\xi$. If we use $\lambda$ for the teacher and a different $\lambda^{'}$ for the student, the affine identity (2) becomes
>
> $$f_{\mathrm{sd}, \lambda, \lambda^{'}} = (1 - \xi) f_{\lambda^{'}} + \xi f_{\mathrm{pd}, \lambda, \lambda^{'}},$$
>
> where $f_{\lambda^{'}}$ is ridge predictor on original data $(X, y)$ at $\lambda^{'}$ and $f_{\mathrm{pd}, \lambda, \lambda^{'}}$ is a ridge predictor trained at $\lambda^{'}$ on $(X, \hat{y}_{\lambda})$ (label from the teacher trained at $\lambda$). Therefore, we still have the quadratic form in $\xi$ for the squared risk and thus obtain the closed-form for the optimal mixing, analogous to our Theorem 2.2. We will add this as one of the variants in our Section 5 as it fits the paper theme well!
>
> You are correct that the asymptotic part will go through with different $\lambda$'s since, for example, our block linearization derivation can apply for different $\lambda$ (see Proposition C.10 in our work). Thank you for the excellent suggestion!
>
> To showcase the setting where a student can use a different $\lambda$, please see Section 7 in our anonymous site here: **https://sites.google.com/view/icmlsubmission22249**. We observe in the experiments that, when tuning over the student's $\lambda^{'}$ and do optimal mixing with $\xi^{\star}$, there are improvements over our optimal SD student (using the same $\lambda$), although the global minimum are not improved. On the other hand, the optimal-tuned $\lambda^{'}$ pure-distilled student (learn only from the teacher's labels) cannot outperform its sub-optimal teacher when $\lambda$ (of the teacher) is over-regularized. This further highlights the benefit of optimal-mixing self-distillation. This setting is a very interesting direction for future work!
>
> **References** Due to the space constraint, we kindly ask the reviewer to allow us to include all the references for all the reviewers here. Thank you!
>
> [SZ20] Saglietti and Zdeborova, Solvable Model for Inheriting the Regularization through Knowledge Distillation, 2020
>
> [DS23] Das and Sanghavi. Understanding Self-Distillation in the Presence of Label Noise. ICML 2023
>
> [PDO24] Pareek et al. Understanding the gains from repeated self-distillation. NeurIPS 2024
>
> [HB21] Hui and Belkin. Evaluation of Neural Architectures Trained with Square Loss vs Cross-Entropy in Classification Tasks. ICLR 2021
>
> [DCO20] Demirkaya et al. Exploring the Role of Loss Functions in Multiclass Classification. CISS 2020
>
> [WHS22] Wei et al. More than a toy: Random matrix models predict how real-world neural representations generalize. ICML 2022
>
> [DP24] Du and Patil. Implicit regularization paths of weighted neural representations. NeurIPS 2024
>
> [SFUB21] Stephenson et al. Can we globally optimize cross-validation loss?Quasiconvexity in ridge regression. NeurIPS 2021
>
> [L22] Cosme Louart. Sharp bounds for the concentration of the resolvent in convex concentration settings. arXiv:2201.00284, 2022
>
> [CM22] Chen Cheng and Andrea Montanari. Dimension free ridge regression. arXiv:2210.08571, 2022
>
> [RZM20] Rad et al. Error bounds in estimating the out-of-sample prediction error using leave-one-out cross validation in high-dimensions. AISTATS, 2020

---

> > ### Author Rebuttal · Reviewer_6Yzx · 2026-04-01
> >
> > I thank the authors for their detailed answer to my question, and the additional experiment. I wish to maintain my positive evaluation of this work.

---

> > > ### Author Response · Authors · 2026-04-06
> > >
> > > Thank you for your reply, we appreciate your endorsement. We would also like to sincerely thank the reviewer for their thoughtful review and valuable feedback.

---

### Official Review · Reviewer_FU7g · 2026-03-07

**Soundness:** 3
**Presentation:** 3
**Significance:** 3
**Originality:** 2
**Overall Recommendation:** 3
**Confidence:** 4

**Summary:**

This paper investigates optimal unconstrained self-distillation in ridge regression. The authors prove that optimally mixed student models strictly outperform the teacher under non-stationary regularization, with the optimal mixing parameter
  potentially being negative (“pro-learning”). They derive exact asymptotic formulas for the optimal SD risk under anisotropic covariance and deterministic signals in the proportional limit. Furthermore, the paper proposes a consistent “one-shot” tuning method that requires neither retraining nor data splitting. These theoretical findings are validated on real-world datasets and features from pre-trained neural networks.

**Compliance With Llm Reviewing Policy:**

Affirmed.

**Final Justification:**

The rebuttal addresses most of my concerns; nevertheless, the theoretical analysis does not go beyond the linear model. I have therefore increased my score to 3.

**Key Questions For Authors:**

See Weaknesses.

**Limitations:**

See weakness.

**Strengths And Weaknesses:**

**Strengths**

1. This presentation is well-organized. The theoretical results are sound.

2. The paper proposes an efficient one-shot tuning procedure that requires no retraining, no data splitting, and no grid search.

**Weaknesses**

1. While the mathematical derivations are clear, the analysis is confined to ridge regression. It remains unclear what insights these results offer when applied to more complex, non-linear models (e.g., deep neural networks). Therefore, the practical impact and generalizability of the work appear limited

2. While this submission highlights the benefits of allowing $\xi<0$ (labeled as 'pro-learning'), it would benefit from a more thorough discussion of related work. The phenomenon of negative optimal $\xi$ has been observed empirically in [1] (https://arxiv.org/abs/2407.04600, Table 1, 1-step self-distillation) and analyzed theoretically in [2] (https://openreview.net/forum?id=TfpcJt3LBV). Consequently, the paper's claim that  extending it to R yields new insights lacks sufficient novelty.

3. Furthermore, the paper's central theoretical claim—that the optimal $\xi$ corresponds to the extremum of a quadratic function—is not new. This mathematical characterization was already proposed in [2]. And the technical details are also similar to [2], both use  random matrix theory.

4. Proposition 3.4 relies on the assumption that the parameter vector follows an isotropic Gaussian prior. The paper does not discuss whether this assumption holds for real-world data. In practice, model parameters often exhibit sparsity, structure, or anisotropy, which are not captured by this i.i.d. Gaussian assumption.

5. The numerical simulations mainly focus on linear regression. Although experiments on CIFAR-10 are also conducted, they do not provide sufficient insights. It remains unclear what implications the conclusions of this paper have for the machine learning community.

---

> ### Author Rebuttal · Authors · 2026-03-29
>
> We sincerely thank the reviewer for the valuable feedback. Below we address your concerns. Based on the response and the valuable feedback from other reviewers, we kindly request the reviewer to reconsider their initial score.
>
> **Weakness [W1] and [W5]**
> > The analysis is confined to ridge regression. It remains unclear what insights offer when applied to more complex, non-linear models. The numerical simulations mainly focus on linear regression.
>
> Please kindly see our answer for **[W1]** from Reviewer $\texttt{TWSo}$.
>
> **Weakness [W2]**
> >  The phenomenon of negative optimal has been observed empirically in [1] and [2].
>
> We were aware that negative $\xi$ is not new; see the “$\xi$ range” column in Table 1 of our paper, where we wrote [DS23] and [PDO24] (your cited [1]) consider $\xi$ over the entire real line (lines 115–116). However, to our knowledge, it has not been theoretically analyzed: [DS23] neither discusses nor empirically studies it, and [PDO24] observes it only empirically. In contrast, we derive a sign rule for $\xi$ from the teacher risk curve (Theorem 2.2) and a clean asymptotic characterization under isotropic signals (Corollary 3.3), clarifying when $\xi$ is positive or negative. Moreover, our analysis applies to a general finite-sample setting with no data assumptions and any squared risk (including OOD).
>
> Regarding [2], we find that it was made public only on OpenReview (and nowhere else) in early February which is after the ICML deadline. Therefore, we were not be able to access to this work before our submission. It studies a complementary cross-domain KD setting. We will cite this work in our revision.
>
> More importantly, we find that our work and [2] have **very different contributions**, settings, and analyses, which we summarize below:
> * We contribute to the finite-sample setting (Section 2): i) a strict improvement property (Theorem 2.2), ii) a sign rule for the mixing weight (Theorem 2.2), and iii) a curvature test to detect global gain over the optimal-tuned teacher (Proposition 2.3). In contrast, [2] focuses on asymptotic risks only. Thus, we believe our key results are not studied there.
> * We propose a novel one-shot tuning method using GCV; tuning is not discussed in [2].
> * [2] assumes a well-specified linear model, which we do not assume.
> * [2] focuses on teacher and student trained on independent datasets, whereas we focus on the same-$X$ setting.
> * [2] derives asymptotic risk for arbitrary $\xi$, with some discussion of optimal $\xi$ (possibly negative). In contrast, we study optimal risk (finite-sample and asymptotic) with optimal mixing $\xi$, the gain over the teacher, and comparisons with the optimally tuned ridge teacher.
> * We cover multiple-round distillation, extension to kernel ridge and generalized ridge, and fresh-X analysis.
> * All experiments in [2] are with synthetic data. We include real-world datasets and pretrained network experiments.
>
> **Weakness [W3]**
> > The paper's central theoretical claim—that the optimal corresponds to the extremum of a quadratic function—is not new. And the technical details are also similar to [2], both use random matrix theory.
>
> Similar to the answer to [W2] above, we do not claim that the quadratic function in $\xi$ is novel, and it is **not our central theoretical contribution**. Please see the **Novelties paragraph** at the end of Section 1 of our paper for our main contributions.
>
> Comparison with [2] in terms of technical details. Both works use random matrix theory, but we characterize different objects with different techniques. Since we analyze the same-$X$ setting, the SD risk involves fourth-order deterministic equivalents, for which we use a block-linearization technique. In contrast, [2] primarily studies the case where the teacher and student are trained on independent datasets, which avoids higher-order resolvents due to independence of the matrices (except for their Section 5.2, which considers some dependencies under random signal assumptions).
>
> **Weakness [W4]**
> > Proposition 3.4 relies on the assumption that the parameter vector follows an isotropic Gaussian prior. In practice, model parameters often exhibit sparsity, structure, or anisotropy.
>
> Please see Section 3 in our anonymous site here **https://sites.google.com/view/icmlsubmission22249**, for the experiments on **anisotropic signal**. Overall, the results are consistent with our observations in the isotropic signal in our submission.
>
> The key reason we analyze isotropic-signal case is that we have a closed form for the Bayes-optimal risk $R^{\star}$. These allow for a clean comparison between the optimal SD and the optimal model. For generic setting, we do not have the closed-form risk for the optimal model to compare with, although we have the closed-form expression for optimal SD risk (Theorem 3.2).
>
> **References** Please kindly see the listed reference (or search the citation key) from our rebuttal for Reviewer $\texttt{6Yzx}$ at the end of the page.

---

> > ### Author Rebuttal · Reviewer_FU7g · 2026-04-01
> >
> > Thank you for the clarification. I will raise my score accordingly.

---

> > > ### Author Response · Authors · 2026-04-06
> > >
> > > Thank you for your endorsement and for increasing the score. Please let us know if you have any further questions about our rebuttal, we would be happy to continue the discussion or address any additional comments.
> > >
> > > Thank you again for your thoughtful review and valuable feedback.

---

### Official Review · Reviewer_TWSo · 2026-03-12

**Soundness:** 4
**Presentation:** 4
**Significance:** 2
**Originality:** 3
**Overall Recommendation:** 5
**Confidence:** 3

**Summary:**

This paper investigates self-distillation (SD) within the framework of ridge regression by introducing an unconstrained mixing weight $\xi \in \mathbb{R}$. The authors theoretically prove that the optimally mixed student strictly outperforms the teacher model at any non-stationary regularization level. Furthermore, they provide precise asymptotic characterizations of the SD risk in the high-dimensional proportional regime and propose a highly practical, consistent one-shot tuning method based on Generalized Cross-Validation (GCV) that avoids retraining.

**Compliance With Llm Reviewing Policy:**

Affirmed.

**Final Justification:**

I consider my concerns to be fully resolved and increase my overall score accordingly.

**Key Questions For Authors:**

I am curious about the extent to which the theoretical insights in this paper might extend beyond the ridge regression setting studied here. In many practical distillation pipelines for deep models, the backbone is frozen and only the final linear head is trained. In such cases, the learning problem effectively reduces to ridge regression on fixed features. Do the authors expect the main phenomena observed in this paper (e.g., strict improvement under optimal mixing or the sign rule for the optimal mixing weight) to persist in this “head-only distillation” setting? If so, it would be interesting to hear the authors’ perspective, and potentially see whether a simple experiment on frozen pretrained features exhibits similar behavior.

**Limitations:**

Yes.

**Strengths And Weaknesses:**

strength

1. The paper rigorously proves that optimal mixing weights can be negative to correct over-regularization, breaking the conventional $\xi \in [0, 1]$ constraint and offering a fresh mathematical perspective on self-distillation dynamics.
2. The proposed one-shot GCV tuning method practically eliminates the heavy computational overhead of standard grid-search cross-validation, bridging the gap between theoretical analysis and practical application.



 weakness

1. The theoretical guarantees are strictly confined to linear smoothers, leaving it unclear how well these structural identities and negative weight mechanisms transfer to deep neural networks characterized by dynamic representation learning.
2. The empirical evaluation primarily compares self-distillation only against the base ridge teacher; incorporating additional standard baselines, such as label smoothing, would significantly strengthen the linear probing experiments.

---

> ### Author Rebuttal · Authors · 2026-03-28
>
> Thank you for taking the time to review our paper and providing valuable suggestions. We appreciate your feedback and have addressed the identified weaknesses and questions below.
>
> **Weakness [W1]**
> > The theoretical guarantees are strictly confined to linear smoothers, leaving it unclear how well these structural identities and negative weight mechanisms transfer to deep neural networks characterized by dynamic representation learning.
>
> We agree that studying self-distillation in more general settings than linear smoothers is a research direction that is both very interesting and important. In this paper, we focus on ridge regression, a widely used regression methodology that is especially amenable to rigorous statistical analysis because of its connections to powerful results from random matrix theory. Indeed, our results paint a fairly complete picture of self-distillation for ridge regression. We expect that investigating non-linear smoothers and other types of losses will likely lead to additional technical and conceptual challenges that are beyond the scope of this paper.
>
> Nevertheless, we have empirically observed that some of our results extend. To demonstrate the **strict improvement and sign rule properties in a non-linear training dynamic**, we train the last-layer classifier of a pre-trained ResNet-18 using cross-entropy loss for classification on CIFAR-10. Please refer to **Section 1** of the following anonymous site: **https://sites.google.com/view/icmlsubmission22249**. In summary, across multiple settings (one-hot labels, label noise, and label smoothing), these properties hold to a good extent. They are most clearly observed in the label noise setting (see **Figure 1b**): (i) strict improvement: the SD classifier strictly improves the teacher at non-stationary $\lambda$; (ii) sign rule of the mixing weight: for under-regularized $\lambda$, a positive $\xi$ is preferred, while for over-regularized $\lambda$, a negative $\xi$ is preferred. We are happy to include these insights in our revision if the reviewers think that they would be helpful for our work.
>
> Furthermore, with the availability of large pre-trained networks, linear probing (freezing all pre-trained layers and training a single linear head) is widely used. This reduces to a ridge regression problem (smooth in $y$), where our analysis applies under test squared loss, including out-of-distribution settings. Squared loss has also been shown to perform well for classification tasks, achieving comparable generalization to cross-entropy trained models [HB21, DCO20]. This perspective aligns with your question [Q1] below.
>
> **Weakness [W2]**
> > The empirical evaluation primarily compares self-distillation only against the base ridge teacher; incorporating additional standard baselines, such as label smoothing, would significantly strengthen the linear probing experiments.
>
> Thank you for your suggestion. We have considered two other baselines in our experiment on classification task: label smoothing and label corruption. Please see the comparison with the teacher trained on label smoothing and label corruption in Figure 1b and 1c in the linear probing CIFAR-10 experiment using cross-entropy loss in our anonymous site. We observe that strict improvement and the sign rule hold in these settings as well.
>
>
> **Question [Q1]**
> > Do the authors expect the main phenomena observed in this paper (e.g., strict improvement under optimal mixing or the sign rule for the optimal mixing weight) to persist in this “head-only distillation” setting? If so, it would be interesting to hear the authors’ perspective, and potentially see whether a simple experiment on frozen pretrained features exhibits similar behavior.
>
> We agree that, for frozen pre-trained features, training the last linear layer is equivalent to a ridge regression problem. Therefore, the observed phenomena in our work hold in this case when evaluated under test squared risk, as our theory applies to any fixed features.
>
> In our paper, we had a linear probing on ResNet-18 pre-trained features using squared loss as the test metric, and all phenomena hold (see Figure 2c for CIFAR-10 and Figure 9 for CIFAR-100 in our submission). Interestingly, although $\xi$ is tuned to minimize squared loss, when evaluating top-1 accuracy, the strict improvement and sign rule for $\xi$ still hold to a good extent. Please see Section 2 of our anonymous site for more details (**https://sites.google.com/view/icmlsubmission22249**). We also report linear probing with cross-entropy loss experiments in Section 1 of the same link.
>
> For our asymptotic results (Section 3) based on random matrix theory (RMT), we also note that prior work suggests neural representations approximately exhibit RMT-like feature behavior [WHS22, DP24].
>
> **Reference** Due to space constraints, please see the references listed (or search the citation key) in our rebuttal for Reviewer $\texttt{6Yzx}$ at the end of the page.

---

> > ### Author Rebuttal · Reviewer_TWSo · 2026-04-01
> >
> > Thank you for the detailed and thorough rebuttal. The authors have convincingly addressed the main concerns, and the additional experimental results further strengthen the paper by demonstrating that the key phenomena extend beyond the original setting. Therefore, I consider my concerns to be fully resolved and increase my overall score accordingly.

---

> > > ### Author Response · Authors · 2026-04-06
> > >
> > > We would like to thank you for your endorsement and for increasing the score. Thank you again for your thoughtful reviews and valuable feedback.

---

### Official Review · Reviewer_Rcqi · 2026-03-16

**Soundness:** 3
**Presentation:** 3
**Significance:** 3
**Originality:** 3
**Overall Recommendation:** 5
**Confidence:** 3

**Summary:**

The paper analyzes self-distillation (SD) for ridge regression when the mixing weight ξ between ground-truth labels and teacher predictions is allowed to be any real number, not just in [0,1]. There are three main results: (1) a finite-sample, assumption-free proof that the optimally mixed student strictly beats the teacher at every λ where the risk derivative is nonzero, plus a sign rule tying ξ⋆ to R’(λ); (2) exact risk asymptotics in the proportional regime (p/n → γ) for anisotropic covariance and deterministic signals; (3) a one-shot GCV estimator for ξ⋆ that doesn’t require retraining or hold-out sets. Extensions cover multi-round SD, kernel ridge, and generalized ridge.

**Compliance With Llm Reviewing Policy:**

Affirmed.

**Final Justification:**

They addressed my concerns

**Key Questions For Authors:**

1. Can you report representative absolute and relative test-risk reductions on the real datasets at a few reference regularization levels, rather than only showing the risk curves?

2. Theorem 4.1 gives consistency under proportional asymptotics. Do you have any finite-sample guidance, empirical scaling study, or heuristic rate intuition for when the one-shot estimator should be reliable in practice?

3. For the recursive multi-round scheme, does the risk sequence converge, and if so, what is the limiting predictor/risk?

4. Your sign-rule derivation appears to rely on the linear-smoother structure in the labels. Do you have intuition for whether any analogue might survive for genuinely nonlinear learners such as logistic regression or neural networks?

**Limitations:**

Yes

**Strengths And Weaknesses:**

### Strengths

Theorem 2.2 is the highlight.  The connection between ξ⋆ and −R’(λ) is clean and gives you immediate intuition: if you’re over-regularized, the optimal mixing is negative (“pro-learning”), if under-regularized, it’s positive. And this holds with zero distributional assumptions, just conditioned on data.

The proportional asymptotic analysis is also interesting. The block linearization trick to get higher-order resolvent DEs is not straightforward and the resulting formulas (Theorem 3.2) match experiments well even at moderate n,p (Figure 4).

The one-shot tuning is the most practically useful piece. Grid searching over (λ, ξ) pairs with CV is painful, especially when n ≈ p. The GCV-based estimator sidesteps this entirely and the real-data experiments show it tracks the oracle ξ⋆ well.

### Weaknesses

The main theory depends on the predictor being a linear smoother in the labels y. Section 5.3 does a good job extending to generalized ridge and kernel ridge, but these are still linear smoothers. It remains unclear how far the sign rule or strict improvement results extend to genuinely nonlinear training dynamics (e.g., logistic regression, neural nets).

The "strict improvement" theorem is mathematically true but can overstate practical significance: gains vanish near the ridge-optimal penalty (the paper itself notes R⋆_sd(λ⋆) = R(λ⋆)), so the practical value is mainly as a cheap correction for a mis-regularized teacher rather than an improvement over a well-tuned one. The paper does frame SD this way in the conclusion, but the abstract and introduction lean more toward the "strict improvement at every λ" angle, which could set misleading expectations.

Proposition 2.3 gives a sufficient condition for SD to beat optimally tuned ridge globally, but the paper provides limited evidence about how tight or informative this curvature test is. Table 2 checks it on four datasets and Eq. (8) holds on two of them, but since it's only sufficient, we can't tell from the two "no" cases whether global gains are actually absent or just not captured by this particular test.

The fresh-X analysis is isotropic only. Given that the anisotropic case is the main selling point elsewhere, this feels like an incomplete story.

---

> ### Author Rebuttal · Authors · 2026-03-28
>
> Thank you for your positive and encouraging feedback. We appreciate your insightful questions and suggestions. In response to the concerns raised, we have conducted additional experiments to clarify the implications of our results in non-linear training settings. Please see Section 1 of our anonymous site: **https://sites.google.com/view/icmlsubmission22249**
>
> **Weakness [W1]**
> > It remains unclear how far the sign rule or strict improvement results extend to genuinely nonlinear training dynamics (e.g., logistic regression, neural nets).
>
> Please kindly see our answer for **[W1]** from Reviewer $\texttt{TWSo}$.
>
> **Weakness  [W2]**
> > The ``strict improvement" theorem is mathematically true but can overstate practical significance.
>
> We agree with the reviewer that the main practical implication of our theoretical results, namely that SD can provably tune a suboptimal ridge regression more efficiently than CV-based methods, should be emphasized earlier in the manuscript. We will highlight this contribution in both the abstract and the introduction. Thank you for the suggestion.
>
> We also note that such pointwise gains are important because finding a globally optimal $\lambda$ is often challenging. For example, [SFUB21] shows that the leave-one-out cross-validation loss is generally neither convex nor quasi-convex in $\lambda$.
>
> **Weakness  [W3]**
> > Proposition 2.3 gives a sufficient condition for SD to beat optimally tuned ridge globally, but the paper provides limited evidence about how tight or informative this curvature test is.
>
> We note that all critical points of the SD risk curve can be obtained from equation (20) (page 16). However, characterizing these points and identifying the global minimum is nontrivial and likely requires additional assumptions. We present a setting on the Air Quality dataset where the test fails at the optimal $\lambda$, yet a global gain in SD risk still exists at a different $\lambda$, see Section 5 of our anonymous site above.
>
> You are right regarding the two “no” cases in Table 2 and whether global gains are truly absent. To clarify, in Table 2 we define the absence of a global gain as follows: over a grid of 200 $\lambda$ values in $[0.001, 50]$, the difference between the minimum SD student risk (over the grid) and the teacher risk is less than $0.001$.
>
> We will clarify these points in our revision.
>
> **Weakness  [W4]**
> > The fresh-X analysis is isotropic only.
>
> There are many variants of SD. Among those we examined, we showcase the fresh-$X$ case (with the same sample size) because it is provably worse than regular SD in the simplest setting of isotropic covariates. We found this somewhat unexpected and intriguing, and included it as a cautionary (but helpful) example of using SD with fresh-$X$ samples. It is not intended to provide a complete treatment of the fresh-$X$ case.
>
> **Question [Q1]**
> > Can you report representative absolute and relative test-risk reductions on the real datasets, rather than only showing the risk curves?
>
> Please see Section 4 of our anonymous site for the table of absolute values reported in our experiments.
>
> **Question [Q2]**
> > Theorem 4.1 gives consistency under proportional asymptotics. Do you have any finite-sample guidance, empirical scaling study, or heuristic rate intuition for when the one-shot estimator should be reliable in practice?
>
> We agree that Theorem 4.1 is asymptotic. That said, both our experiments and prior high-dimensional CV theory suggest that the one-shot estimator is reliable at moderate sample sizes. Please see Section 8 of our anonymous site for more details and discussions on the evidence, heuristic rates, and related works that could be combined with our work to derive finite-sample bounds under stronger distributional assumptions.
>
> **Question [Q3]**
> > For the recursive multi-round scheme, does the risk sequence converge, and if so, what is the limiting predictor/risk?
>
> Please see Section 6 of our anonymous site for experiments on multiple-round distillation.
>
> In our work, we consider two variants of multi-round optimal SD: (i) recursive, where the predictor learned in the previous round (starting from the original ridge model) is used as the teacher for the next round; and (ii) anchored, where the original ridge teacher is reused in every round. In summary, the recursive method converges quickly (within two rounds), whereas anchored mixing exhibits unstructured, non-monotonic behavior in the experiments.
>
> The limiting predictor/risk under the recursive method is of independent interest, and we leave it for future work since the expressions become quite involved.
>
> **Question [Q4]**
> > Do you have intuition for whether any analogue might survive for genuinely nonlinear learners such as logistic regression or neural networks?
>
> Same as [W1] above.
>
> **References** Due to space constraints, please see the references listed (or search the citation key) in our rebuttal for Reviewer $\texttt{6Yzx}$ at the end of the page.

---

> > ### Author Rebuttal · Reviewer_Rcqi · 2026-04-02
> >
> > They answered all my questions

---

> > > ### Author Response · Authors · 2026-04-06
> > >
> > > We are glad to hear that our responses address your questions. We would like to thank the reviewer again for your thoughtful reviews and valuable feedback.

---

### Decision · Program_Chairs · 2026-04-30

**Decision:**

Accept (regular)

**Comment:**

This paper studies self-distillation for the ridge regression problem with unconstrained mixing weights, presenting new theoretical results supported by experiments. The reviews were somewhat mixed, with the majority voting to accept (5,5,5) and one weak reject (3).

The reviewers praised the method's originality and the non-triviality of the proportional asymptotic analysis. The reviewers were especially excited about the one-shot tuning component due to its simplicity and efficiency. They also liked the clearness of presentation and the soundness of the theory. The main concern in the negative review was the narrow focus of the proposed method on linear cases, which cannot be applied to non-linear models, thereby limiting its practicality.

After reading the paper, the reviews, the rebuttal comments, and the discussion, I feel that the positive contributions outweigh the concerns raised by reviewer FU7g, and thus I recommend acceptance. Specifically, I think the proposed solution is sufficiently non-trivial to warrant acceptance, in the hope that this result will spur follow-up work extending it to nonlinear cases.